# Efficient and Minimax Optimal In-context Nonparametric Regression with Transformers

**Michelle Ching** [1] [*] **Ioana Popescu** [2] [*] **Nico Smith** [1] [*] **Tianyi Ma** [1]
**William G. Underwood** [1] **Richard J. Samworth** [1]

## Abstract

We study in-context learning for nonparametric regression with $\alpha$-Hölder smooth regression functions, for some $\alpha > 0$. We prove that, with $n$ in-context examples and $d$-dimensional regression covariates, a pretrained transformer with $\Theta(\log n)$ parameters and $\Omega\big(n^{2\alpha/(2\alpha+d)} \log^3 n\big)$ pretraining sequences can achieve the minimax optimal rate of convergence $O\big(n^{-2\alpha/(2\alpha+d)}\big)$ in mean squared error. Our result requires substantially fewer transformer parameters and pretraining sequences than previous results in the literature. This is achieved by showing that transformers are able to approximate local polynomial estimators efficiently by implementing a kernel-weighted polynomial basis and then running gradient descent.

## 1. Introduction

Deep learning models based on the transformer architecture (Vaswani et al., 2017) have achieved remarkable empirical successes in recent years; prominent examples include large language models (Devlin, 2018; Shoeybi et al., 2019; OpenAI, 2023) and contemporary computer vision models (Dosovitskiy et al., 2021). By allowing data points to interact directly via the attention mechanism, such models enjoy a high degree of flexibility, while remaining trainable and avoiding overfitting.

In-context learning (ICL) offers a framework for examining the generalisation abilities of large language models (Brown et al., 2020; Garg et al., 2022). When presented with a prompt (context) containing a few input–output examples, pretrained transformers can often generalise to new unseen queries without requiring any parameter updates. Empirical

---

[*]Equal contribution [1]Statistical Laboratory, University of Cambridge, Cambridge, UK [2]Department of Computer Science, ETH Zürich, Zürich, Switzerland. Correspondence to: Tianyi Ma <tm681@cam.ac.uk>.

*Proceedings of the $43^{rd}$ International Conference on Machine Learning*, Seoul, South Korea. PMLR 306, 2026. Copyright 2026 by the author(s).

studies have observed this behaviour across a diverse range of tasks, including translation, answering questions, and arithmetic (Brown et al., 2020).

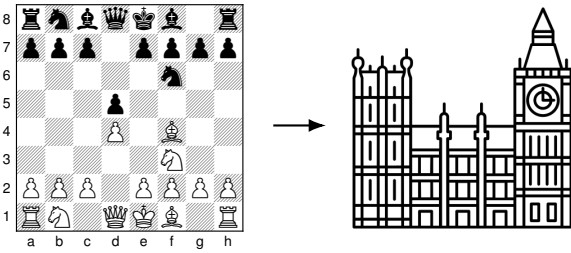

*Figure 1.* What is the relationship between these two concepts?

As an example of ICL behaviour, consider the concept-to-concept relationship depicted in Figure 1. Here, a chess opening known as the London System is paired with the iconic London landmark Big Ben. A language model might learn this type of relationship via a text prompt as shown in Figure 2. This prompt includes several well-known chess openings (London System, French Defence, Spanish Game, King's Indian Defence, Scotch Game), paired with famous landmarks from the corresponding countries. As such, a suitable response for the query point (the Italian Game) might be 'Colosseum'; indeed, this is the response provided by OpenAI's GPT-5 mini model. This large language model is able to identify the abstract relationship from chess opening to location to landmark, despite not having been explicitly trained for this task.

Recently, pretrained transformers have demonstrated the ability to achieve state-of-the art empirical performance on various tasks with tabular data via ICL (Hollmann et al., 2025), outperforming classical approaches including popular tree-based boosting algorithms. One of the central such challenges, and the focus of this work, is that of nonparametric regression.

1. d4 d5 2. Nf3 Nf6 3. Bf4 $\longrightarrow$ Big Ben

1. e4 e6 2. d4 d5 $\longrightarrow$ Eiffel Tower

1. e4 e5 2. Nf3 Nc6 3. Bb5 $\longrightarrow$ Sagrada Família

1. d4 Nf6 2. c4 g6 $\longrightarrow$ Taj Mahal

1. e4 e5 2. Nf3 Nc6 3. d4 $\longrightarrow$ Loch Ness

1. e4 e5 2. Nf3 Nc6 3. Bc4 $\longrightarrow$ ?

*Figure 2.* Example of ICL in large language models. Here, there are $n = 5$ in-context examples and one query point in the prompt.

## 1.1. Related Work

The theoretical properties of ICL have been extensively studied in the regression and classification settings. Several prior works have shown that transformers are able to approximate classical algorithms and generalise to unseen tasks. For example, transformers are able to learn parametric models via gradient descent (Ahn et al., 2023; Akyürek et al., 2023; Bai et al., 2023; Li et al., 2023; von Oswald et al., 2023), implement classifiers by acting as meta-optimisers (Dai et al., 2023), perform nonparametric regression using basis expansions (Kim et al., 2024) or by fitting Nadaraya–Watson estimators (Shen et al., 2025), and solve reinforcement learning problems (Lin et al., 2024). Other papers study implicit Bayesian inference for next token prediction (Xie et al., 2022), Bayesian frameworks for tabular foundation models (Nagler, 2023), adaptivity and distributional robustness of ICL (Ma et al., 2025; Wakayama & Suzuki, 2025), task generalisation (Abedsoltan et al., 2025), prompt engineering (Nakada et al., 2025), and transformer training dynamics (Oko et al., 2024; Zhang et al., 2024; Kuwataka & Suzuki, 2025; Lu et al., 2025). There have also been many empirical studies of ICL phenomena in recent years. For example, Bhattamishra et al. (2024) consider ICL of discrete functions, Qin et al. (2024) explore the factors influencing multi-modal ICL in vision large language models, Sia et al. (2024) investigate where in the transformer architecture ICL occurs, and Bertsch et al. (2025) conduct a systematic study of long-context ICL.

Closest to our work are the recent papers by Kim et al. (2024) and Shen et al. (2025). Kim et al. (2024) prove that transformers can learn nonparametric regression problems in context by estimating the coefficients of the regression functions from a B-spline wavelet basis expansion. They achieve nearly minimax optimal rates in $n$, the number of in-context examples, when the regression functions belong to a suitable Besov space. Shen et al. (2025) show that transformers are able to approximate Nadaraya–Watson (local constant) estimators. For $\alpha$-Hölder regression functions, where $\alpha \in (0, 1]$, they achieve optimal rates, and further they show adaptation to covariates supported on a low-dimensional manifold, thereby avoiding the curse of di-

mensionality. Both of these works require transformers with polynomially many parameters (in $n$) in order to achieve minimax optimal rates, leading to restrictive assumptions on the number of pretraining sequences.

## 1.2. Contribution

We consider the problem of in-context nonparametric regression using transformers. In particular, we show that transformers trained via empirical risk minimisation, under mild architecture conditions, achieve the minimax optimal rate (in the number $n$ of in-context examples) for nonparametric regression under mean squared prediction risk. Our main result, Theorem 3.2, holds whenever the underlying regression functions are $\alpha$-Hölder, for some $\alpha > 0$; the corresponding minimax rate is $n^{-2\alpha/(2\alpha+d)}$, where $d$ is the dimension of the covariates. Compared with existing works, our approach attains this optimal rate while imposing much milder conditions on both the number of pretraining sequences and the number of transformer parameters. Our proofs use linear attention for simplicity, but they can be generalised to ReLU or softmax attention with minor changes; see the discussion at the end of Section 3.

Our primary technical innovation is to show that transformers with single-head attention layers and $\Theta(\log n)$ parameters are able to attain the optimal rate of convergence. This improves on the $\Theta(n^{d/(2\alpha+d)})$ parameters required by Kim et al. (2024) and $\Theta(n)$ parameters needed by Shen et al. (2025). As a direct consequence, we also require substantially fewer pretraining sequences. In order to prove this, we demonstrate that such transformers are able to approximate local polynomial estimators of suitable degree (Theorem 3.1), which amounts to solving a weighted least squares problem (4) in the monomial basis. Our approach consists of constructing a transformer which, given an input prompt, first constructs a kernel-weighted monomial basis matrix and then uses this matrix to perform $\Theta(\log n)$ steps of gradient descent towards the desired local polynomial least-squares solution (4). An advantage of gradient descent for this problem is that it avoids the need to compute matrix inverses. Optimality then follows by combining well-known properties of local polynomial estimators (Theorem 2.5) with a bound on the expected excess risk of the transformer-based estimator, obtained via empirical process theory and covering number bounds.

## 1.3. Overview

In Section 2, we present a mathematical formulation of our in-context nonparametric regression problem with Hölder-smooth regression functions, define our transformer class using linear attention and ReLU feed-forward layers, and provide some auxiliary definitions and results concerning classical local polynomial estimators (Theorem 2.5). Sec-

tion 3 gives our main results, beginning with some novel approximation theory for transformers in Theorem 3.1. Combining this with a bound on the expected excess risk for empirical risk-minimising transformers yields our primary contribution as Theorem 3.2. Here, we establish the minimax optimality of transformers for performing in-context nonparametric regression under standard Hölder smoothness assumptions, and under substantially weaker conditions on the number of transformer parameters and on the number of pretraining sequences than existing results in the literature. In Section 4, we provide some insight into the main ideas underlying the proofs of our main theorems. First, we explain how to construct a transformer that uses gradient descent to approximate the output of a local polynomial estimator, and second we bound the expected excess risk of our estimator by controlling the covering numbers of the corresponding transformer class.

## 1.4. Notation

We write $\mathbb{N} := \{1, 2, \ldots\}$ and $\mathbb{N}_0 := \{0, 1, 2, \ldots\}$. For $n \in \mathbb{N}$, we set $[n] := \{1, \ldots, n\}$. For $d \in \mathbb{N}$ and a multi-index $\nu \in \mathbb{N}_0^d$, define $|\nu| := \sum_{j=1}^d \nu_j$ and $\nu! := \prod_{j=1}^d \nu_j!$ and, for a sufficiently smooth function $g : \mathbb{R}^d \to \mathbb{R}$, define the order-$\nu$ partial derivative $\partial_\nu g(x) := \partial^{|\nu|} g(x) / \prod_{j=1}^d \partial x_j^{\nu_j}$. For $x \in \mathbb{R}^d$, let $\|x\|_2^2 := \sum_{j=1}^d x_j^2$ and $\|x\|_\infty := \max_{j \in [d]} |x_j|$. If $x, y \in \mathbb{R}$, we write $x \wedge y := \min\{x, y\}$ and $x \vee y := \max\{x, y\}$, and set $x_+ := x \vee 0$. For a matrix $A$, we write $\|A\|_{\mathrm{op}}$ for the $\ell_2$–$\ell_2$ operator norm, $\|A\|_{\max}$ for its maximum absolute entry, and $\lambda_{\min}(A)$ and $\lambda_{\max}(A)$ for its minimum and maximum eigenvalues, respectively. For non-negative sequences $(a_n)$ and $(b_n)$, we write $a_n = O(b_n)$ if there exists $C > 0$ and $N \in \mathbb{N}$ such that $a_n \leq C b_n$ for all $n \geq N$. Similarly, we write $a_n = \Omega(b_n)$ if there exists $c > 0$ and $N \in \mathbb{N}$ such that $a_n \geq c b_n$ for all $n \geq N$; thus $a_n = O(b_n)$ if and only if $b_n = \Omega(a_n)$. If $a_n = O(b_n)$ and $a_n = \Omega(b_n)$, then we write $a_n = \Theta(b_n)$. For a non-empty normed space $(X, \|\cdot\|)$, $A \subseteq X$ and $\delta > 0$, we say that a non-empty finite set $A' \subseteq X$ is a $\delta$-*cover* of $A$ if $\sup_{x \in A} \min_{x' \in A'} \|x - x'\| \leq \delta$. We write $N(A, \delta, \|\cdot\|)$ for the minimal cardinality of such a cover, when one exists.

**Definition 1.1** (Hölder-ball). Let $d \in \mathbb{N}$ and $\alpha, M > 0$, and take $\underline{\alpha} := \lceil \alpha \rceil - 1$ to be the largest integer strictly less than $\alpha$. Write $\mathcal{H}(d, \alpha, M)$ for the set of $\underline{\alpha}$-times differentiable functions $g : [0, 1]^d \to [-M, M]$ that satisfy

$$\max_{\nu \in \mathbb{N}_0^d : |\nu| = \underline{\alpha}} \left| \partial_\nu g(x) - \partial_\nu g(x') \right| \leq M \|x - x'\|_2^{\alpha - \underline{\alpha}}$$

for $x, x' \in [0, 1]^d$.

## 2. Problem Set-up

In this section, we formalise our in-context nonparametric regression problem, define our transformer class and

provide some auxiliary results on local polynomial estimators. In the ICL framework, pretraining data is modelled as a collection of $\Gamma$ pretraining sequences, each generated according to a similar (but not identical) mechanism. We consider an in-context regression model, supposing that each such sequence, indexed by $\gamma \in [\Gamma]$, consists of covariates $X_i^{(\gamma)} \in \mathbb{R}^d$ and responses $Y_i^{(\gamma)} \in \mathbb{R}$ for $i \in [n]$. We assume that the covariates $X_i^{(\gamma)}$ and errors $\varepsilon_i^{(\gamma)} \in \mathbb{R}$ are all drawn from a common distribution; the responses are then generated as $Y_i^{(\gamma)} = m^{(\gamma)}(X_i^{(\gamma)}) + \varepsilon_i^{(\gamma)}$, where the regression functions $m^{(\gamma)}$ are drawn randomly from a distribution, and therefore may vary across pretraining indices.

### 2.1. Data Generating Mechanism

Let $P_{X,\varepsilon}$ be a distribution on $[0, 1]^d \times [-1, 1]$ such that if $(X, \varepsilon) \sim P_{X,\varepsilon}$ then $X$ admits a Lebesgue density function $f_X$ with $0 < c_X \leq f_X(x) \leq C_X < \infty$ for all $x \in [0, 1]^d$ and $\varepsilon$ satisfies $\mathbb{E}(\varepsilon \mid X) = 0$ almost surely with $\sigma^2 := \mathbb{E}(\varepsilon^2) < \infty$. Given $\alpha, M > 0$, let $P_{\mathcal{H}}$ be a distribution on $\mathcal{H}(d, \alpha, M)$ with respect to the Borel $\sigma$-algebra associated with $\|\cdot\|_\infty$. Now fix $\Gamma, n \in \mathbb{N}$. For $\gamma \in [\Gamma]$, let $m^{(\gamma)} \sim P_{\mathcal{H}}$ be independent. Further, for $\gamma \in [\Gamma]$ and $i \in [n + 1]$, and conditional on $(m^{(1)}, \ldots, m^{(\Gamma)})$, let $(X_i^{(\gamma)}, \varepsilon_i^{(\gamma)}) \sim P_{X,\varepsilon}$ be independent, and set $Y_i^{(\gamma)} := m^{(\gamma)}(X_i^{(\gamma)}) + \varepsilon_i^{(\gamma)}$. For $\gamma \in [\Gamma]$, define

$$\mathcal{D}_n^{(\gamma)} := \left( X_i^{(\gamma)}, Y_i^{(\gamma)} \right)_{i \in [n]}.$$

For each $\gamma \in [\Gamma]$, we try to predict $Y_{n+1}^{(\gamma)}$ given the $n$ examples $\mathcal{D}_n^{(\gamma)}$ and a query $X_{n+1}^{(\gamma)}$.

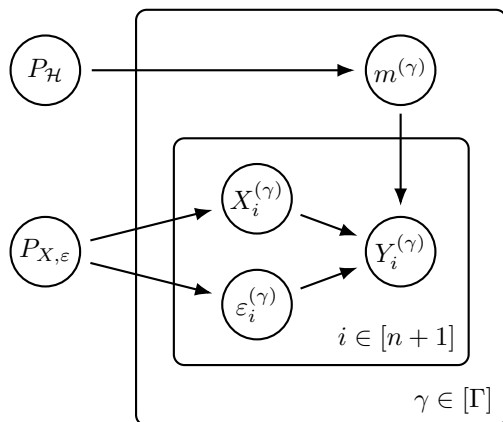

*Figure 3.* Plate diagram showing the data generating mechanism of our in-context nonparametric regression problem. For each pretraining sequence index $\gamma \in [\Gamma]$, a regression function is drawn from $P_{\mathcal{H}}$. Then, for each $i \in [n + 1]$, i.i.d. covariates $X_i^{(\gamma)}$ and errors $\varepsilon_i^{(\gamma)}$ are sampled from $P_{X,\varepsilon}$. Finally, responses are generated as $Y_i^{(\gamma)} = m^{(\gamma)}(X_i^{(\gamma)}) + \varepsilon_i^{(\gamma)}$.

## 2.2. Empirical and Population Risk

Let $\mathcal{F}$ be a class of functions from $\big([0,1]^d \times \mathbb{R}\big)^n \times [0,1]^d$ to $\mathbb{R}$ and define the *empirical risk minimiser* $\hat{f}_\Gamma \in \arg\min_{f \in \mathcal{F}} \hat{R}_\Gamma(f)$, where

$$\hat{R}_\Gamma(f) \coloneqq \frac{1}{\Gamma} \sum_{\gamma=1}^{\Gamma} \Big\{ Y_{n+1}^{(\gamma)} - f\big(\mathcal{D}_n^{(\gamma)}, X_{n+1}^{(\gamma)}\big) \Big\}^2. \quad (1)$$

The *population risk* of a measurable function $f : \big([0,1]^d \times \mathbb{R}\big)^n \times [0,1]^d \to \mathbb{R}$ is

$$R(f) \coloneqq \mathbb{E}\Big( \big\{ Y_{n+1} - f\big(\mathcal{D}_n, X_{n+1}\big) \big\}^2 \Big), \quad (2)$$

where $\mathcal{D}_n \coloneqq (X_i, Y_i)_{i \in [n]}$ and $(X_{n+1}, Y_{n+1})$ are independent copies of $\mathcal{D}_n^{(1)}$ and $(X_{n+1}^{(1)}, Y_{n+1}^{(1)})$ respectively. Throughout the paper, we quantify performance of a procedure by bounding its population risk.

## 2.3. Transformer Class

We take $\mathcal{F}$ to be a parametrised class of transformer neural networks. The architecture of such a network consists of a series of *transformer blocks*, each of which is composed of a (single-head) *linear attention layer* and a *feed-forward network layer*; see Figure 4. We remark that our proofs would also carry over, with minor changes, for ReLU attention, softmax attention or other attention mechanisms that can approximate linear attention; see the discussion following Theorem 3.2. However, in our main exposition, we restrict to linear attention for simplicity.

**Definition 2.1** (Linear attention layer). Let $n, d_\mathrm{e} \in \mathbb{N}$ and $\boldsymbol{Q}, \boldsymbol{K}, \boldsymbol{V} \in \mathbb{R}^{d_\mathrm{e} \times d_\mathrm{e}}$. Define $\mathrm{Attn}_{\boldsymbol{Q}, \boldsymbol{K}, \boldsymbol{V}} : \mathbb{R}^{(n+1) \times d_\mathrm{e}} \to \mathbb{R}^{(n+1) \times d_\mathrm{e}}$ by

$$\mathrm{Attn}_{\boldsymbol{Q}, \boldsymbol{K}, \boldsymbol{V}}(\boldsymbol{Z}) \coloneqq \boldsymbol{Z} + \boldsymbol{Z}\boldsymbol{Q}(\boldsymbol{Z}\boldsymbol{K})^\mathsf{T}\boldsymbol{Z}\boldsymbol{V}.$$

The parameters $\boldsymbol{Q}$, $\boldsymbol{K}$ and $\boldsymbol{V}$ are referred to as the *query*, *key* and *value* matrices, respectively.

**Definition 2.2** (Feed-forward network layer). Let $n, d_\mathrm{e}, d_\mathrm{ffn} \in \mathbb{N}$ and $\boldsymbol{W}_1 \in \mathbb{R}^{d_\mathrm{ffn} \times d_\mathrm{e}}$, $\boldsymbol{W}_2 \in \mathbb{R}^{d_\mathrm{e} \times d_\mathrm{ffn}}$, $b_1 \in \mathbb{R}^{d_\mathrm{ffn}}$ and $b_2 \in \mathbb{R}^{d_\mathrm{e}}$. Define $\mathrm{FFN}_{\boldsymbol{W}_1, \boldsymbol{W}_2, b_1, b_2} : \mathbb{R}^{(n+1) \times d_\mathrm{e}} \to \mathbb{R}^{(n+1) \times d_\mathrm{e}}$ by

$$\mathrm{FFN}_{\boldsymbol{W}_1, \boldsymbol{W}_2, b_1, b_2}(\boldsymbol{Z})$$
$$\coloneqq \boldsymbol{Z} + \big\{ \boldsymbol{W}_2 \mathrm{ReLU}(\boldsymbol{W}_1 \boldsymbol{Z}^\mathsf{T} + b_1 \mathbf{1}_{n+1}^\mathsf{T}) + b_2 \mathbf{1}_{n+1}^\mathsf{T} \big\}^\mathsf{T},$$

where $\mathrm{ReLU} : \mathbb{R} \to \mathbb{R}$ is defined by $\mathrm{ReLU}(x) \coloneqq x \vee 0$ and is applied entrywise.

The architecture given in Definition 2.2 is equivalent to applying a standard one-hidden layer feed-forward neural network with ReLU activation and skip connection (He et al., 2016) to each row in the input matrix $\boldsymbol{Z}$.

**Definition 2.3** (Transformer). Let $n, d_\mathrm{e}, d_\mathrm{ffn}, L \in \mathbb{N}$. For $\ell \in [L]$, take parameter vectors $\boldsymbol{\theta}^{(\ell)} \coloneqq \big(\boldsymbol{Q}^{(\ell)}, \boldsymbol{K}^{(\ell)}, \boldsymbol{V}^{(\ell)}, \boldsymbol{W}_1^{(\ell)}, \boldsymbol{W}_2^{(\ell)}, b_1^{(\ell)}, b_2^{(\ell)}\big) \in \mathbb{R}^{d_\mathrm{e} \times d_\mathrm{e}} \times \mathbb{R}^{d_\mathrm{e} \times d_\mathrm{e}} \times \mathbb{R}^{d_\mathrm{e} \times d_\mathrm{e}} \times \mathbb{R}^{d_\mathrm{ffn} \times d_\mathrm{e}} \times \mathbb{R}^{d_\mathrm{e} \times d_\mathrm{ffn}} \times \mathbb{R}^{d_\mathrm{ffn}} \times \mathbb{R}^{d_\mathrm{e}}$ and define $\mathrm{Block}_{\boldsymbol{\theta}^{(\ell)}} : \mathbb{R}^{(n+1) \times d_\mathrm{e}} \to \mathbb{R}^{(n+1) \times d_\mathrm{e}}$ by

$$\mathrm{Block}_{\boldsymbol{\theta}^{(\ell)}}(\boldsymbol{Z})$$
$$\coloneqq \mathrm{FFN}_{\boldsymbol{W}_1^{(\ell)}, \boldsymbol{W}_2^{(\ell)}, b_1^{(\ell)}, b_2^{(\ell)}} \circ \mathrm{Attn}_{\boldsymbol{Q}^{(\ell)}, \boldsymbol{K}^{(\ell)}, \boldsymbol{V}^{(\ell)}}(\boldsymbol{Z}).$$

Let $\boldsymbol{\theta} \coloneqq (\boldsymbol{\theta}^{(\ell)})_{\ell=1}^L$ and define the *transformer* $\mathrm{TF}_{\boldsymbol{\theta}} : \mathbb{R}^{(n+1) \times d_\mathrm{e}} \to \mathbb{R}^{(n+1) \times d_\mathrm{e}}$ by

$$\mathrm{TF}_{\boldsymbol{\theta}}(\boldsymbol{Z}) \coloneqq \mathrm{Block}_{\boldsymbol{\theta}^{(L)}} \circ \cdots \circ \mathrm{Block}_{\boldsymbol{\theta}^{(1)}}(\boldsymbol{Z}). \quad (3)$$

Finally, define $\mathcal{T}(d_\mathrm{e}, d_\mathrm{ffn}, L, B)$ to be the collection of all transformers of the form (3) with every entry of each parameter in $\boldsymbol{\theta}$ bounded in absolute value by $B > 0$.

The function $\mathrm{TF}_{\boldsymbol{\theta}}$ depends on the number $n$ of in-context examples only via the dimension of its argument $\boldsymbol{Z} \in \mathbb{R}^{(n+1) \times d_\mathrm{e}}$; its parametrisation is fully determined by $\boldsymbol{\theta}$ and is independent of $n$.

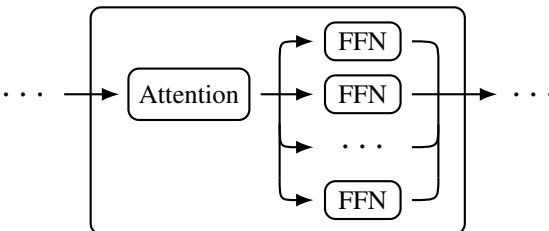

*Figure 4.* A single block in the transformer architecture, consisting of a single head of linear attention followed by a one-layer feed-forward neural network applied identically to every row.

## 2.4. Transformer-based Estimator

Let $d_\mathrm{e} \geq d + 2$. For $(x_i, y_i)_{i \in [n]} \in [0,1]^d \times \mathbb{R}$ and $x_{n+1} \in [0,1]^d$, define the embedding function $\mathrm{Embed}_{d_\mathrm{e}} : \big([0,1]^d \times \mathbb{R}\big)^n \times [0,1]^d \to \mathbb{R}^{(n+1) \times d_\mathrm{e}}$ by

$$\mathrm{Embed}_{d_\mathrm{e}}\big((x_i, y_i)_{i \in [n]}, x_{n+1}\big)$$
$$\coloneqq \begin{pmatrix} x_1^\mathsf{T} & y_1 & 0_{d_\mathrm{e}-d-2}^\mathsf{T} & 0 \\ \vdots & \vdots & \vdots & \vdots \\ x_n^\mathsf{T} & y_n & 0_{d_\mathrm{e}-d-2}^\mathsf{T} & 0 \\ x_{n+1}^\mathsf{T} & 0 & 0_{d_\mathrm{e}-d-2}^\mathsf{T} & 1 \end{pmatrix} \in \mathbb{R}^{(n+1) \times d_\mathrm{e}}.$$

We do not require any sophisticated forms of positional encoding: the $\mathrm{Embed}$ function merely identifies the test point $i = n + 1$ in its last column.

We also define $\mathrm{Read}_{M,d} : \mathbb{R}^{(n+1) \times d_\mathrm{e}} \to \mathbb{R}$ by

$$\mathrm{Read}_{M,d}(\boldsymbol{Z}) \coloneqq (-M) \vee \boldsymbol{Z}_{n+1, d+1} \wedge M,$$

where we recall that the regression functions are bounded by $M$.

**Definition 2.4** (Transformer-based estimator)**.** Let $n, d, d_{\mathrm{e}}, d_{\mathrm{ffn}}, L, \Gamma \in \mathbb{N}$ with $d_{\mathrm{e}} \geq d + 2$ and take $B > 0$. Consider the class of functions from $\left([0,1]^d \times \mathbb{R}\right)^n \times [0,1]^d$ to $\mathbb{R}$ given by $\mathcal{F}(d_{\mathrm{e}}, d_{\mathrm{ffn}}, L, B, M) := \left\{ \mathrm{Read}_{M,d} \circ f \circ \mathrm{Embed}_{d_{\mathrm{e}}} : f \in \mathcal{T}(d_{\mathrm{e}}, d_{\mathrm{ffn}}, L, B) \right\}$ and define the empirical risk minimiser

$$\hat{f}_\Gamma \in \underset{f \in \mathcal{F}(d_{\mathrm{e}}, d_{\mathrm{ffn}}, L, B, M)}{\mathrm{argmin}} \hat{R}_\Gamma(f),$$

where $\hat{R}_\Gamma$ is defined in (1). Thus, $\hat{f}_\Gamma$ is trained on the data $\left(X_i^{(\gamma)}, Y_i^{(\gamma)}\right)_{i \in [n+1], \gamma \in [\Gamma]}$.

We remark that, in practice, computation of the global empirical risk minimiser $\hat{f}_\Gamma$ is typically intractable, but gradient-based optimisation methods often yield good solutions. In this paper, we focus only on the statistical aspects of in-context learning; the analysis of transformer training dynamics is an independent and challenging problem.

### 2.5. Local Polynomial Estimators

Let $K : \mathbb{R}^d \to [0, C_K]$ be a Borel measurable function supported on $[-1,1]^d$ satisfying $|K(x) - K(x')| \leq L_K \|x - x'\|_\infty$ for all $x, x' \in \mathbb{R}^d$ and some $L_K > 0$. We do not assume that $K$ integrates to 1. Assume that $K(x) \geq c_K$ for $x \in [-c_K, c_K]^d$, where $c_K \in (0, 1]$. Let $h > 0$ be the bandwidth and define $K_h(\cdot) := K(\cdot/h)/h^d$. For $p \in \mathbb{N}_0$, write $D := \binom{d+p}{p}$ and let $P_h : \mathbb{R}^d \to \mathbb{R}^D$ be defined by $P_h(x) := \left(x^\nu/(\nu! h^{|\nu|}) : \nu \in \mathbb{N}_0^d, 0 \leq |\nu| \leq p\right) \in \mathbb{R}^D$, with components ordered in increasing lexicographic ordering in $\nu$, so that its first coordinate is $P_h(x)_1 = 1$. For $x \in [0,1]^d$, define the random diagonal matrix $\boldsymbol{K}_h(x) \in \mathbb{R}^{n \times n}$ with $i$th diagonal entry given by $\boldsymbol{K}_h(x)_{ii} := K_h(X_i - x)$ for $i \in [n]$. Let $\boldsymbol{P}_h(x) \in \mathbb{R}^{n \times D}$ be defined by $\boldsymbol{P}_h(x)_{ij} := P_h(X_i - x)_j$ for $i \in [n]$ and $j \in [D]$. Define $\tilde{\boldsymbol{X}} := n^{-1/2} \boldsymbol{K}_h(X_{n+1})^{1/2} \boldsymbol{P}_h(X_{n+1}) \in \mathbb{R}^{n \times D}$ and $\tilde{Y} := n^{-1/2} \boldsymbol{K}_h(X_{n+1})^{1/2} Y \in \mathbb{R}^n$, where $Y := (Y_1, \ldots, Y_n)^\top$. Let

$$\begin{aligned} w_* &:= \underset{w \in \mathbb{R}^D}{\mathrm{argmin}} \frac{1}{n} \sum_{i=1}^n K_h(X_i - X_{n+1}) \\ &\qquad \times \left\{ Y_i - w^\top P_h(X_i - X_{n+1}) \right\}^2 \\ &= \underset{w \in \mathbb{R}^D}{\mathrm{argmin}} \left\| \tilde{Y} - \tilde{\boldsymbol{X}} w \right\|_2^2. \end{aligned} \tag{4}$$

Then the local polynomial estimator is the first component $w_{*,1}$ of $w_*$ (Fan & Gijbels, 1996), and we define the $M$-truncated local polynomial estimator at $X_{n+1}$ as

$$f_{\mathrm{LocPol}}(\mathcal{D}_n, X_{n+1}) := (-M) \vee w_{*,1} \wedge M. \tag{5}$$

We first establish the minimax optimality of the truncated local polynomial estimator for Hölder-smooth functions.

For alternative versions of this result, see Fan & Gijbels (1996), Györfi et al. (2002), Tsybakov (2009) and Samworth & Shah (2026+).

**Theorem 2.5.** *Let $f_{\mathrm{LocPol}}$ be the $M$-truncated local polynomial estimator defined in* (5) *with degree $p := \lceil \alpha \rceil$ and bandwidth $h := n^{-1/(2\alpha+d)}$. There exists $C > 0$ depending only on $d$, $\alpha$, $M$, $c_X$, $C_X$, $c_K$, $C_K$ and $L_K$, such that*

$$R(f_{\mathrm{LocPol}}) - \sigma^2 \leq C n^{-2\alpha/(2\alpha+d)}.$$

*Moreover, we have with probability at least $1 - n^{C/(2\alpha+d)} \exp(-n^{2\alpha/(2\alpha+d)}/C)$ that*

$$C^{-1} \leq \lambda_{\min}\left(\tilde{\boldsymbol{X}}^\top \tilde{\boldsymbol{X}}\right) \leq \lambda_{\max}\left(\tilde{\boldsymbol{X}}^\top \tilde{\boldsymbol{X}}\right) \leq C. \tag{6}$$

On the high-probability event (6), $\tilde{\boldsymbol{X}}^\top \tilde{\boldsymbol{X}}$ is invertible and we have the closed form expression $w_* = \left(\tilde{\boldsymbol{X}}^\top \tilde{\boldsymbol{X}}\right)^{-1} \tilde{\boldsymbol{X}}^\top \tilde{Y}$; further, the objective function in the optimisation problem (4) is strongly convex.

## 3. Main Results

Our first main result asserts the existence of a transformer that approximates a local polynomial estimator. The central idea is to first approximate the kernel-weighted monomial basis matrix $\tilde{\boldsymbol{X}}$ and response vector $\tilde{Y}$, and then run gradient descent (using attention layers) to find an approximate solution to the least squares problem (4). We provide sketch proofs of our main results in Section 4; complete proofs are deferred to the appendices.

**Theorem 3.1.** *Let $f_{\mathrm{LocPol}}$ be the $M$-truncated local polynomial estimator defined in* (5) *with degree $p := \lceil \alpha \rceil$, kernel $K(x) := (1 - \|x\|_1)_+^2$ and bandwidth $h := n^{-1/(2\alpha+d)}$. There exists $C > 0$, depending only on $d$, $\alpha$, $M$, $c_X$ and $C_X$, such that if $D := \binom{d+p}{p}$, $d_{\mathrm{e}} := 2d + 2D + 5$, $d_{\mathrm{ffn}} := 6(D+1)(14+p)$, $L := \lceil C \log(en) \rceil$ and $B := Cn^2$, then there exists a transformer $f_{\mathrm{TF}} \in \mathcal{F}(d_{\mathrm{e}}, d_{\mathrm{ffn}}, L, B, M)$ satisfying*

$$\left| R(f_{\mathrm{TF}}) - R(f_{\mathrm{LocPol}}) \right| \leq \frac{C}{n}.$$

The bound in Theorem 3.1 could be improved to $O(1/n^c)$ for any fixed $c \geq 1$ by adjusting the constant $C$. However, $O(1/n)$ suffices for our purposes. The choice of kernel is convenient (see the discussion in Section 4), but we would expect our results to carry over, with minor modifications, to other commonly-used kernels. The next theorem shows that a transformer trained by minimising the empirical risk is minimax optimal.

**Theorem 3.2.** *Let $n, d \in \mathbb{N}$ and suppose that the data are generated according to Section 2.1. There exists $C > 0$ depending only on $d, \alpha, M, c_X$ and $C_X$, such that if $\hat{f}_\Gamma$ is*

*constructed as in Definition 2.4 with $p := \lceil \alpha \rceil$, $D := \binom{d+p}{p}$, embedding dimension $d_e := 2d + 2D + 5$, FFN width $d_{\text{ffn}} := 6(D + 1)(14 + p)$, number of transformer blocks $L := \lceil C \log(en) \rceil$, parameter bound $B := Cn^2$ and number of pretraining sequences $\Gamma \geq Cn^{2\alpha/(2\alpha+d)} \log^3(en)$, then*

$$\mathbb{E}\{R(\hat{f}_\Gamma)\} - \sigma^2 \leq Cn^{-\frac{2\alpha}{2\alpha+d}}.$$

We achieve the minimax optimal rate by taking $\Gamma = \Omega(n^{2\alpha/(2\alpha+d)} \log^3 n)$ and using $\Theta(\log n)$ transformer parameters. In contrast, Shen et al. (2025) require $\Gamma = \Omega(n^{(6\alpha+d)/(2\alpha+d)} \log n)$ and $\Theta(n)$ transformer parameters, while Kim et al. (2024) need $\Gamma = \Omega(n^{(2\alpha+2d)/(2\alpha+d)} \log n)$ and $\Theta(n^{d/(2\alpha+d)})$ parameters. These improvements are due to our new approximation result, Theorem 3.1, which shows that transformers can efficiently represent an approximate version of local polynomial estimation via gradient descent. In particular, our use of transformer blocks permits the approximation error to decrease polynomially in $n$ while the total number of parameters grows only logarithmically. This is because FFN layers can approximate polynomials exponentially fast (Lu et al., 2021), and attention layers can implement gradient descent (Bai et al., 2023; von Oswald et al., 2023), which also converges exponentially quickly. In the standard (i.e. not in-context) regression setting, existing neural network regression constructions need $\Omega(n^{d/(2\alpha+d)})$ parameters for minimax optimal estimation (Schmidt-Hieber, 2020; Lu et al., 2021). In contrast, the FFN layers in transformers permit heavy parameter sharing while the attention layers allow data points to interact with each other directly; see Figure 4. We emphasise that, in contrast to some prior work, our main result guarantees minimax optimality regardless of the smoothness level $\alpha > 0$.

Computing the empirical risk minimiser $\hat{f}_\Gamma$ involves optimising over the parameters $\boldsymbol{\theta}$ of the underlying transformer network. In general, this is a non-convex optimisation problem, so standard gradient-based approaches such as Adam (Kingma & Ba, 2015) and AdamW (Loshchilov & Hutter, 2019a) are not generally guaranteed to converge to a global minimum. However, our main result (Theorem 3.2) holds if instead $\hat{f}_\Gamma$ is taken to be any transformer achieving an empirical risk within $O(n^{-2\alpha/(2\alpha+d)})$ of the optimal value. A detailed analysis of transformer training dynamics is beyond the scope of this paper.

We remark that it is straightforward to adapt our results to certain other non-linear attention mechanisms. For instance, $\text{ReLU}(x) - \text{ReLU}(-x) = x$, so two ReLU attention heads can implement a single linear attention head. It follows that Theorems 3.1 and 3.2 hold for transformers with two ReLU attention heads in each attention layer. Moreover, it is possible to use two softmax attention layers followed by $O(\log n)$ FFN layers to approximate the output of a single

linear attention layer; see Appendix F. Thus, similar results to our Theorems 3.1 and 3.2 hold for softmax transformers with $\Theta(\log^2 n)$ layers.

## 4. Proof Sketches

The proof of our approximation result, Theorem 3.1, proceeds via several steps; see Figure 5. First, we use three transformer blocks to construct the centred and scaled covariates $(\boldsymbol{X} - 1_n X_{n+1}^\mathsf{T})/h$, along with the square root of the diagonal kernel matrix $\boldsymbol{K}_h(X_{n+1})$. Here, it is convenient for us to use the specific kernel $K(x) := (1 - \|x\|_1)_+^2$ because its square root is piecewise linear and so can be constructed exactly using ReLU FFNs. Next, we approximate the monomial basis $\boldsymbol{P}_h(X_{n+1})$, relying on the ability of ReLU FFNs to approximate polynomials with error $O(1/n^c)$ using $\Theta(\log n)$ layers (Lu et al., 2021). The monomial basis matrix and the responses are then premultiplied by the square root kernel matrix, yielding approximations of $\tilde{\boldsymbol{X}}$ and $\tilde{Y}$. Finally, the local polynomial estimator is given by the solution to the least-squares optimisation problem (4). We obtain an approximately optimal solution to this problem by implementing gradient descent using transformers (Bai et al., 2023; von Oswald et al., 2023). We only need $\Theta(\log n)$ gradient descent steps because the optimisation problem is strongly convex on the high-probability event (6). We keep track of the errors incurred, first due to having access only to approximations of $\tilde{\boldsymbol{X}}$ and $\tilde{Y}$, and second because we perform only finitely many steps of gradient descent. See Appendix B.2 for the proofs.

$$\begin{pmatrix} \boldsymbol{X} & Y \\ X_{n+1}^\mathsf{T} & . \end{pmatrix} \xrightarrow{\text{Three blocks}} \left( \frac{\boldsymbol{X} - 1_n X_{n+1}^\mathsf{T}}{h} \quad \sqrt{\frac{\boldsymbol{K}_h(X_{n+1})}{n}} \right)$$

$$\xrightarrow{\Theta(\log n) \text{ blocks}} \left( \sqrt{\frac{\boldsymbol{K}_h(X_{n+1})}{n}} \boldsymbol{P}_h(X_{n+1}) \quad \sqrt{\frac{\boldsymbol{K}_h(X_{n+1})}{n}} Y \right)$$

$$\xrightarrow{\Theta(\log n) \text{ blocks}} w_*$$

*Figure 5.* Key steps in the construction of the approximating transformer $f_{\text{TF}}$; unchanged quantities are omitted for clarity. The first three blocks compute the centred and scaled covariates $\boldsymbol{X}$, relative to the test point $X_{n+1}$, along with the kernel matrix $\boldsymbol{K}_h(X_{n+1})$. Next, $\Theta(\log n)$ blocks are used to repeatedly multiply these, producing approximations of the kernel-weighted monomial basis $\tilde{\boldsymbol{X}}$ and responses $\tilde{Y}$. Finally, $\Theta(\log n)$ steps of gradient descent are applied to arrive at an approximation of the optimal point $w_*$.

Theorem 3.2 is proved using a decomposition of the population risk of the empirical risk minimising transformer $\hat{f}_\Gamma$; see Figure 6. Since the transformer functions in $\mathcal{F}$ have at most $O(\log n)$ parameters, each bounded in magnitude by $O(n^2)$, and as the transformer output is Lipschitz in the parameters, we are able to show (Lemma C.4) that the

covering numbers of this class satisfy

$$\log N\big(\mathcal{F}, \delta, \|\cdot\|_\infty\big) = O\bigg\{\log^3 n + (\log n)\log\bigg(\frac{1}{\delta}\bigg)\bigg\},$$

where $\delta > 0$. Applying a standard empirical process theory result for $L_2$-empirical risk minimisers (Györfi et al., 2002) yields the following expected excess risk bound:

$$\mathbb{E}\{R(\hat{f}_\Gamma)\} - \sigma^2 \le 2\big\{R(f_{\mathrm{TF}}) - R(f_{\mathrm{LocPol}})\big\}$$

$$+ 2\big\{R(f_{\mathrm{LocPol}}) - \sigma^2\big\} + O\bigg(\frac{\log^3 n + (\log n)\log\Gamma}{\Gamma}\bigg).$$

The remaining terms are bounded by applying Theorem 3.1 and Theorem 2.5.

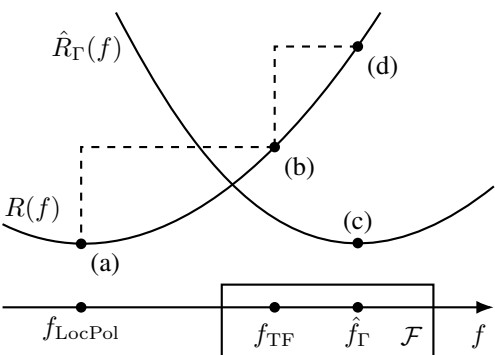

*Figure 6.* Overview of the proof of Theorem 3.2. (a) The truncated local polynomial estimator attains the minimax optimal population risk of $R(f_{\mathrm{LocPol}}) - \sigma^2 = O\big(n^{-2\alpha/(2\alpha+d)}\big)$, by Theorem 2.5. (b) We construct a specific transformer with excess risk $R(f_{\mathrm{TF}}) - R(f_{\mathrm{LocPol}}) = O(1/n)$; see Theorem 3.1. (c) Our estimator $\hat{f}_\Gamma$ minimises the empirical risk $\hat{R}_\Gamma$ over the transformer class $\mathcal{F}$ (Definition 2.4). (d) By bounding the covering numbers of $\mathcal{F}$, we show that $\mathbb{E}\{R(\hat{f}_\Gamma)\} - 2R(f_{\mathrm{TF}}) + \sigma^2 = O\big((\log^3 n + (\log n)\log\Gamma)/\Gamma\big)$; see Appendix C.

## 5. Simulations

In this section, we compare the performance of a pretrained transformer with local polynomial estimators. To this end, we consider random regression tasks (drawn using a random Fourier series) with $d = 3$, $\alpha = 3$, $\sigma = 0.01$ and $n \in \{15, 20, 25, 30, 35\}$. We use a linear attention transformer with embedding dimension $d_e = 256$, FFN width $d_{\mathrm{ffn}} = 1024$ and $L = 12$ transformer blocks. The transformer was trained for $50,000$ optimisation steps using AdamW (Loshchilov & Hutter, 2019b) with weight decay $10^{-3}$ and a cosine annealing learning rate schedule. At each optimisation step, we sampled 40 random regression functions, and randomly generated 16 pretraining sequences for each function (i.e. the covariate vectors and queries were random while the regression function was held fixed), yielding a total of 640 pretraining sequences per optimisation step. The pretraining was carried out on an NVIDIA

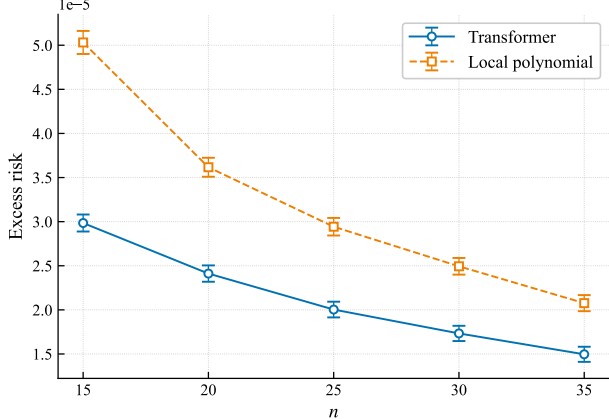

*Figure 7.* Estimated excess risks for a pretrained transformer and local polynomial estimator. Error bars represent 90% confidence intervals for the excess population risks based on $10^5$ test sequences.

A100 GPU (80GB) over approximately 2.25 hours. At test time, for each $n \in \{15, 20, 25, 30, 35\}$, we generated $10^5$ random regression tasks, each with $n$ in-context examples, and compared the performance of our pretrained transformer with a local cubic estimator with bandwidth $h \in \{0.30, 0.40, 0.50, 0.60, 0.70, 0.80, 1.00, 1.25, 1.50, 2.00\}$ and ridge penalty[1] in $\{0.001, 0.005, 0.01, 0.05, 0.1, 0.5\}$ chosen via 5-fold cross validation. Figure 7 shows the estimated excess risks for the pretrained transformer and the local polynomial estimator. We see that the transformer and the local polynomial estimator exhibit a similar dependence on $n$, while the transformer outperforms the local polynomial estimator. The code for our simulations is available at https://github.com/tianyima2000/ICL_LocPol.

## 6. Summary

We have presented a theoretical study of in-context learning for nonparametric regression problems. With $n$ in-context examples, $d$-dimensional covariates and $\alpha$-Hölder smooth regression functions, we have shown that suitable pretrained transformers with only $\Theta(\log n)$ parameters are able to attain the minimax optimal rate $O\big(n^{-2\alpha/(2\alpha+d)}\big)$. Moreover, we showed that this rate is achievable whenever $\Gamma = \Omega(n^{2\alpha/(2\alpha+d)}\log^3 n)$ pretraining sequences are available. Our approach involved first demonstrating that transformers are able to approximate local polynomial estimators to arbitrary accuracy, and then bounding the risk of our transformer estimator by deriving covering number bounds and applying results from empirical process theory.

---

[1]A local cubic estimator uses a monomial basis with dimension $\binom{6}{3} = 20$. Thus, in our simulations, we added a ridge penalty to (4) to improve its empirical performance.

Future directions could include extensions to next token prediction in dependent data settings; this would allow for more accurate modelling of large language models. Alternatively, one might attempt to discover and exploit low-dimensional structure (such as sparsity, an index structure or a manifold hypothesis) in the regression function, thereby avoiding the curse of dimensionality. Finally, one could aim to establish minimax theory for transformers trained via gradient descent (or related optimisation algorithms).

## Acknowledgement

The first three authors were funded by Summer Research in Mathematics bursaries from the University of Cambridge. The last three authors were funded by RJS's European Research Council Advanced Grant 101019498.

## Impact Statement

This paper presents work whose goal is to advance the field of Machine Learning. There are many potential societal consequences of our work, none which we feel must be specifically highlighted here.

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

# A. Local Polynomial Estimation

We consider the classical degree-$p$ local polynomial estimator for $d$-dimensional covariates (Fan & Gijbels, 1996; Györfi et al., 2002; Härdle et al., 2004; Tsybakov, 2009), truncating the output to avoid overly poor estimation on low-probability events. Our main contribution in this section is to provide high-probability bounds with sub-exponential tails for the integrated squared error of this estimator (Theorem A.6). As an intermediate result, we also provide a high-probability uniform guarantee on the spectra of the associated kernel-weighted monomial basis matrices in Lemma A.3, which is also essential for showing that transformers can successfully approximate local polynomial methods. We remark that truncation is important for passing from high-probability guarantees to bounds on the expected squared error: see, for example, Györfi et al. (2002, Problem 20.4).

## A.1. Local Polynomial Estimation Set-up

Let $P_{X,\varepsilon}$ be as defined in Section 2.1 and let $m \in \mathcal{H}(d, \alpha, M)$ be fixed, where $\alpha, M > 0$. Let $(X_1, \varepsilon_1), \ldots, (X_n, \varepsilon_n) \overset{\text{i.i.d.}}{\sim} P_{X,\varepsilon}$ and define

$$Y_i := m(X_i) + \varepsilon_i$$

for $i \in [n]$. Suppose that $K : \mathbb{R}^d \to [0, C_K]$ is a Borel measurable function supported on $[-1, 1]^d$ satisfying $|K(x) - K(x')| \leq L_K \|x - x'\|_\infty$ for all $x, x' \in \mathbb{R}^d$. Assume that $K(x) \geq c_K$ for each $x \in [-c_K, c_K]^d$, where $c_K \in (0, 1]$. Let $h > 0$ and define $K_h(\cdot) := K(\cdot/h)/h^d$. For $p \in \mathbb{N}_0$ with $p \geq \underline{\alpha}$, write $D := \binom{d+p}{p}$ and let $P_h : \mathbb{R}^d \to \mathbb{R}^D$ be defined by $P_h(x) := \left(x^\nu/(\nu! h^{|\nu|}) : \nu \in \mathbb{N}_0^d, 0 \leq |\nu| \leq p\right)$, with components ordered in increasing lexicographic ordering in $\nu$, so that $P_h(x)^\mathsf{T} e_1 = 1$, where $e_1$ denotes the first standard basis vector in $\mathbb{R}^D$. We may index elements of $\mathbb{R}^D$ using either regular indices $j \in [D]$ or multi-indices $\nu \in \mathbb{N}_0^d$, as convenient. For $x \in [0, 1]^d$, define the random diagonal matrix $\boldsymbol{K}_h(x) \in \mathbb{R}^{n \times n}$ with $i$th diagonal entry given by $\boldsymbol{K}_h(x)_{ii} := K_h(X_i - x)$ for $i \in [n]$. Let $\boldsymbol{P}_h(x) \in \mathbb{R}^{n \times D}$ be defined by $\boldsymbol{P}_h(x)_{ij} := P_h(X_i - x)_j$ for $i \in [n]$ and $j \in [D]$. Define $\hat{\boldsymbol{H}}(x) := \boldsymbol{P}_h(x)^\mathsf{T} \boldsymbol{K}_h(x) \boldsymbol{P}_h(x)/n \in \mathbb{R}^{D \times D}$ and $\boldsymbol{H}(x) := \mathbb{E}\{\hat{\boldsymbol{H}}(x)\} = \mathbb{E}\{P_h(X_1 - x) K_h(X_1 - x) P_h(X_1 - x)^\mathsf{T}\}$. With $Y := (Y_1, \ldots, Y_n)^\mathsf{T}$, define the local polynomial estimator of degree $p$ at $x$ by $\hat{m}_n(x) := e_1^\mathsf{T} \hat{\boldsymbol{H}}(x)^{-1} \boldsymbol{P}_h(x)^\mathsf{T} \boldsymbol{K}_h(x) Y/n$ and let $\tilde{m}_n(x) := (-M) \vee \hat{m}_n(x) \wedge M$. Write $m(\boldsymbol{X}) := \left(m(X_1), \ldots, m(X_n)\right)^\mathsf{T}$ and $\hat{B}_n(x) := e_1^\mathsf{T} \hat{\boldsymbol{H}}(x)^{-1} \boldsymbol{P}_h(x)^\mathsf{T} \boldsymbol{K}_h(x) m(\boldsymbol{X})/n \in \mathbb{R}$.

## A.2. Results for Local Polynomial Estimation

**Lemma A.1.** *Under the set-up of Section A.1, there exists $C \geq 1$, depending only on $c_K, C_K, c_X, C_X, d$ and $p$, such that for $h \leq 1/C$, we have $\sup_{x \in [0,1]^d} \|\boldsymbol{H}(x)\|_{\text{op}} \leq C$ and moreover, $\boldsymbol{H}(x)$ is invertible for every $x \in [0, 1]^d$, with $\sup_{x \in [0,1]^d} \|\boldsymbol{H}(x)^{-1}\|_{\text{op}} \leq C$.*

*Proof of Lemma A.1.* Throughout the proof, $C_1, C_2, \ldots > 0$ denote quantities depending only on $c_K, C_K, c_X, C_X, d$ and $p$. For $z \in [-1, 1]^d$, each entry of $P_h(z) K_h(z) P_h(z)^\mathsf{T}$ is bounded in magnitude by $K_h(z)$, as $K_h(z) = 0$ unless $\|z\|_\infty \leq h$. Thus, for each $x \in [0, 1]^d$,

$$\begin{aligned}
\|\boldsymbol{H}(x)\|_{\text{op}} &= \left\| \int_{[0,1]^d} P_h(y - x) K_h(y - x) P_h(y - x)^\mathsf{T} f_X(y) \, \mathrm{d}y \right\|_{\text{op}} \\
&\leq DC_X \int_{[0,1]^d} \left\| P_h(y - x) K_h(y - x) P_h(y - x)^\mathsf{T} \right\|_{\max} \mathrm{d}y \\
&\leq DC_X \int_{[0,1]^d} K_h(y - x) \, \mathrm{d}y \leq DC_X \int_{\mathbb{R}^d} K(z) \, \mathrm{d}z \leq 2^d C_K D C_X =: C_1.
\end{aligned}$$

For the second result, take $v \in \mathbb{R}^D$ with $\|v\|_2 = 1$. Since $K(x) \geq c_K$ for each $x \in [-c_K, c_K]^d$, and as either $-u/h \leq -c_K$

or $(1-u)/h \geq c_K$ for $h \leq 1/(2c_K)$ and $u \in [-c_K, c_K]$, we have for $x = (x_1, \ldots, x_d)^\mathsf{T} \in [0,1]^d$ that

$$
\begin{aligned}
v^\mathsf{T} \boldsymbol{H}(x) v &= \int_{[0,1]^d} K_h(y-x) \{v^\mathsf{T} P_h(y-x)\}^2 f_X(y) \, \mathrm{d}y \\
&\geq c_X \int_{\prod_{j=1}^d [-x_j/h, (1-x_j)/h]} K(z) \{v^\mathsf{T} P_1(z)\}^2 \, \mathrm{d}z \\
&\geq c_X c_K \min_{(\zeta_1, \ldots, \zeta_d) \in \{-1,1\}^d} \int_{\prod_{j=1}^d (\zeta_j [0, c_K])} \{v^\mathsf{T} P_1(z)\}^2 \, \mathrm{d}z,
\end{aligned}
$$

where $(-1) \cdot [0, c_K] := [-c_K, 0]$. The integrand above is a non-negative polynomial in $z$ that is not identically zero, so there are finitely many points $z \in \mathbb{R}^d$ satisfying $v^\mathsf{T} P_1(z) = 0$. As the Lebesgue integral of an almost everywhere positive continuous function over a positive measure set is positive, we deduce that there exists $c(v, d, p, c_K) > 0$ such that $v^\mathsf{T} \boldsymbol{H}(x) v \geq c_X c(v, d, p, c_K)$. Therefore, the map $v \mapsto v^\mathsf{T} \boldsymbol{H}(x) v$ defined for $v \in \mathbb{R}^D$ with $\|v\|_2 = 1$ is a continuous function on a compact set that is positive everywhere, so in particular it is bounded away from zero. We conclude that

$$
\inf_{v \in \mathbb{R}^D : \|v\|_2 = 1} v^\mathsf{T} \boldsymbol{H}(x) v \geq \frac{1}{C_2},
$$

as required. $\qquad\square$

**Lemma A.2.** *Let $d \in \mathbb{N}$ and $\nu \in \mathbb{N}_0^d$. For each $x, y \in \mathbb{R}^d$,*

$$
|x^\nu - y^\nu| \leq |\nu| (\|x\|_\infty \vee \|y\|_\infty)^{|\nu|-1} \|x - y\|_\infty.
$$

*Proof of Lemma A.2.* Write $x = (x_1, \ldots, x_d)^\mathsf{T}$ and $y = (y_1, \ldots, y_d)^\mathsf{T}$. By telescoping the sum and the mean value theorem,

$$
\begin{aligned}
|x^\nu - y^\nu| &= \left| \prod_{j=1}^d x_j^{\nu_j} - \prod_{j=1}^d y_j^{\nu_j} \right| = \left| \sum_{r=1}^d \left\{ \left( \prod_{j=r+1}^d x_j^{\nu_j} \right) \left( \prod_{\ell=1}^{r-1} y_\ell^{\nu_\ell} \right) (x_r^{\nu_r} - y_r^{\nu_r}) \right\} \right| \\
&\leq \sum_{r=1}^d \left( \prod_{j=r+1}^d |x_j|^{\nu_j} \right) \left( \prod_{\ell=1}^{r-1} |y_\ell|^{\nu_\ell} \right) |x_r^{\nu_r} - y_r^{\nu_r}| \\
&\leq \sum_{r=1}^d \left( \prod_{j=r+1}^d |x_j|^{\nu_j} \right) \left( \prod_{\ell=1}^{r-1} |y_\ell|^{\nu_\ell} \right) \nu_r (|x_r| \vee |y_r|)^{\nu_r - 1} |x_r - y_r| \\
&\leq \sum_{r=1}^d \nu_r \|x - y\|_\infty (\|x\|_\infty \vee \|y\|_\infty)^{|\nu|-1} = |\nu| (\|x\|_\infty \vee \|y\|_\infty)^{|\nu|-1} \|x - y\|_\infty,
\end{aligned}
$$

as required. $\qquad\square$

**Lemma A.3.** *Assume the set-up of Section A.1. There exists $C \geq 1$, depending only on $c_K$, $C_K$, $c_X$, $C_X$, $L_K$, $d$ and $p$, such that, for $h \leq 1/C$, we have with probability at least $1 - \exp(-nh^d/C)/h^C$ that $\sup_{x \in [0,1]^d} \|\hat{\boldsymbol{H}}(x)\|_{\mathrm{op}} \leq C$ and moreover, $\hat{\boldsymbol{H}}(x)$ is invertible for every $x \in [0,1]^d$, satisfying $\sup_{x \in [0,1]^d} \|\hat{\boldsymbol{H}}(x)^{-1}\|_{\mathrm{op}} \leq C$.*

*Proof of Lemma A.3.* Throughout the proof, $C_1, C_2, \ldots > 0$ denote quantities depending only on $c_K$, $C_K$, $c_X$, $C_X$, $L_K$, $d$ and $p$. For $i \in [n]$ and $j, k \in [D]$, write

$$
u_{ijk}(x) := P_h(X_i - x)_j K_h(X_i - x) P_h(X_i - x)_k - \mathbb{E}\{P_h(X_i - x)_j K_h(X_i - x) P_h(X_i - x)_k\},
$$

so that $\hat{\boldsymbol{H}}(x)_{jk} - \boldsymbol{H}(x)_{jk} = \sum_{i=1}^n u_{ijk}/n$. Further, $\mathbb{E}(u_{ijk}) = 0$ and $|u_{ijk}| \leq 2C_K/h^d$. Also,

$$
\begin{aligned}
\mathbb{E}(u_{ijk}^2) &\leq \mathbb{E}\{P_h(X_i - x)_j^2 K_h(X_i - x)^2 P_h(X_i - x)_k^2\} \leq \mathbb{E}\{K_h(X_i - x)^2\} \\
&= \int_{[0,1]^d} K_h(y - x)^2 f_X(y) \, \mathrm{d}y \leq \frac{C_X}{h^d} \int_{\mathbb{R}^d} K(z)^2 \, \mathrm{d}z \leq \frac{2^d C_X C_K^2}{h^d}.
\end{aligned}
$$

Now, by Bernstein's inequality, for all $t > 0$,

$$\mathbb{P}\left(\left|\frac{1}{n}\sum_{i=1}^{n} u_{ijk}\right| > \frac{C_1 t}{\sqrt{nh^d}} + \frac{C_1 t^2}{nh^d}\right) \leq 2e^{-t^2/2}.$$

Therefore, by a union bound and as $\|A\|_{\mathrm{op}} \leq D\|A\|_{\max}$ for $A \in \mathbb{R}^{D \times D}$,

$$\mathbb{P}\left(\left\|\hat{\boldsymbol{H}}(x) - \boldsymbol{H}(x)\right\|_{\mathrm{op}} > \frac{C_1 Dt}{\sqrt{nh^d}} + \frac{C_1 Dt^2}{nh^d}\right) \leq D(D+1)e^{-t^2/2}.$$

Therefore, by Weyl's inequality (e.g. Bhatia, 1997, Corollary III.2.6) and Lemma A.1, with probability at least $1 - C_2 e^{-t^2/2}$, where $C_2 \geq 1$,

$$\lambda_{\min}\left(\hat{\boldsymbol{H}}(x)\right) \geq \lambda_{\min}\left(\boldsymbol{H}(x)\right) - \left\|\hat{\boldsymbol{H}}(x) - \boldsymbol{H}(x)\right\|_{\mathrm{op}} \geq \frac{1}{C_2} - \frac{C_2 Dt}{\sqrt{nh^d}} - \frac{C_2 Dt^2}{nh^d},$$

$$\lambda_{\max}\left(\hat{\boldsymbol{H}}(x)\right) \leq \lambda_{\max}\left(\boldsymbol{H}(x)\right) + \left\|\hat{\boldsymbol{H}}(x) - \boldsymbol{H}(x)\right\|_{\mathrm{op}} \leq C_2 + \frac{C_2 Dt}{\sqrt{nh^d}} + \frac{C_2 Dt^2}{nh^d}.$$

Setting $t := \sqrt{nh^d}/(4C_2^2 D)$, we see that with probability at least $1 - C_3 \exp(-nh^d/C_3)$,

$$\lambda_{\min}\left(\hat{\boldsymbol{H}}(x)\right) \geq \frac{1}{C_2} - \frac{1}{4C_2} - \frac{1}{16C_2^3 D} \geq \frac{1}{C_3},$$

$$\lambda_{\max}\left(\hat{\boldsymbol{H}}(x)\right) \leq C_2 + \frac{1}{4C_2} + \frac{1}{16C_2^3 D} \leq C_3.$$

Observe that $K_h(\cdot)$ is bounded by $C_K/h^d$ and is $L_K/h^{d+1}$-Lipschitz on $\mathbb{R}^d$ with respect to $\|\cdot\|_\infty$. Further, for $j \in [D]$, the restriction of $P_h(\cdot)_j$ to $[-h, h]^d$ is bounded by 1 and is $p/h$-Lipschitz with respect to $\|\cdot\|_\infty$, by Lemma A.2. Thus, for $x, y, z \in [0,1]^d$ and $j, k \in [D]$,

$$\left|P_h(z-y)_j K_h(z-y)P_h(z-y)_k - P_h(z-x)_j K_h(z-x)P_h(z-x)_k\right| \leq \frac{L_K + 2pC_K}{h^{d+1}}\|y-x\|_\infty \leq \frac{C_4}{h^{d+1}}\|y-x\|_\infty.$$

We deduce that $x \mapsto \hat{\boldsymbol{H}}(x)$ is $C_4/h^{d+1}$-Lipschitz from $\|\cdot\|_\infty$ to $\|\cdot\|_{\max}$, where we may take $C_4 \geq C_3 \vee 1$, so it is also $C_4 D/h^{d+1}$-Lipschitz from $\|\cdot\|_\infty$ to $\|\cdot\|_{\mathrm{op}}$. It follows by Weyl's inequality that both $x \mapsto \lambda_{\min}\left(\hat{\boldsymbol{H}}(x)\right)$ and $x \mapsto \lambda_{\max}\left(\hat{\boldsymbol{H}}(x)\right)$ are $C_4 D/h^{d+1}$-Lipschitz with respect to $\|\cdot\|_\infty$. Let $\delta := h^{d+1}/(2C_4^2 D) \leq 1$ and let $\mathcal{X}_h$ denote a $\delta$-cover of $[0,1]^d$ with respect to $\|\cdot\|_\infty$ of cardinality at most $(2/\delta)^d$. By a union bound, with probability at least $1 - C_3(2/\delta)^d \exp(-nh^d/C_3)$,

$$\inf_{x \in [0,1]^d} \lambda_{\min}\left(\hat{\boldsymbol{H}}(x)\right) \geq \min_{x \in \mathcal{X}_h} \lambda_{\min}\left(\hat{\boldsymbol{H}}(x)\right) - \frac{1}{2C_4} \geq \frac{1}{C_3} - \frac{1}{2C_4} \geq \frac{1}{2C_4}.$$

Applying the same logic to the maximum eigenvalue, we see that both $\inf_{x \in [0,1]^d} \lambda_{\min}\left(\hat{\boldsymbol{H}}(x)\right) \geq 1/C_5$ and $\sup_{x \in [0,1]^d} \lambda_{\max}\left(\hat{\boldsymbol{H}}(x)\right) \leq C_5$, with probability at least $1 - \exp(-nh^d/C_5)/h^{C_5}$ for $h \leq 1/C_5$. $\qquad\square$

**Lemma A.4.** *Assume the set-up of Section A.1. There exists $C > 0$, depending only on $c_K$, $C_K$, $c_X$, $C_X$, $L_K$, $d$ and $p$, such that for all $h \leq 1/C$ and $t > 0$, we have with probability at least $1 - C\exp(-t^2) - \exp(-nh^d/C)/h^C$ that*

$$\int_{[0,1]^d} \left(\hat{m}_n(x) - \hat{B}_n(x)\right)^2 dx \leq \frac{Ct^2}{nh^d}.$$

*Proof of Lemma A.4.* Throughout the proof, $C_1, C_2, \ldots > 0$ denote quantities depending only on $c_K$, $C_K$, $c_X$, $C_X$, $L_K$, $d$ and $p$. With $\varepsilon := (\varepsilon_1, \ldots, \varepsilon_n)^\mathsf{T}$, we have

$$\int_{[0,1]^d} \left(\hat{m}_n(x) - \hat{B}_n(x)\right)^2 dx = \frac{1}{n^2} \int_{[0,1]^d} \left(e_1^\mathsf{T} \hat{\boldsymbol{H}}(x)^{-1} \boldsymbol{P}_h(x)^\mathsf{T} \boldsymbol{K}_h(x)\varepsilon\right)^2 dx$$

$$\leq \frac{1}{n^2} \sup_{x \in [0,1]^d} \left\|\hat{\boldsymbol{H}}(x)^{-1}\right\|_{\mathrm{op}}^2 \int_{[0,1]^d} \left\|\boldsymbol{P}_h(x)^\mathsf{T} \boldsymbol{K}_h(x)\varepsilon\right\|_2^2 dx.$$

For $i, j \in [n]$, define the random variables

$$U_{ij} := \sum_{k=1}^{D} \int_{[0,1]^d} P_h(X_i - x)_k P_h(X_j - x)_k K_h(X_i - x) K_h(X_j - x) \varepsilon_i \varepsilon_j \, dx,$$

so that, for each $i, j \in [n]$, we have $\mathbb{E}(U_{ij} \mid X_i, \varepsilon_i) = 0$ and

$$\int_{[0,1]^d} \left\| \boldsymbol{P}_h(x)^\mathsf{T} \boldsymbol{K}_h(x) \varepsilon \right\|_2^2 \, dx = \int_{[0,1]^d} \left\| \sum_{i=1}^{n} P_h(X_i - x) K_h(X_i - x) \varepsilon_i \right\|_2^2 \, dx = \sum_{i=1}^{n} \sum_{j=1}^{n} U_{ij}.$$

By the Cauchy–Schwarz inequality, for $i, j \in [n]$, and since $|\varepsilon_i| \leq 1$,

$$|U_{ij}| \leq \max_{r \in [n]} \sum_{k=1}^{D} \int_{[0,1]^d} P_h(X_r - x)_k^2 K_h(X_r - x)^2 \, dx \leq D \sup_{y \in [0,1]^d} \int_{[0,1]^d} K_h(y - x)^2 \, dx$$

$$= \frac{D}{h^{2d}} \sup_{y \in [0,1]^d} \int_{[0,1]^d} K\left( \frac{y - x}{h} \right)^2 \, dx \leq \frac{D}{h^d} \int_{\mathbb{R}^d} K(u)^2 \, du \leq \frac{C_K^2 2^d D}{h^d} \leq \frac{C_2}{h^d}.$$

In particular, $\sum_{i=1}^{n} U_{ii} \leq C_2 n / h^d$. Moreover, by Hoeffding's inequality for degenerate second-order $U$-statistics (e.g., de la Peña & Giné, 1999, Theorem 4.1.12b), for all $t > 0$,

$$\mathbb{P}\left( \left| \sum_{i=1}^{n} \sum_{j \in [n] \setminus \{i\}} U_{ij} \right| > \frac{C_3 n t^2}{h^d} \right) \leq C_3 e^{-t^2}.$$

Therefore, with probability at least $1 - C_3 e^{-t^2}$, where $C_3 \geq e$,

$$\int_{[0,1]^d} \left\| \boldsymbol{P}_h(x)^\mathsf{T} \boldsymbol{K}_h(x) \varepsilon \right\|_2^2 \, dx \leq \frac{C_2 n}{h^d} + \frac{C_3 n t^2}{h^d} \leq \frac{C_4 n t^2}{h^d},$$

as there is nothing to prove if $t \in [0, 1]$. By Lemma A.3 and a union bound, we conclude that with probability at least $1 - C_5 \exp(-t^2) - \exp(-nh^d / C_5) / h^{C_5}$, we have

$$\int_{[0,1]^d} \left( \hat{m}_n(x) - \hat{B}_n(x) \right)^2 \, dx \leq \frac{C_5 t^2}{n h^d}. \qquad \square$$

**Lemma A.5.** *Assume the set-up of Section A.1. There exists $C \geq 1$, depending only on $c_K$, $C_K$, $c_X$, $C_X$, $L_K$, $d$, $p$ and $M$ such that for $h \leq 1/C$, we have with probability at least $1 - \exp(-nh^d / C) / h^C$ that*

$$\int_{[0,1]^d} \left( \hat{B}_n(x) - m(x) \right)^2 \, dx \leq C h^{2\alpha}.$$

*Proof of Lemma A.5.* Throughout the proof, $C_1, C_2, \ldots > 0$ denote quantities depending only on $c_K$, $C_K$, $c_X$, $C_X$, $L_K$, $d$, $p$ and $M$. By Taylor's theorem, as $m \in \mathcal{H}(d, \alpha, M)$, for all $x, x' \in [0, 1]^d$, there exists $\tilde{x}$ on the line segment between $x$ and $x'$ such that

$$m(x') - m(x) = \sum_{\nu \in \mathbb{N}_0^d : |\nu| \in [\underline{\alpha}]} \frac{(x' - x)^\nu}{\nu!} \partial_\nu m(x) + \sum_{\nu \in \mathbb{N}_0^d : |\nu| = \underline{\alpha}} \frac{(x' - x)^\nu}{\nu!} \left\{ \partial_\nu m(\tilde{x}) - \partial_\nu m(x) \right\}. \qquad (7)$$

Since $p \geq \underline{\alpha}$, the first term on the right-hand side of (7) satisfies

$$\sum_{\nu \in \mathbb{N}_0^d : |\nu| \in [\underline{\alpha}]} \frac{(x' - x)^\nu}{\nu!} \partial_\nu m(x) = \sum_{\nu \in \mathbb{N}_0^d : |\nu| \in [\underline{\alpha}]} h^{|\nu|} P_h(x' - x)^\mathsf{T} e_\nu \partial_\nu m(x). \qquad (8)$$

For the second term on the right-hand side of (7), for $\|x' - x\|_\infty \le h$, we have

$$\left| \sum_{\nu \in \mathbb{N}_0^d : |\nu| = \underline{\alpha}} \frac{(x' - x)^\nu}{\nu!} \{\partial_\nu m(\tilde{x}) - \partial_\nu m(x)\} \right| \le \sum_{\nu \in \mathbb{N}_0^d : |\nu| = \underline{\alpha}} h^{\underline{\alpha}} M \|\tilde{x} - x\|_2^{\alpha - \underline{\alpha}} \le C_1 h^\alpha. \tag{9}$$

Note that $1_n := (1, \ldots, 1)^\mathsf{T} = \boldsymbol{P}_h(x)e_1 \in \mathbb{R}^n$, so by (7), (8) and (9),

$$
\begin{aligned}
\left| \hat{B}_n(x) - m(x) \right| &= \left| e_1^\mathsf{T} \hat{\boldsymbol{H}}(x)^{-1} \boldsymbol{P}_h(x)^\mathsf{T} \boldsymbol{K}_h(x) m(\boldsymbol{X})/n - m(x) \right| \\
&= \frac{1}{n} \left| e_1^\mathsf{T} \hat{\boldsymbol{H}}(x)^{-1} \boldsymbol{P}_h(x)^\mathsf{T} \boldsymbol{K}_h(x) \{m(\boldsymbol{X}) - m(x) 1_n\} \right| \\
&= \frac{1}{n} \left| e_1^\mathsf{T} \hat{\boldsymbol{H}}(x)^{-1} \sum_{i=1}^n P_h(X_i - x) K_h(X_i - x) \{m(X_i) - m(x)\} \right| \\
&\le \sum_{\nu \in \mathbb{N}_0^d : |\nu| \in [\underline{\alpha}]} \left| h^{|\nu|} e_1^\mathsf{T} e_\nu \partial_\nu m(x) \right| + \frac{C_1 h^\alpha}{n} \left\| \hat{\boldsymbol{H}}(x)^{-1} \right\|_{\mathrm{op}} \left\| \sum_{i=1}^n \|P_h(X_i - x)\|_\infty 1_D K_h(X_i - x) \right\|_2 \\
&\le \frac{C_1 \sqrt{D} h^\alpha}{n} \left\| \hat{\boldsymbol{H}}(x)^{-1} \right\|_{\mathrm{op}} \sum_{i=1}^n K_h(X_i - x).
\end{aligned}
$$

Therefore,

$$\int_{[0,1]^d} \left( \hat{B}_n(x) - m(x) \right)^2 \mathrm{d}x \le \frac{C_1^2 D h^{2\alpha}}{n^2} \sup_{x \in [0,1]^d} \left\| \hat{\boldsymbol{H}}(x)^{-1} \right\|_{\mathrm{op}}^2 \int_{[0,1]^d} \left( \sum_{i=1}^n K_h(X_i - x) \right)^2 \mathrm{d}x.$$

For $i \in [n]$ and $x \in [0,1]^d$, let $u(x) := \mathbb{E}\{K_h(X_1 - x)\}$ and $U_i(x) := K_h(X_i - x) - u(x)$, so

$$
\begin{aligned}
\int_{[0,1]^d} \left( \sum_{i=1}^n K_h(X_i - x) \right)^2 \mathrm{d}x &= \sum_{i=1}^n \sum_{j=1}^n \int_{[0,1]^d} K_h(X_i - x) K_h(X_j - x) \mathrm{d}x \\
&= \sum_{i=1}^n \sum_{j=1}^n \int_{[0,1]^d} \{U_i(x) + u(x)\} \{U_j(x) + u(x)\} \mathrm{d}x \\
&= \sum_{i=1}^n \sum_{j=1}^n \int_{[0,1]^d} U_i(x) U_j(x) \mathrm{d}x + 2n \sum_{i=1}^n \int_{[0,1]^d} u(x) U_i(x) \mathrm{d}x + n^2 \int_{[0,1]^d} u(x)^2 \mathrm{d}x. \tag{10}
\end{aligned}
$$

Now, $0 \le u(x) \le 2^d C_K C_X$, so

$$\int_{[0,1]^d} u(x)^2 \mathrm{d}x \le 2^{2d} C_K^2 C_X^2 \le C_2.$$

Next, for $i \in [n]$,

$$\left| \int_{[0,1]^d} u(x) U_i(x) \mathrm{d}x \right| \le 2^d C_K C_X \int_{[0,1]^d} \left| K_h(X_i - x) - u(x) \right| \mathrm{d}x \le 2^{2d} C_K^2 C_X + 2^{2d} C_K^2 C_X^2 \le C_3.$$

Therefore, by Hoeffding's inequality, for each $t > 0$, we have with probability at least $1 - e^{-t^2}$ that

$$\sum_{i=1}^n \int_{[0,1]^d} u(x) U_i(x) \mathrm{d}x \le C_3 t \sqrt{2n}.$$

Since $|U_i(x)| \le C_K h^{-d}$ for each $i \in [n]$, we have for $i, j \in [n]$ that

$$\left| \int_{[0,1]^d} U_i(x) U_j(x) \mathrm{d}x \right| \le C_K h^{-d} \int_{[0,1]^d} \left| K_h(X_i - x) - u(x) \right| \mathrm{d}x \le C_4 h^{-d}.$$

In particular, $\sum_{i=1}^{n} \int_{[0,1]^d} U_i(x)^2 \, \mathrm{d}x \leq C_4 n h^{-d}$. Further, by Hoeffding's inequality for degenerate second-order $U$-statistics (e.g., de la Peña & Giné, 1999, Theorem 4.1.12b), for all $t > 0$, with probability at least $1 - C_5 e^{-t^2}$,

$$\sum_{i=1}^{n} \sum_{j=1}^{n} \int_{[0,1]^d} U_i(x) U_j(x) \, \mathrm{d}x \leq C_5 n h^{-d} t^2.$$

Combining the bounds on terms in (10), we deduce that, with probability at least $1 - C_6 e^{-t^2}$,

$$\int_{[0,1]^d} \left( \sum_{i=1}^{n} K_h(X_i - x) \right)^2 \mathrm{d}x \leq C_6 n h^{-d} t^2 + C_6 t n^{3/2} + C_6 n^2.$$

Taking $t := \sqrt{n h^d}$ and since $h \leq 1$, we obtain that with probability at least $1 - C_6 \exp(-n h^d)$,

$$\int_{[0,1]^d} \left( \sum_{i=1}^{n} K_h(X_i - x) \right)^2 \mathrm{d}x \leq 3 C_6 n^2.$$

Therefore, by Lemma A.3, with probability at least $1 - \exp(-n h^d / C_7)/h^{C_7}$,

$$\int_{[0,1]^d} \left( \hat{B}_n(x) - m(x) \right)^2 \mathrm{d}x \leq C_7 h^{2\alpha},$$

as required. $\qquad \square$

**Theorem A.6.** *Assume the set-up of Section A.1. There exists $C \geq 1$ depending only on $c_K$, $C_K$, $c_X$, $C_X$, $L_K$, $d$, $p$ and $M$ such that, for $h \leq 1/C$ and all $t > 0$, with probability at least $1 - C \exp(-t^2) - \exp(-n h^d / C)/h^C$,*

$$\int_{[0,1]^d} \left( \hat{m}_n(x) - m(x) \right)^2 \mathrm{d}x \leq C h^{2\alpha} + \frac{C t^2}{n h^d}.$$

*In particular, if $h = n^{-1/(2\alpha+d)}$, then with probability at least $1 - C e^{-t^2} - n^C e^{-n^{2\alpha/(2\alpha+d)}/C}$,*

$$\int_{[0,1]^d} \left( \hat{m}_n(x) - m(x) \right)^2 \mathrm{d}x \leq C t^2 n^{-\frac{2\alpha}{2\alpha+d}}.$$

*Moreover, under the same condition on $h$, the truncated local polynomial estimator satisfies*

$$\int_{[0,1]^d} \mathbb{E}\left\{ \left( \tilde{m}_n(x) - m(x) \right)^2 \right\} \mathrm{d}x \leq C n^{-\frac{2\alpha}{2\alpha+d}}.$$

*Proof of Theorem A.6.* Throughout the proof, $C_1, C_2, \ldots > 0$ are quantities depending only on $c_K$, $C_K$, $c_X$, $C_X$, $L_K$, $d$, $p$ and $M$. For the first result, by Lemmas A.4 and A.5, with probability at least $1 - C_1 \exp(-t^2) - \exp(-n h^d / C_1)/h^{C_1}$,

$$\int_{[0,1]^d} \left( \hat{m}_n(x) - m(x) \right)^2 \mathrm{d}x \leq 2 \int_{[0,1]^d} \left( \hat{m}_n(x) - \hat{B}_n(x) \right)^2 \mathrm{d}x + 2 \int_{[0,1]^d} \left( \hat{B}_n(x) - m(x) \right)^2 \mathrm{d}x \leq \frac{C_1 t^2}{n h^d} + C_1 h^{2\alpha}.$$

For the second bound, taking $h = n^{-1/(2\alpha+d)}$ yields

$$\int_{[0,1]^d} \left( \hat{m}_n(x) - m(x) \right)^2 \mathrm{d}x \leq C_1 (t^2 + 1) n^{-\frac{2\alpha}{2\alpha+d}} \leq C_2 t^2 n^{-\frac{2\alpha}{2\alpha+d}},$$

with probability at least $1 - C_1 \exp(-t^2) - n^{C_1} \exp(-n^{2\alpha/(2\alpha+d)}/C_1)$, as taking $C_1 \geq e$ makes this probability trivial unless

$t \geq 1$. For the third inequality, by Fubini's theorem and integrating the tail probability, as $\sup_{x \in [0,1]^d} |\tilde{m}_n(x) - m(x)| \leq 2M$,

$$\int_{[0,1]^d} \mathbb{E}\left\{ \left( \tilde{m}_n(x) - m(x) \right)^2 \right\} \mathrm{d}x = C_2 n^{\frac{-2\alpha}{2\alpha+d}} \mathbb{E}\left\{ \frac{n^{\frac{2\alpha}{2\alpha+d}}}{C_2} \int_{[0,1]^d} \left( \tilde{m}_n(x) - m(x) \right)^2 \mathrm{d}x \right\}$$

$$= C_2 n^{\frac{-2\alpha}{2\alpha+d}} \int_0^\infty \mathbb{P}\left( \int_{[0,1]^d} \left( \tilde{m}_n(x) - m(x) \right)^2 \mathrm{d}x > C_2 s n^{-\frac{2\alpha}{2\alpha+d}} \right) \mathrm{d}s$$

$$\leq C_2 n^{\frac{-2\alpha}{2\alpha+d}} \int_0^{4M^2 n^{\frac{2\alpha}{2\alpha+d}}/C_2} \left\{ C_1 \exp(-s) + n^{C_1} \exp\left( -n^{2\alpha/(2\alpha+d)}/C_1 \right) \right\} \mathrm{d}s$$

$$\leq C_1 C_2 n^{\frac{-2\alpha}{2\alpha+d}} + 4M^2 n^{C_1} \exp\left( -n^{2\alpha/(2\alpha+d)}/C_1 \right).$$

Note that $C_1(C_1 + 1) \log n - n^{2\alpha/(2\alpha+d)} \to -\infty$ as $n \to \infty$. Thus, there exists $C_3 > 0$ such that for $n \geq C_3$, we have $n^{C_1} \exp\left( -n^{2\alpha/(2\alpha+d)}/C_1 \right) \leq 1/n$. We deduce that for $n \geq C_3$,

$$\int_{[0,1]^d} \mathbb{E}\left\{ \left( \tilde{m}_n(x) - m(x) \right)^2 \right\} \mathrm{d}x \leq C_1 C_2 n^{\frac{-2\alpha}{2\alpha+d}} + \frac{4M^2}{n} \leq C_4 n^{\frac{-2\alpha}{2\alpha+d}}.$$

As $\sup_{x \in [0,1]^d} |\tilde{m}_n(x) - m(x)| \leq 2M$, this holds for all $n \in \mathbb{N}$ after increasing $C_4$ to $C_5$ if necessary. $\square$

# B. Approximation Theory

We construct an explicit transformer that produces outputs similar to those of truncated local polynomial estimation, keeping track of its architecture and parameter magnitudes.

## B.1. ReLU Neural Networks

We begin by summarising some approximation properties of ReLU neural networks that will be useful later for our transformer construction.

**Definition B.1.** Let $d_{\mathrm{in}}, d_{\mathrm{out}}, N, L \in \mathbb{N}$ and $B > 0$. A function $f : \mathbb{R}^{d_{\mathrm{in}}} \to \mathbb{R}^{d_{\mathrm{out}}}$ is a *(ReLU) neural network* with width $N$, depth $L$ and all parameters bounded by $B$ if there exist $\boldsymbol{W}_\ell \in [-B, B]^{d_\ell \times d_{\ell-1}}$ and $b_\ell \in [-B, B]^{d_\ell}$ for $\ell \in [L+1]$, where $d_0 := d_{\mathrm{in}}, d_{L+1} := d_{\mathrm{out}}$ and $d_\ell \in [N]$ for $\ell \in [L]$, such that

$$f(\cdot) = A_{L+1} \circ \mathrm{ReLU} \circ A_L \circ \mathrm{ReLU} \circ \cdots \circ A_2 \circ \mathrm{ReLU} \circ A_1(\cdot),$$

where $A_\ell(z) := \boldsymbol{W}_\ell z + b_\ell$ for $\ell \in [L+1]$.

We may assume without loss of generality that $d_\ell = N$ for $\ell \in [L]$ by padding the weight matrices $\boldsymbol{W}_\ell$ and bias vectors $b_\ell$ with zeros.

**Lemma B.2** (Network composition). *Suppose that for $r \in \{1, 2\}$,*

$$f^{(r)}(\cdot) := A_{L^{(r)}+1}^{(r)} \circ \mathrm{ReLU} \circ \cdots \circ A_2^{(r)} \circ \mathrm{ReLU} \circ A_1^{(r)}(\cdot),$$

*where $A_\ell^{(r)}(z) := \boldsymbol{W}_\ell^{(r)} z + b_\ell^{(r)}$, $\boldsymbol{W}_\ell^{(r)} \in [-B^{(r)}, B^{(r)}]^{d_\ell^{(r)} \times d_{\ell-1}^{(r)}}$ and $b_\ell^{(r)} \in [-B^{(r)}, B^{(r)}]^{d_\ell^{(r)}}$ for $\ell \in [L^{(r)} + 1]$. Suppose further that the output dimension of $f^{(1)}$ is equal to the input dimension of $f^{(2)}$, i.e. $d_{L^{(1)}+1}^{(1)} = d_0^{(2)} =: m$. Then $f^{(2)} \circ f^{(1)}$ is a neural network with width $\max_{\ell \in [L^{(1)}]} d_\ell^{(1)} \vee \max_{\ell \in [L^{(2)}]} d_\ell^{(2)}$, depth $L^{(1)} + L^{(2)}$ and all parameters bounded by $\left\{ m \|\boldsymbol{W}_1^{(2)}\|_{\max} \left( \|\boldsymbol{W}_{L^{(1)}+1}^{(1)}\|_{\max} \vee \|b_{L^{(1)}+1}^{(1)}\|_\infty \right) + \|b_1^{(2)}\|_\infty \right\} \vee B^{(1)} \vee B^{(2)}.$*

*Proof.* Since

$$\boldsymbol{W}_1^{(2)} \left( \boldsymbol{W}_{L^{(1)}+1}^{(1)} z + b_{L^{(1)}+1}^{(1)} \right) + b_1^{(2)} = \boldsymbol{W}_1^{(2)} \boldsymbol{W}_{L^{(1)}+1}^{(1)} z + \left( \boldsymbol{W}_1^{(2)} b_{L^{(1)}+1}^{(1)} + b_1^{(2)} \right),$$

the result follows. $\square$

**Lemma B.3.** *Let $d' \in \mathbb{N}$, $B \geq 1$, $\boldsymbol{W}_1, \boldsymbol{W}_2 \in [-B, B]^{d' \times d'}$ and $b_1, b_2 \in [-B, B]^{d'}$. There exist $\boldsymbol{W}_1' \in [-B, B]^{3d' \times d'}$, $\boldsymbol{W}_2' \in [-B, B]^{d' \times 3d'}$, $b_1' \in [-B, B]^{3d'}$ and $b_2' \in [-B, B]^{d'}$ such that, for all $x \in \mathbb{R}^{d'}$,*

$$x + \boldsymbol{W}_2' \operatorname{ReLU}(\boldsymbol{W}_1' x + b_1') + b_2' = \boldsymbol{W}_2 \operatorname{ReLU}(\boldsymbol{W}_1 x + b_1) + b_2.$$

*Proof.* Take $b_1' := (b_1^\mathsf{T}, 0_{2d'}^\mathsf{T})^\mathsf{T}$ and $b_2' := b_2$, and let

$$\boldsymbol{W}_1' := \begin{pmatrix} \boldsymbol{W}_1 \\ \boldsymbol{I}_{d' \times d'} \\ -\boldsymbol{I}_{d' \times d'} \end{pmatrix}, \qquad\qquad \boldsymbol{W}_2' := \begin{pmatrix} \boldsymbol{W}_2 & -\boldsymbol{I}_{d' \times d'} & \boldsymbol{I}_{d' \times d'} \end{pmatrix}.$$

Since $\operatorname{ReLU}(-x) - \operatorname{ReLU}(x) = -x$ for all $x \in \mathbb{R}^{d'}$, we deduce that

$$x + \boldsymbol{W}_2' \operatorname{ReLU}(\boldsymbol{W}_1' x + b_1') + b_2' = x + \begin{pmatrix} \boldsymbol{W}_2 & -\boldsymbol{I}_{d' \times d'} & \boldsymbol{I}_{d' \times d'} \end{pmatrix} \begin{pmatrix} \operatorname{ReLU}(\boldsymbol{W}_1 x + b_1) \\ \operatorname{ReLU}(x) \\ \operatorname{ReLU}(-x) \end{pmatrix} + b_2$$

$$= \boldsymbol{W}_2 \operatorname{ReLU}(\boldsymbol{W}_1 x + b_1) + b_2,$$

as required. $\qquad\square$

**Lemma B.4.** *Let $C \geq 1$ and $N, L \in \mathbb{N}$. There exists a ReLU neural network $\phi : \mathbb{R}^2 \to \mathbb{R}$ with width $9N + 1$, depth $L$ and all parameters bounded by $32C^2 N$ such that for all $(x, y) \in [-C, C]^2$,*

$$|\phi(x, y) - xy| \leq 24C^2 N^{-L}.$$

*Proof.* The result follows from Lu et al. (2021, Lemma 4.2), where the upper bound on the magnitude of the parameters follows by inspecting the proofs of their Lemmas 5.1, 5.2 and 4.2. $\qquad\square$

The following lemma is an analogue of Lu et al. (2021, Lemma 5.3), but we extend the input domain from $[0, 1]^k$ to $[-C, C]^k$ and track the magnitude of the parameters.

**Lemma B.5.** *Let $C \geq 1$, $k \geq 2$ and $N, L \in \mathbb{N}$. There exists a ReLU neural network $\phi : \mathbb{R}^k \to \mathbb{R}$ with width $9(N+1) + 2k - 1$, depth $7kL(k-1)$ and all parameters bounded by $3C^k(40N + 40)^2$ such that for all $(x_1, \ldots, x_k) \in [-C, C]^k$,*

$$|\phi(x_1, \ldots, x_k) - x_1 \cdots x_k| \leq 30C^k(k-1)(N+1)^{-7kL}.$$

*Proof.* We first assume that $C = 1$. By Lemma B.4, there exists a ReLU neural network $\phi_2 : \mathbb{R}^2 \to \mathbb{R}$ with width $9(N+1) + 1$, depth $7kL$ and all parameters bounded by $40N + 40$ such that for all $(x, y) \in [-1.1, 1.1]^2$,

$$|\phi_2(x, y) - xy| \leq 30(N+1)^{-7kL}. \tag{11}$$

Now suppose that for some $m \in \{2, \ldots, k-1\}$, there exists $\phi_m : \mathbb{R}^m \to \mathbb{R}$ with width $9(N+1) + 2m - 1$, depth $7kL(m-1)$ and all parameters bounded by $3(40N+40)^2$, such that

$$|\phi_m(x_1, \ldots, x_m) - x_1 \cdots x_m| \leq 30(m-1)(N+1)^{-7kL}.$$

Here, the case $m = 2$ is proved by (11). We then define

$$\phi_{m+1}(x_1, \ldots, x_{m+1}) := \phi_2\big(\phi_m(x_1, \ldots, x_m), x_{m+1}\big).$$

Since $\operatorname{ReLU}(x_{m+1}) - \operatorname{ReLU}(-x_{m+1}) = x_{m+1}$, the identity function $x_{m+1} \mapsto x_{m+1}$ can be implemented by a ReLU neural network with width 2, any depth, and all parameters bounded by 1. Hence, by network composition (Lemma B.2), $\phi_{m+1} : \mathbb{R}^{m+1} \to \mathbb{R}$ can be implemented by a ReLU neural network with width $9(N+1) + 2(m+1) - 1$, depth $7kLm$ and all parameters bounded by $3(40N+40)^2$. Moreover, since $30(m-1)(N+1)^{-7kL} \leq 30(k-1)2^{-7k} \leq 0.1$, we have $\phi_m(x_1, \ldots, x_m) \in [-1.1, 1.1]$. Thus, by (11),

$$\big|\phi_{m+1}(x_1, \ldots, x_{m+1}) - x_1 \cdots x_{m+1}\big| = \big|\phi_2\big(\phi_m(x_1, \ldots, x_m), x_{m+1}\big) - x_1 \cdots x_{m+1}\big|$$

$$\leq \big|\phi_2\big(\phi_m(x_1, \ldots, x_m), x_{m+1}\big) - \phi_m(x_1, \ldots, x_m) \cdot x_{m+1}\big| + \big|\phi_m(x_1, \ldots, x_m) - x_1 \cdots x_m\big| \cdot |x_{m+1}|$$

$$\leq 30(N+1)^{-7kL} + 30(m-1)(N+1)^{-7kL} = 30m(N+1)^{-7kL}.$$

The claim for $C = 1$ thus follows from induction. Now for any $C \geq 1$, we have that $(x_1, \ldots, x_k) \mapsto C^k \phi_k(x_1/C, \ldots, x_k/C)$ is a neural network with width $9(N+1) + 2k - 1$, depth $7kL(k-1)$ and all parameters bounded by $3C^k(40N + 40)^2$. Moreover,

$$
\begin{aligned}
\left| C^k \phi_k(x_1/C, \ldots, x_k/C) - x_1 \cdots x_k \right| &= C^k \left| \phi_k(x_1/C, \ldots, x_k/C) - (x_1/C) \cdots (x_k/C) \right| \\
&\leq 30 C^k (k-1)(N+1)^{-7kL},
\end{aligned}
$$

for all $(x_1, \ldots, x_k) \in [-C, C]^k$. $\qquad\square$

**Lemma B.6.** *Let $C \geq 1$, $d, k, N, L \in \mathbb{N}$ and $\nu = (\nu_1, \ldots, \nu_d) \in \mathbb{N}_0^d$ be such that $|\nu| \leq k$. There exists a ReLU neural network $\psi : \mathbb{R}^d \to \mathbb{R}$ with width $9(N+1) + 2k - 1$, depth $7kL(k-1) + 1$ and all parameters bounded by $3(k+1)C^k(40N+40)^2$ such that for all $x = (x_1, \ldots, x_d) \in [-C, C]^d$,*

$$
|\psi(x_1, \ldots, x_d) - x^\nu| \leq 30 C^k (k-1)(N+1)^{-7kL}.
$$

*Proof.* Assume without loss of generality that $x^\nu = x_1^{\nu_1} \cdots x_m^{\nu_m}$, where $\nu_j > 0$ for $j \in [m]$ and $\sum_{j=1}^m \nu_m = k_0 \leq k$. Since $\mathrm{ReLU}(a) - \mathrm{ReLU}(-a) = a$ for all $a \in \mathbb{R}$, there exists a neural network $\psi_1 : \mathbb{R}^d \to \mathbb{R}^{k_0}$ with width $2k$, depth $1$ and all parameters bounded by $1$ such that

$$
\psi_1(x) := (x_1 \mathbf{1}_{\nu_1}^\mathsf{T}, \cdots, x_m \mathbf{1}_{\nu_m}^\mathsf{T})^\mathsf{T} \in \mathbb{R}^{k_0}.
$$

Therefore, by Lemma B.5, there exists a neural network $\phi$ with width $9(N+1) + 2k - 1$, depth $7kL(k-1)$ and all parameters bounded by $3C^k(40N+40)^2$ such that

$$
\left| \phi(\psi_1(x)) - x^\nu \right| \leq 30 C^k (k-1)(N+1)^{-7kL}.
$$

Finally, by network composition, we have that $\psi := \phi \circ \psi_1$ is a neural network with width $9(N+1) + 2k - 1$, depth $7kL(k-1) + 1$ and all parameters bounded by $3(k+1)C^k(40N+40)^2$. $\qquad\square$

### B.2. Transformer Construction

In the next lemma, we construct a transformer that yields the centred covariates and a scaled version of the kernel matrix.

**Lemma B.7.** *Let $n, d \in \mathbb{N}$ and take $X_1, \ldots, X_n \in [0,1]^d$ and $Y_1, \ldots, Y_n \in \mathbb{R}$. Write $\mathbf{X} := (X_1, \ldots, X_n)^\mathsf{T} \in \mathbb{R}^{n \times d}$ and $Y := (Y_1, \ldots, Y_n)^\mathsf{T} \in \mathbb{R}^n$. Let $h > 0$ and define $K_h(\mathbf{X}, X_{n+1}) \in \mathbb{R}^n$ by $K_h(\mathbf{X}, X_{n+1})_i := K\big((X_i - X_{n+1})/h\big)/h^d$ for $i \in [n]$, where $K : \mathbb{R}^d \to \mathbb{R}$ is given by $K(x) := (1 - \|x\|_1)_+^2$. Let $\sqrt{K_h(\mathbf{X}, X_{n+1})} \in \mathbb{R}^n$ be defined entrywise. Take $d_e, d_{\mathrm{ffn}} \in \mathbb{N}$ with $d_e \geq 2d + 4$ and $d_{\mathrm{ffn}} \geq 2d + 2$, and let $B := 1 \vee (d/h) \vee (n^{-1/2} h^{-d/2})$. Then there exists a transformer $\mathrm{TF} \in \mathcal{T}(d_e, d_{\mathrm{ffn}}, 3, B)$ such that*

$$
\begin{aligned}
&\mathrm{TF} \circ \mathrm{Embed}_{d_e}\big((X_i, Y_i)_{i \in [n]}, X_{n+1}\big) \\
&= \begin{pmatrix} \mathbf{X} & Y & (\mathbf{X} - \mathbf{1}_n X_{n+1}^\mathsf{T})/h & n^{-1/2}\sqrt{K_h(\mathbf{X}, X_{n+1})} & \mathbf{0}_{n \times (d_e - 2d - 4)} & \mathbf{1}_n & \mathbf{0}_n \\ X_{n+1}^\mathsf{T} & 0 & \mathbf{0}_d^\mathsf{T} & n^{-1/2} h^{-d/2} & \mathbf{0}_{d_e - 2d - 4}^\mathsf{T} & 1 & 1 \end{pmatrix} \in \mathbb{R}^{(n+1) \times d_e}.
\end{aligned}
$$

*Proof.* The input to the desired transformer can be written as

$$
\mathbf{Z}_{\mathrm{in}} := \mathrm{Embed}_{d_e}\big((X_i, Y_i)_{i \in [n]}, X_{n+1}\big) = \begin{pmatrix} \mathbf{X} & Y & \mathbf{0}_{n \times (d_e - d - 2)} & \mathbf{0}_n \\ X_{n+1}^\mathsf{T} & 0 & \mathbf{0}_{d_e - d - 2}^\mathsf{T} & 1 \end{pmatrix} \in \mathbb{R}^{(n+1) \times d_e}.
$$

Let $\mathrm{Attn}^{(1)}$ be a linear attention layer as in Definition 2.1 with query, key and value matrices all zero, so that $\mathrm{Attn}^{(1)}$ is the identity map on $\mathbb{R}^{(n+1) \times d_e}$. Define $b_1^{(1)} := -\mathbf{1}_{d_{\mathrm{ffn}}} \in \mathbb{R}^{d_{\mathrm{ffn}}}$ and $b_2^{(1)} := (\mathbf{0}_{d_e - 2}^\mathsf{T}, 1, 0)^\mathsf{T} \in \mathbb{R}^{d_e}$, and let

$$
\begin{aligned}
\mathbf{W}_1^{(1)} &:= \begin{pmatrix} \mathbf{I}_{d \times d} & \mathbf{0}_{d \times (d_e - d - 1)} & \mathbf{1}_d \\ \mathbf{0}_{(d_{\mathrm{ffn}} - d) \times d} & \mathbf{0}_{(d_{\mathrm{ffn}} - d) \times (d_e - d - 1)} & \mathbf{0}_{d_{\mathrm{ffn}} - d} \end{pmatrix} \in \mathbb{R}^{d_{\mathrm{ffn}} \times d_e}, \\
\mathbf{W}_2^{(1)} &:= \begin{pmatrix} \mathbf{0}_{(d+1) \times d} & \mathbf{0}_{(d+1) \times (d_{\mathrm{ffn}} - d)} \\ \mathbf{I}_{d \times d} & \mathbf{0}_{d \times (d_{\mathrm{ffn}} - d)} \\ \mathbf{0}_{(d_e - 2d - 1) \times d} & \mathbf{0}_{(d_e - 2d - 1) \times (d_{\mathrm{ffn}} - d)} \end{pmatrix} \in \mathbb{R}^{d_e \times d_{\mathrm{ffn}}}.
\end{aligned}
$$

Then, since $X_i \in [0,1]^d$ for $i \in [n+1]$, we have

$$\mathrm{ReLU}\big(\boldsymbol{W}_1^{(1)}\boldsymbol{Z}_{\mathrm{in}}^{\mathsf{T}} + b_1^{(1)}1_{n+1}^{\mathsf{T}}\big) = \mathrm{ReLU}\begin{pmatrix} \boldsymbol{X}^{\mathsf{T}} - 1_d1_n^{\mathsf{T}} & X_{n+1} \\ -1_{d_{\mathrm{ffn}}-d}1_n^{\mathsf{T}} & -1_{d_{\mathrm{ffn}}-d} \end{pmatrix} = \begin{pmatrix} \boldsymbol{0}_{d\times n} & X_{n+1} \\ \boldsymbol{0}_{(d_{\mathrm{ffn}}-d)\times n} & 0_{d_{\mathrm{ffn}}-d} \end{pmatrix} \in \mathbb{R}^{d_{\mathrm{ffn}}\times(n+1)}.$$

Hence, writing $\mathrm{FFN}^{(1)} := \mathrm{FFN}_{\boldsymbol{W}_1^{(1)},\boldsymbol{W}_2^{(1)},b_1^{(1)},b_2^{(1)}}$, the first transformer block gives

$$\begin{aligned} \boldsymbol{Z}^{(1)} &:= \mathrm{FFN}^{(1)} \circ \mathrm{Attn}^{(1)}(\boldsymbol{Z}_{\mathrm{in}}) = \mathrm{FFN}^{(1)}(\boldsymbol{Z}_{\mathrm{in}}) \\ &= \boldsymbol{Z}_{\mathrm{in}} + \big\{\boldsymbol{W}_2^{(1)}\mathrm{ReLU}\big(\boldsymbol{W}_1^{(1)}\boldsymbol{Z}_{\mathrm{in}}^{\mathsf{T}} + b_1^{(1)}1_{n+1}^{\mathsf{T}}\big) + b_2^{(1)}1_{n+1}^{\mathsf{T}}\big\}^{\mathsf{T}} \\ &= \begin{pmatrix} \boldsymbol{X} & Y & \boldsymbol{0}_{n\times d} & \boldsymbol{0}_{n\times(d_{\mathrm{e}}-2d-3)} & 1_n & 0_n \\ X_{n+1}^{\mathsf{T}} & 0 & X_{n+1}^{\mathsf{T}} & 0_{d_{\mathrm{e}}-2d-3}^{\mathsf{T}} & 1 & 1 \end{pmatrix} \in \mathbb{R}^{(n+1)\times d_{\mathrm{e}}}. \end{aligned}$$

For the second attention layer, let $\mathrm{Attn}^{(2)} := \mathrm{Attn}_{\boldsymbol{Q}^{(2)},\boldsymbol{K}^{(2)},\boldsymbol{V}^{(2)}}$ where

$$\boldsymbol{Q}^{(2)} := \frac{1}{\sqrt{d_{\mathrm{e}}}}\begin{pmatrix} \boldsymbol{0}_{(d_{\mathrm{e}}-2)\times d_{\mathrm{e}}} \\ 1_{d_{\mathrm{e}}}^{\mathsf{T}} \\ -1_{d_{\mathrm{e}}}^{\mathsf{T}} \end{pmatrix} \in \mathbb{R}^{d_{\mathrm{e}}\times d_{\mathrm{e}}}, \qquad \boldsymbol{K}^{(2)} := \frac{1}{\sqrt{d_{\mathrm{e}}}}\begin{pmatrix} \boldsymbol{0}_{(d_{\mathrm{e}}-2)\times d_{\mathrm{e}}} \\ 1_{d_{\mathrm{e}}}^{\mathsf{T}} \\ 0_{d_{\mathrm{e}}}^{\mathsf{T}} \end{pmatrix} \in \mathbb{R}^{d_{\mathrm{e}}\times d_{\mathrm{e}}},$$

$$\boldsymbol{V}^{(2)} := \begin{pmatrix} \boldsymbol{0}_{(d+1)\times(d+1)} & \boldsymbol{0}_{(d+1)\times d} & \boldsymbol{0}_{(d+1)\times(d_{\mathrm{e}}-2d-1)} \\ \boldsymbol{0}_{d\times(d+1)} & \boldsymbol{I}_{d\times d} & \boldsymbol{0}_{d\times(d_{\mathrm{e}}-2d-1)} \\ \boldsymbol{0}_{(d_{\mathrm{e}}-2d-1)\times(d+1)} & \boldsymbol{0}_{(d_{\mathrm{e}}-2d-1)\times d} & \boldsymbol{0}_{(d_{\mathrm{e}}-2d-1)\times(d_{\mathrm{e}}-2d-1)} \end{pmatrix} \in \mathbb{R}^{d_{\mathrm{e}}\times d_{\mathrm{e}}}.$$

Then we may write

$$\boldsymbol{Z}^{(1)}\boldsymbol{Q}^{(2)} = \frac{1}{\sqrt{d_{\mathrm{e}}}}\begin{pmatrix} 1_n1_{d_{\mathrm{e}}}^{\mathsf{T}} \\ 0_{d_{\mathrm{e}}}^{\mathsf{T}} \end{pmatrix}, \quad \boldsymbol{Z}^{(1)}\boldsymbol{K}^{(2)} = \frac{1}{\sqrt{d_{\mathrm{e}}}}1_{n+1}1_{d_{\mathrm{e}}}^{\mathsf{T}},$$

$$\boldsymbol{Z}^{(1)}\boldsymbol{V}^{(2)} = \begin{pmatrix} \boldsymbol{0}_{n\times(d+1)} & \boldsymbol{0}_{n\times d} & \boldsymbol{0}_{n\times(d_{\mathrm{e}}-2d-1)} \\ 0_{d+1}^{\mathsf{T}} & X_{n+1}^{\mathsf{T}} & 0_{d_{\mathrm{e}}-2d-1}^{\mathsf{T}} \end{pmatrix}.$$

Therefore, the output of the second attention layer is

$$\begin{aligned} \mathrm{Attn}^{(2)}(\boldsymbol{Z}^{(1)}) &= \boldsymbol{Z}^{(1)} + \boldsymbol{Z}^{(1)}\boldsymbol{Q}^{(2)}(\boldsymbol{Z}^{(1)}\boldsymbol{K}^{(2)})^{\mathsf{T}}\boldsymbol{Z}^{(1)}\boldsymbol{V}^{(2)} \\ &= \begin{pmatrix} \boldsymbol{X} & Y & 1_n X_{n+1}^{\mathsf{T}} & \boldsymbol{0}_{n\times(d_{\mathrm{e}}-2d-3)} & 1_n & 0_n \\ X_{n+1}^{\mathsf{T}} & 0 & X_{n+1}^{\mathsf{T}} & 0_{d_{\mathrm{e}}-2d-3}^{\mathsf{T}} & 1 & 1 \end{pmatrix} \in \mathbb{R}^{(n+1)\times d_{\mathrm{e}}}. \end{aligned}$$

For the second feed-forward layer, let $b_1^{(2)} := 0_{d_{\mathrm{ffn}}} \in \mathbb{R}^{d_{\mathrm{ffn}}}$ and define

$$\boldsymbol{W}_1^{(2)} := \begin{pmatrix} \boldsymbol{I}_{d\times d}/h & 0_d & -\boldsymbol{I}_{d\times d}/h & \boldsymbol{0}_{d\times(d_{\mathrm{e}}-2d-1)} \\ -\boldsymbol{I}_{d\times d}/h & 0_d & \boldsymbol{I}_{d\times d}/h & \boldsymbol{0}_{d\times(d_{\mathrm{e}}-2d-1)} \\ \boldsymbol{0}_{(d_{\mathrm{ffn}}-2d)\times d} & 0_{d_{\mathrm{ffn}}-2d} & \boldsymbol{0}_{(d_{\mathrm{ffn}}-2d)\times d} & \boldsymbol{0}_{(d_{\mathrm{ffn}}-2d)\times(d_{\mathrm{e}}-2d-1)} \end{pmatrix} \in \mathbb{R}^{d_{\mathrm{ffn}}\times d_{\mathrm{e}}}.$$

Then

$$\mathrm{ReLU}\Big(\boldsymbol{W}_1^{(2)}\mathrm{Attn}^{(2)}\big(\boldsymbol{Z}^{(1)}\big)^{\mathsf{T}} + b_1^{(2)}1_{n+1}^{\mathsf{T}}\Big) = \begin{pmatrix} \mathrm{ReLU}(\boldsymbol{X}^{\mathsf{T}} - X_{n+1}1_n^{\mathsf{T}})/h & 0_d \\ \mathrm{ReLU}(X_{n+1}1_n^{\mathsf{T}} - \boldsymbol{X}^{\mathsf{T}})/h & 0_d \\ \boldsymbol{0}_{(d_{\mathrm{ffn}}-2d)\times n} & 0_{d_{\mathrm{ffn}}-2d} \end{pmatrix}.$$

Next, let $b_2^{(2)} := (0_{2d+1}^{\mathsf{T}}, 1, 0_{d_{\mathrm{e}}-2d-2}^{\mathsf{T}})^{\mathsf{T}} \in \mathbb{R}^{d_{\mathrm{e}}}$ and define

$$\boldsymbol{W}_2^{(2)} := \begin{pmatrix} \boldsymbol{0}_{(2d+1)\times d} & \boldsymbol{0}_{(2d+1)\times d} & \boldsymbol{0}_{(2d+1)\times(d_{\mathrm{ffn}}-2d)} \\ -1_d^{\mathsf{T}} & -1_d^{\mathsf{T}} & 0_{d_{\mathrm{ffn}}-2d}^{\mathsf{T}} \\ \boldsymbol{0}_{(d_{\mathrm{e}}-2d-2)\times d} & \boldsymbol{0}_{(d_{\mathrm{e}}-2d-2)\times d} & \boldsymbol{0}_{(d_{\mathrm{e}}-2d-2)\times(d_{\mathrm{ffn}}-2d)} \end{pmatrix} \in \mathbb{R}^{d_{\mathrm{e}}\times d_{\mathrm{ffn}}}.$$

Defining $\tilde{K} \in \mathbb{R}^n$ by $\tilde{K}^{\mathsf{T}} := 1_n^{\mathsf{T}} - 1_d^{\mathsf{T}}\mathrm{ReLU}(\boldsymbol{X}^{\mathsf{T}} - X_{n+1}1_n^{\mathsf{T}})/h - 1_d^{\mathsf{T}}\mathrm{ReLU}(X_{n+1}1_n^{\mathsf{T}} - \boldsymbol{X}^{\mathsf{T}})/h$, it follows that

$$\boldsymbol{W}_2^{(2)}\mathrm{ReLU}\Big(\boldsymbol{W}_1^{(2)}\mathrm{Attn}^{(2)}\big(\boldsymbol{Z}^{(1)}\big)^{\mathsf{T}} + b_1^{(2)}1_{n+1}^{\mathsf{T}}\Big) + b_2^{(2)}1_{n+1}^{\mathsf{T}} = \begin{pmatrix} \boldsymbol{0}_{(2d+1)\times n} & 0_{2d+1} \\ \tilde{K}^{\mathsf{T}} & 1 \\ \boldsymbol{0}_{(d_{\mathrm{e}}-2d-2)\times n} & 0_{d_{\mathrm{e}}-2d-2} \end{pmatrix}.$$

Writing $\text{FFN}^{(2)} := \text{FFN}_{\boldsymbol{W}_1^{(2)},\boldsymbol{W}_2^{(2)},b_1^{(2)},b_2^{(2)}}$, the output of the second transformer block is

$$
\begin{aligned}
\boldsymbol{Z}^{(2)} &:= \text{FFN}^{(2)} \circ \text{Attn}^{(2)}\big(\boldsymbol{Z}^{(1)}\big) \\
&= \text{Attn}^{(2)}\big(\boldsymbol{Z}^{(1)}\big) + \Big\{\boldsymbol{W}_2^{(2)}\text{ReLU}\Big(\boldsymbol{W}_1^{(2)}\text{Attn}^{(2)}\big(\boldsymbol{Z}^{(1)}\big)^{\mathsf{T}} + b_1^{(2)}1_{n+1}^{\mathsf{T}}\Big) + b_2^{(2)}1_{n+1}^{\mathsf{T}}\Big\}^{\mathsf{T}} \\
&= \begin{pmatrix} \boldsymbol{X} & Y & 1_n X_{n+1}^{\mathsf{T}} & \tilde{K} & \boldsymbol{0}_{n\times(d_e-2d-4)} & 1_n & 0_n \\ X_{n+1}^{\mathsf{T}} & 0 & X_{n+1}^{\mathsf{T}} & 1 & 0_{d_e-2d-4}^{\mathsf{T}} & 1 & 1 \end{pmatrix} \in \mathbb{R}^{(n+1)\times d_e}.
\end{aligned}
$$

We take the third attention layer $\text{Attn}^{(3)}$ to be the identity function. For the third feed-forward layer, let $b_1^{(3)} := (0, d/h, 1_d^{\mathsf{T}}/h, 0_{d_{\text{ffn}}-d-2}^{\mathsf{T}})^{\mathsf{T}} \in \mathbb{R}^{d_{\text{ffn}}}$ and

$$
\boldsymbol{W}_1^{(3)} := \begin{pmatrix} \boldsymbol{0}_{2\times d} & 0_2 & \boldsymbol{0}_{2\times d} & 1_2 & \boldsymbol{0}_{2\times(d_e-2d-2)} \\ \boldsymbol{I}_{d\times d}/h & 0_d & -\boldsymbol{I}_{d\times d}/h & 0_d & \boldsymbol{0}_{d\times(d_e-2d-2)} \\ \boldsymbol{0}_{d\times d} & 0_d & \boldsymbol{I}_{d\times d} & 0_d & \boldsymbol{0}_{d\times(d_e-2d-2)} \\ \boldsymbol{0}_{(d_{\text{ffn}}-2d-2)\times d} & 0_{d_{\text{ffn}}-2d-2} & \boldsymbol{0}_{(d_{\text{ffn}}-2d-2)\times d} & 0_{d_{\text{ffn}}-2d-2} & \boldsymbol{0}_{(d_{\text{ffn}}-2d-2)\times(d_e-2d-2)} \end{pmatrix}.
$$

Then

$$
\boldsymbol{W}_1^{(3)}\boldsymbol{Z}^{(2),\mathsf{T}} + b_1^{(3)}1_{n+1}^{\mathsf{T}} = \begin{pmatrix} \tilde{K}^{\mathsf{T}} & 1 \\ \tilde{K}^{\mathsf{T}} + d1_n^{\mathsf{T}}/h & 1+d/h \\ \big(\boldsymbol{X}^{\mathsf{T}} - X_{n+1}1_n^{\mathsf{T}} + 1_d1_n^{\mathsf{T}}\big)/h & 1_d/h \\ X_{n+1}1_n^{\mathsf{T}} & X_{n+1} \\ \boldsymbol{0}_{(d_{\text{ffn}}-2d-2)\times n} & 0_{d_{\text{ffn}}-2d-2} \end{pmatrix} \in \mathbb{R}^{d_{\text{ffn}}\times(n+1)}.
$$

Moreover, writing $\tilde{K}_i$ for the $i$th component of $\tilde{K}$ for $i \in [n]$, we have

$$
\begin{aligned}
\text{ReLU}(\tilde{K}_i) &= \left\{ 1 - \sum_{j=1}^d \left(\frac{X_{i,j} - X_{n+1,j}}{h}\right)_+ - \sum_{j=1}^d \left(\frac{X_{n+1,j} - X_{i,j}}{h}\right)_+ \right\}_+ \\
&= \left\{ 1 - \sum_{j=1}^d \left|\frac{X_{i,j} - X_{n+1,j}}{h}\right| \right\}_+ = \left\{ 1 - \left\|\frac{X_i - X_{n+1}}{h}\right\|_1 \right\}_+ \\
&= \sqrt{h^d K_h(\boldsymbol{X}, X_{n+1})_i}.
\end{aligned}
$$

Therefore, interpreting the square root as acting entrywise, $\text{ReLU}(\tilde{K}) = \sqrt{h^d K_h(\boldsymbol{X}, X_{n+1})}$. As $X_i \in [0,1]^d$ for each $i \in [n+1]$ and $\tilde{K}_i \geq -d/h$ for $i \in [n]$, we have

$$
\text{ReLU}\Big(\boldsymbol{W}_1^{(3)}\boldsymbol{Z}^{(2),\mathsf{T}} + b_1^{(3)}1_{n+1}^{\mathsf{T}}\Big) = \begin{pmatrix} \sqrt{h^d K_h(\boldsymbol{X}, X_{n+1})^{\mathsf{T}}} & 1 \\ \tilde{K}^{\mathsf{T}} + d1_n^{\mathsf{T}}/h & 1+d/h \\ \big(1_d1_n^{\mathsf{T}} + \boldsymbol{X}^{\mathsf{T}} - X_{n+1}1_n^{\mathsf{T}}\big)/h & 1_d/h \\ X_{n+1}1_n^{\mathsf{T}} & X_{n+1} \\ 0_{(d_{\text{ffn}}-2d-2)\times n} & 0_{d_{\text{ffn}}-2d-2} \end{pmatrix} \in \mathbb{R}^{d_{\text{ffn}}\times(n+1)}.
$$

Now let $b_2^{(3)} := (0_{d+1}^{\mathsf{T}}, -1_d^{\mathsf{T}}/h, d/h, 0_{d_e-2d-2}^{\mathsf{T}})^{\mathsf{T}} \in \mathbb{R}^{d_e}$ and

$$
\boldsymbol{W}_2^{(3)} := \begin{pmatrix} 0_{d+1} & 0_{d+1} & \boldsymbol{0}_{(d+1)\times d} & \boldsymbol{0}_{(d+1)\times d} & \boldsymbol{0}_{(d+1)\times(d_{\text{ffn}}-2d-2)} \\ 0_d & 0_d & \boldsymbol{I}_{d\times d} & -\boldsymbol{I}_{d\times d} & \boldsymbol{0}_{d\times(d_{\text{ffn}}-2d-2)} \\ n^{-1/2}h^{-d/2} & -1 & 0_d^{\mathsf{T}} & 0_d^{\mathsf{T}} & 0_{d_{\text{ffn}}-2d-2}^{\mathsf{T}} \\ 0_{d_e-2d-2} & 0_{d_e-2d-2} & \boldsymbol{0}_{(d_e-2d-2)\times d} & \boldsymbol{0}_{(d_e-2d-2)\times d} & \boldsymbol{0}_{(d_e-2d-2)\times(d_{\text{ffn}}-2d-2)} \end{pmatrix} \in \mathbb{R}^{d_e\times d_{\text{ffn}}}.
$$

Then

$$
\begin{aligned}
&\boldsymbol{W}_2^{(3)}\text{ReLU}\Big(\boldsymbol{W}_1^{(3)}\boldsymbol{Z}^{(2),\mathsf{T}} + b_1^{(3)}1_{n+1}^{\mathsf{T}}\Big) + b_2^{(3)}1_{n+1}^{\mathsf{T}} \\
&\qquad = \begin{pmatrix} \boldsymbol{0}_{(d+1)\times n} & 0_{d+1} \\ \big(\boldsymbol{X}^{\mathsf{T}} - X_{n+1}1_n^{\mathsf{T}}\big)/h - X_{n+1}1_n^{\mathsf{T}} & -X_{n+1} \\ n^{-1/2}\sqrt{K_h(\boldsymbol{X}, X_{n+1})^{\mathsf{T}}} - \tilde{K}^{\mathsf{T}} & n^{-1/2}h^{-d/2} - 1 \\ \boldsymbol{0}_{(d_e-2d-2)\times n} & 0_{d_e-2d-2} \end{pmatrix} \in \mathbb{R}^{d_e\times(n+1)}.
\end{aligned}
$$

Finally, with $\mathrm{FFN}^{(3)} := \mathrm{FFN}_{\boldsymbol{W}_1^{(3)}, \boldsymbol{W}_2^{(3)}, b_1^{(3)}, b_2^{(3)}}$, the output of the third transformer block is

$$
\begin{aligned}
\boldsymbol{Z}^{(3)} &:= \mathrm{FFN}^{(3)} \circ \mathrm{Attn}^{(3)}\big(\boldsymbol{Z}^{(2)}\big) = \mathrm{FFN}^{(3)}\big(\boldsymbol{Z}^{(2)}\big) \\
&= \boldsymbol{Z}^{(2)} + \left\{ \boldsymbol{W}_2^{(3)} \mathrm{ReLU}\Big( \boldsymbol{W}_1^{(3)} \boldsymbol{Z}^{(2),\mathsf{T}} + b_1^{(3)} 1_{n+1}^{\mathsf{T}} \Big) + b_2^{(3)} 1_{n+1}^{\mathsf{T}} \right\}^{\mathsf{T}} \\
&= \begin{pmatrix} \boldsymbol{X} & Y & \big(\boldsymbol{X} - 1_n X_{n+1}^{\mathsf{T}}\big)/h & n^{-1/2}\sqrt{K_h(\boldsymbol{X}, X_{n+1})} & \boldsymbol{0}_{n \times (d_{\mathrm{e}} - 2d - 4)} & 1_n & 0_n \\ X_{n+1}^{\mathsf{T}} & 0 & 0_d^{\mathsf{T}} & n^{-1/2}h^{-d/2} & 0_{d_{\mathrm{e}} - 2d - 4}^{\mathsf{T}} & 1 & 1 \end{pmatrix},
\end{aligned}
$$

as required. $\qquad\square$

The following lemma constructs the kernel-weighted polynomials and responses using a transformer. Let $h := n^{-1/(2\alpha + d)}$, let $\boldsymbol{P}_h(X_{n+1}) \in \mathbb{R}^{n \times D}$ be defined as in Section A.1, and write $\tilde{\boldsymbol{X}} := n^{-1/2}\sqrt{K_h(X_{n+1})}\boldsymbol{P}_h(X_{n+1}) \in \mathbb{R}^{n \times D}$ and $\tilde{Y} := n^{-1/2}\sqrt{K_h(X_{n+1})}Y \in \mathbb{R}^n$.

**Lemma B.8.** *Let $L_0 > 0$, $p := \lceil \alpha \rceil$, $D := \binom{d+p}{p}$, $d_{\mathrm{e}} := 2d + 2D + 5$, $d_{\mathrm{ffn}} := 3(D+1)(28+2p)$, $L := 7p(p+1)\lceil L_0 \log n \rceil + 4$ and $B := 3d \cdot 120^2 \cdot (p+2)(2M+2)^{p+1} n^{(p+1)/(2\alpha + d)}$. Then there exists a transformer $\mathrm{TF} \in \mathcal{T}(d_{\mathrm{e}}, d_{\mathrm{ffn}}, L, B)$ such that*

$$
\begin{aligned}
&\mathrm{TF} \circ \mathrm{Embed}_{d_{\mathrm{e}}}\big((X_i, Y_i)_{i \in [n]}, X_{n+1}\big) \\
&= \begin{pmatrix} \boldsymbol{X} & Y & \big(\boldsymbol{X} - 1_n X_{n+1}^{\mathsf{T}}\big)/h & n^{-1/2}\sqrt{K_h(\boldsymbol{X}, X_{n+1})} & \check{\boldsymbol{X}} & \check{Y} & \boldsymbol{0}_{n \times D} & 1_n & 0_n \\ X_{n+1}^{\mathsf{T}} & 0 & 0_d^{\mathsf{T}} & n^{-1/2}h^{-d/2} & a^{\mathsf{T}} & b & 0_D^{\mathsf{T}} & 1 & 1 \end{pmatrix},
\end{aligned}
$$

*where $\check{\boldsymbol{X}} \in \mathbb{R}^{n \times D}$, $\check{Y} \in \mathbb{R}^n$, $a \in \mathbb{R}^D$ and $b \in \mathbb{R}$ satisfy*

$$
\|\check{\boldsymbol{X}} - \tilde{\boldsymbol{X}}\|_{\max} \vee \|\check{Y} - \tilde{Y}\|_\infty \vee \|a\|_\infty \vee |b| \leq 30p \cdot (2M+2)^{p+1} \cdot n^{-7(p+1)L_0 + 2}.
$$

*Proof.* By Lemma B.7, there exists $\mathrm{TF}^{(1)} \in \mathcal{T}(d_{\mathrm{e}}, d_{\mathrm{ffn}}, 3, B)$ such that

$$
\begin{aligned}
\boldsymbol{Z}^{(1)} &:= \mathrm{TF}^{(1)} \circ \mathrm{Embed}_{d_{\mathrm{e}}}\big((X_i, Y_i)_{i \in [n]}, X_{n+1}\big) \\
&= \begin{pmatrix} \boldsymbol{X} & Y & \big(\boldsymbol{X} - 1_n X_{n+1}^{\mathsf{T}}\big)/h & n^{-1/2}\sqrt{K_h(\boldsymbol{X}, X_{n+1})} & \boldsymbol{0}_{n \times (2D+1)} & 1_n & 0_n \\ X_{n+1}^{\mathsf{T}} & 0 & 0_d^{\mathsf{T}} & n^{-1/2}h^{-d/2} & 0_{2D+1}^{\mathsf{T}} & 1 & 1 \end{pmatrix} \in \mathbb{R}^{(n+1) \times d_{\mathrm{e}}}.
\end{aligned}
$$

Next, note that each entry of the $i$th row of $\tilde{\boldsymbol{X}}$, as well as the $i$th entry of $\tilde{Y}$, is a monomial in the $i$th row of $\big(Y, (\boldsymbol{X} - 1_n X_{n+1}^{\mathsf{T}})/h, n^{-1/2}\sqrt{K_h(\boldsymbol{X}, X_{n+1})}\big)$ of degree at most $p + 1$. Thus, by Lemma B.3 and by applying Lemma B.6 with $N = 2$, $k = p + 1$, $L = \lceil L_0 \log n \rceil$ and $C = 2(M+1)n^{1/(2\alpha + d)}$ therein, there exists $\mathrm{TF}^{(2)} \in \mathcal{T}(d_{\mathrm{e}}, d_{\mathrm{ffn}}, L - 3, B)$ with all attention layers being the identity map such that

$$
\mathrm{TF}^{(2)}(\boldsymbol{Z}^{(1)}) = \begin{pmatrix} \boldsymbol{X} & Y & \big(\boldsymbol{X} - 1_n X_{n+1}^{\mathsf{T}}\big)/h & n^{-1/2}\sqrt{K_h(\boldsymbol{X}, X_{n+1})} & \check{\boldsymbol{X}} & \check{Y} & \boldsymbol{0}_{n \times D} & 1_n & 0_n \\ X_{n+1}^{\mathsf{T}} & 0 & 0_d^{\mathsf{T}} & n^{-1/2}h^{-d/2} & a^{\mathsf{T}} & b & 0_D^{\mathsf{T}} & 1 & 1 \end{pmatrix},
$$

where

$$
\begin{aligned}
\|\check{\boldsymbol{X}} - \tilde{\boldsymbol{X}}\|_{\max} \vee \|\check{Y} - \tilde{Y}\|_\infty \vee \|a\|_\infty \vee |b| &\leq 30\big(2(M+1)n^{1/(2\alpha + d)}\big)^{p+1} p \cdot 3^{-7(p+1)L_0 \log n} \\
&\leq 30p \cdot (2M+2)^{p+1} \cdot n^{-7(p+1)L_0 + 2}.
\end{aligned}
$$

Thus $\mathrm{TF} := \mathrm{TF}^{(2)} \circ \mathrm{TF}^{(1)} \in \mathcal{T}(d_{\mathrm{e}}, d_{\mathrm{ffn}}, L, B)$ satisfies the conditions of the lemma. $\qquad\square$

The next lemma quantifies the rate of convergence of the gradient descent algorithm for strongly convex functions with errors in the gradients. A similar result can be found in Bai et al. (2023).

**Lemma B.9.** *Let $C_1, C_2 > 0$, $\epsilon \geq 0$, $m \in \mathbb{N}$, and let $f : \mathbb{R}^m \to \mathbb{R}$. Suppose that $w \mapsto f(w) - (C_1/2)\|w\|_2^2$ is convex, let $w_* := \mathrm{argmin}_{w \in \mathbb{R}^m} f(w)$, and let $R \geq 2\|w_*\|_2$. Further suppose that $f$ is differentiable with $\|\nabla f(w) - \nabla f(w')\|_2 \leq C_2 \|w - w'\|_2$ for all $w, w' \in \mathbb{R}^m$ such that $\|w\|_2, \|w'\|_2 \leq R$. Let $w_0 := 0_m$, and for $t \in \mathbb{N}_0$, define $w_{t+1} := w_t - g_t/C_2$ where $g_t \in \mathbb{R}^m$ satisfies $\|g_t - \nabla f(w_t)\|_2 \leq \epsilon$. Then, for any $T \leq RC_2/(2\epsilon)$, we have*

$$
\|w_T\|_2 \leq R \quad \text{and} \quad \|w_T - w_*\|_2 \leq \exp\left(-\frac{C_1 T}{2C_2}\right)\|w_*\|_2 + \frac{T\epsilon}{C_2}.
$$

*Proof.* Let $v_0 := 0_m$ and $v_{t+1} := v_t - \nabla f(v_t)/C_2$ for $t \in \mathbb{N}_0$. Then, by Bubeck (2015, Theorem 3.10), we have

$$\|v_T - w_*\|_2 \leq \exp\left(-\frac{C_1 T}{2C_2}\right)\|w_*\|_2.$$

Moreover, by Bai et al. (2023, Lemma D.1), $\|w_T\|_2 \leq R$ and $\|w_T - v_T\|_2 \leq T\epsilon/C_2$. Therefore,

$$\|w_T - w_*\|_2 \leq \|v_T - w_*\|_2 + \|v_T - w_T\|_2 \leq \exp\left(-\frac{C_1 T}{2C_2}\right)\|w_*\|_2 + \frac{T\epsilon}{C_2},$$

as required. $\qquad\square$

Recall that we wish to solve the weighted least squares problem (4). For $w \in \mathbb{R}^D$, let

$$f(w) := \left\|\tilde{Y} - \tilde{\boldsymbol{X}}w\right\|_2^2, \tag{12}$$

and let $w_* := \arg\min_{w \in \mathbb{R}^D} f(w)$. Then the gradient of $f$ at $w$ is

$$\nabla f(w) = 2\left(\tilde{\boldsymbol{X}}^\mathsf{T}\tilde{\boldsymbol{X}}w - \tilde{\boldsymbol{X}}^\mathsf{T}\tilde{Y}\right) \in \mathbb{R}^D, \tag{13}$$

and we can write

$$1_{n+1}\nabla f(w)^\mathsf{T} = 2\begin{pmatrix} w^\mathsf{T} & -\frac{1_D^\mathsf{T}}{D} \\ \vdots & \vdots \\ w^\mathsf{T} & -\frac{1_D^\mathsf{T}}{D} \end{pmatrix}\begin{pmatrix} \tilde{\boldsymbol{X}}^\mathsf{T} & 0_D \\ 1_D\tilde{Y}^\mathsf{T} & 0_D \end{pmatrix}\begin{pmatrix} \tilde{\boldsymbol{X}} \\ 0_D^\mathsf{T} \end{pmatrix}. \tag{14}$$

The right-hand side of (14) is a product of three matrices, and so can be implemented by one linear attention layer with an appropriate input matrix. However, we can only approximate $\tilde{\boldsymbol{X}}$ and $\tilde{Y}$ by a transformer, and the approximation errors are controlled by Lemma B.8. We combine these ideas in the following lemma to show that one attention layer can approximate one step of gradient descent for minimising $f$, with an error in the gradient vector.

**Lemma B.10.** *Let $D, d_\mathrm{e}, d_\mathrm{ffn} \in \mathbb{N}$ be defined as in Lemma B.8. Let $\eta > 0$, $B := (2\eta) \vee 1$ and let $w \in \mathbb{R}^D$ be such that $\|w\|_2 \leq R$ for some $R > 0$. Define*

$$\boldsymbol{Z} := \begin{pmatrix} \boldsymbol{X} & Y & (\boldsymbol{X} - 1_n X_{n+1}^\mathsf{T})/h & n^{-1/2}\sqrt{K_h(\boldsymbol{X}, X_{n+1})} & \check{\boldsymbol{X}} & \check{Y} & 1_n w^\mathsf{T} & 1_n & 0_n \\ X_{n+1}^\mathsf{T} & 0 & 0_d^\mathsf{T} & n^{-1/2}h^{-d/2} & a^\mathsf{T} & b & w^\mathsf{T} & 1 & 1 \end{pmatrix} \in \mathbb{R}^{(n+1) \times d_\mathrm{e}},$$

*where $\check{\boldsymbol{X}} \in \mathbb{R}^{n \times D}$, $\check{Y} \in \mathbb{R}^n$, $a \in \mathbb{R}^D$ and $b \in \mathbb{R}$ satisfy*

$$\|\check{\boldsymbol{X}} - \tilde{\boldsymbol{X}}\|_\mathrm{max} \vee \|\check{Y} - \tilde{Y}\|_\infty \vee \|a\|_\infty \vee |b| \leq \xi, \tag{15}$$

*for some $\xi \in [0, 1]$. Then there exists $\mathrm{TF} \in \mathcal{T}(d_\mathrm{e}, d_\mathrm{ffn}, 1, B)$ such that*

$$\mathrm{TF}(\boldsymbol{Z}) = \begin{pmatrix} \boldsymbol{X} & Y & (\boldsymbol{X} - 1_n X_{n+1}^\mathsf{T})/h & n^{-1/2}\sqrt{K_h(\boldsymbol{X}, X_{n+1})} & \check{\boldsymbol{X}} & \check{Y} & 1_n(w - \eta g)^\mathsf{T} & 1_n & 0_n \\ X_{n+1}^\mathsf{T} & 0 & 0_d^\mathsf{T} & n^{-1/2}h^{-d/2} & a^\mathsf{T} & b & (w - \eta g)^\mathsf{T} & 1 & 1 \end{pmatrix},$$

*where $g \in \mathbb{R}^D$ satisfies*

$$\|g - \nabla f(w)\|_2 \leq 18D(M + 3)(R + 1)n\xi.$$

*Proof.* We can choose $\boldsymbol{Q}, \boldsymbol{K}, \boldsymbol{V} \in [-(2\eta \vee 1), (2\eta \vee 1)]^{d_\mathrm{e} \times d_\mathrm{e}}$ such that

$$\boldsymbol{ZQ} = -2\eta\begin{pmatrix} w^\mathsf{T} & -\frac{1_D^\mathsf{T}}{D} & 0_{d_\mathrm{e}-2D}^\mathsf{T} \\ \vdots & \vdots & \vdots \\ w^\mathsf{T} & -\frac{1_D^\mathsf{T}}{D} & 0_{d_\mathrm{e}-2D}^\mathsf{T} \end{pmatrix} \in \mathbb{R}^{(n+1) \times d_\mathrm{e}},$$

$$\boldsymbol{ZK} = \begin{pmatrix} \check{\boldsymbol{X}} & \check{Y}1_D^\mathsf{T} & \boldsymbol{0}_{n \times (d_\mathrm{e}-2D)} \\ a^\mathsf{T} & b1_D^\mathsf{T} & 0_{d_\mathrm{e}-2D}^\mathsf{T} \end{pmatrix} \in \mathbb{R}^{(n+1) \times d_\mathrm{e}},$$

$$\boldsymbol{ZV} = \begin{pmatrix} \boldsymbol{0}_{n \times (d_\mathrm{e}-D-2)} & \check{\boldsymbol{X}} & \boldsymbol{0}_{n \times 2} \\ 0_{d_\mathrm{e}-D-2}^\mathsf{T} & a^\mathsf{T} & 0_2^\mathsf{T} \end{pmatrix} \in \mathbb{R}^{(n+1) \times d_\mathrm{e}}.$$

Then, writing $g := 2\big(\check{\boldsymbol{X}}^\mathsf{T}\check{\boldsymbol{X}}w - \check{\boldsymbol{X}}^\mathsf{T}\check{Y} + (w^\mathsf{T}a - b)a\big)$, we have

$$\mathrm{Attn}_{\boldsymbol{Q},\boldsymbol{K},\boldsymbol{V}}(\boldsymbol{Z}) = \boldsymbol{Z} + \boldsymbol{Z}\boldsymbol{Q}(\boldsymbol{Z}\boldsymbol{K})^\mathsf{T}\boldsymbol{Z}\boldsymbol{V} = \boldsymbol{Z} - \eta\begin{pmatrix} 0_{d_e-D-2}^\mathsf{T} & g^\mathsf{T} & 0_2^\mathsf{T} \\ \vdots & \vdots & \vdots \\ 0_{d_e-D-2}^\mathsf{T} & g^\mathsf{T} & 0_2^\mathsf{T} \end{pmatrix}.$$

Moreover,

$$\begin{aligned}
\big\|g - \nabla f(w)\big\|_2 &= 2\big\|\check{\boldsymbol{X}}^\mathsf{T}\check{\boldsymbol{X}}w - \check{\boldsymbol{X}}^\mathsf{T}\check{Y} + (w^\mathsf{T}a - b)a - \tilde{\boldsymbol{X}}^\mathsf{T}\tilde{\boldsymbol{X}}w + \tilde{\boldsymbol{X}}^\mathsf{T}\tilde{Y}\big\|_2 \\
&\leq 2\Big\{\|\check{\boldsymbol{X}} - \tilde{\boldsymbol{X}}\|_{\mathrm{op}}\big(\|\check{\boldsymbol{X}}\|_{\mathrm{op}} + \|\tilde{\boldsymbol{X}}\|_{\mathrm{op}}\big)\|w\|_2 + \|\check{\boldsymbol{X}} - \tilde{\boldsymbol{X}}\|_{\mathrm{op}}\|\check{Y}\|_2 + \|\tilde{\boldsymbol{X}}\|_{\mathrm{op}}\|\check{Y} - \tilde{Y}\|_2 + \big(\|w\|_2\|a\|_2 + |b|\big)\|a\|_2\Big\} \\
&\leq 2\Big\{\xi(n+1)D(2+\xi)\|w\|_2 + \xi(n+1)(M+\xi+2)\sqrt{D} + \big(\|w\|_2\xi\sqrt{D} + \xi\big)\xi\sqrt{D}\Big\} \\
&\leq 18D(M+3)(R+1)n\xi,
\end{aligned}$$

where the second inequality uses (15) and the fact that $\|\tilde{\boldsymbol{X}}\|_{\mathrm{max}} \leq 1$ and $\|\tilde{Y}\|_\infty \leq M+1$. Finally, we take the FFN layer to be the identity map by setting all its parameters to zero, and this proves the claim. $\square$

**Proposition B.11.** *Let $p, D, d_e, d_{\mathrm{ffn}} \in \mathbb{N}$ and $B > 0$ be defined as in Lemma B.8, and let $\tilde{m}_n$ denote the $M$-truncated local polynomial estimator defined in Section A.1 with degree $p$, kernel $K(x) := (1 - \|x\|_1)_+^2$ and bandwidth $h := n^{-1/(2\alpha+d)}$. There exists $C > 0$, depending only on $d, \alpha, M, c_X, C_X$, such that if $L := \lceil C\log(en)\rceil$, then we can find a transformer $\mathrm{TF} \in \mathcal{T}(d_e, d_{\mathrm{ffn}}, L, B)$ satisfying*

$$\big|\mathrm{Read}_{M,d} \circ \mathrm{TF} \circ \mathrm{Embed}_{d_e}\big((X_i, Y_i)_{i\in[n]}, X_{n+1}\big) - \tilde{m}_n(X_{n+1})\big| \leq \frac{C}{n}$$

*with probability at least $1 - n^{C/(2\alpha+d)}\exp\big(-n^{2\alpha/(2\alpha+d)}/C\big)$.*

*Proof.* In this proof, $n_0, n_1, C_0, C_1, \ldots$ are positive quantities depending only on $d, \alpha, M, c_X, C_X$. By Lemma A.3, there exists $C_0 \geq 1$ such that if $n \geq C_0^{2\alpha+d}$, then the event

$$\mathcal{E}_0 := \Big\{C_0^{-1} \leq \lambda_{\min}\big(\tilde{\boldsymbol{X}}^\mathsf{T}\tilde{\boldsymbol{X}}\big) \leq \lambda_{\max}\big(\tilde{\boldsymbol{X}}^\mathsf{T}\tilde{\boldsymbol{X}}\big) \leq C_0\Big\}$$

has probability at least $1 - n^{C_0/(2\alpha+d)}\exp\big(-n^{2\alpha/(2\alpha+d)}/C_0\big)$. Let $w_* := \mathrm{argmin}_{w\in\mathbb{R}^D}\|\tilde{Y} - \tilde{\boldsymbol{X}}w\|_2^2$ be the minimiser of $f$, as defined in (12). Then, on the event $\mathcal{E}_0$, we have

$$\|w_*\|_2 = \big\|(\tilde{\boldsymbol{X}}^\mathsf{T}\tilde{\boldsymbol{X}})^{-1}\tilde{\boldsymbol{X}}^\mathsf{T}\tilde{Y}\big\|_2 \leq C_0\|\tilde{\boldsymbol{X}}^\mathsf{T}\tilde{Y}\|_2 \leq C_0(M+1)n\sqrt{D}, \tag{16}$$

where the final inequality uses the fact that $\|\tilde{\boldsymbol{X}}\|_{\mathrm{max}} \leq 1$ and $\|\tilde{Y}\|_\infty \leq M+1$. Moreover, on the event $\mathcal{E}_0$, the map $w \mapsto f(w) - \|w\|_2^2/C_0$ is convex and $\|\nabla f(w) - \nabla f(w')\|_2 \leq 2C_0\|w - w'\|_2$ for all $w, w' \in \mathbb{R}^D$, by (13). We define $R := 2C_0(M+1)n\sqrt{D}$ and work on the event $\mathcal{E}_0$ for the rest of the proof.

By Lemma B.8, there exists a transformer $\mathrm{TF}^{(0)} \in \mathcal{T}(d_e, d_{\mathrm{ffn}}, L^{(0)}, B)$ where $L^{(0)} := 7p(p+1)\lceil(\log n)/(p+1)\rceil + 4$ such that

$$\begin{aligned}
\boldsymbol{Z}^{(0)} &:= \mathrm{TF}^{(0)} \circ \mathrm{Embed}_{d_e}\big((X_i, Y_i)_{i\in[n]}, X_{n+1}\big) \\
&= \begin{pmatrix} \boldsymbol{X} & Y & (\boldsymbol{X} - 1_nX_{n+1}^\mathsf{T})/h & n^{-1/2}\sqrt{K_h(\boldsymbol{X}, X_{n+1})} & \check{\boldsymbol{X}} & \check{Y} & 1_nw_0^\mathsf{T} & 1_n & 0_n \\ X_{n+1}^\mathsf{T} & 0 & 0_d^\mathsf{T} & n^{-1/2}h^{-d/2} & a^\mathsf{T} & b & w_0^\mathsf{T} & 1 & 1 \end{pmatrix},
\end{aligned} \tag{17}$$

where $w_0 := 0_D$, and where $\check{\boldsymbol{X}} \in \mathbb{R}^{n\times D}, \check{Y} \in \mathbb{R}^n, a \in \mathbb{R}^D$ and $b \in \mathbb{R}$ satisfy

$$\|\check{\boldsymbol{X}} - \tilde{\boldsymbol{X}}\|_{\mathrm{max}} \vee \|\check{Y} - \tilde{Y}\|_\infty \vee \|a\|_\infty \vee |b| \leq 30p \cdot (2M+2)^{p+1} \cdot n^{-5} =: \xi. \tag{18}$$

Next, define $\epsilon := 18D(M+3)(R+1)n\xi$. Assume initially that $n \geq n_0$, where $n_0$ is large enough that $\xi \leq 1$. We claim that for every $T \in \big\{0, 1, \ldots, \lfloor RC_0/\epsilon\rfloor\big\}$, there exist $\mathrm{TF}^{(0)} \in \mathcal{T}(d_e, d_{\mathrm{ffn}}, L^{(0)}, B), \mathrm{TF}^{(1)}, \ldots, \mathrm{TF}^{(T)} \in \mathcal{T}(d_e, d_{\mathrm{ffn}}, 1, B)$,

$w_0, \ldots, w_T \in \mathbb{R}^D$ and $g_0, \ldots, g_{T-1} \in \mathbb{R}^D$ such that $w_0 = 0_D$, $w_{t+1} = w_t - g_t/(2C_0)$, $\|g_t - \nabla f(w_t)\|_2 \leq \epsilon$, $\|w_{t+1}\|_2 \leq R$ for all $t \in \{0, 1, \ldots, T-1\}$ and

$$\boldsymbol{Z}^{(t)} := \mathrm{TF}^{(t)} \circ \cdots \circ \mathrm{TF}^{(1)} \circ \mathrm{TF}^{(0)} \circ \mathrm{Embed}_{d_e}\big((X_i, Y_i)_{i \in [n]}, X_{n+1}\big)$$
$$= \begin{pmatrix} \boldsymbol{X} & Y & (\boldsymbol{X} - 1_n X_{n+1}^\mathsf{T})/h & n^{-1/2}\sqrt{K_h(\boldsymbol{X}, X_{n+1})} & \check{\boldsymbol{X}} & \check{Y} & 1_n w_t^\mathsf{T} & 1_n & 0_n \\ X_{n+1}^\mathsf{T} & 0 & 0_d^\mathsf{T} & n^{-1/2}h^{-d/2} & a^\mathsf{T} & b & w_t^\mathsf{T} & 1 & 1 \end{pmatrix},$$

for $t \in \{0, \ldots, T\}$. We argue by induction. The case $T = 0$ is verified by (17). Now suppose that the claim is true for some $T \leq \lfloor RC_0/\epsilon \rfloor - 1$. We may apply Lemma B.10 to deduce that there exists $\mathrm{TF}^{(T+1)} \in \mathcal{T}(d_e, d_{\mathrm{ffn}}, 1, B)$ such that

$$\boldsymbol{Z}^{(T+1)} := \mathrm{TF}^{(T+1)}\big(\boldsymbol{Z}^{(T)}\big)$$
$$= \begin{pmatrix} \boldsymbol{X} & Y & (\boldsymbol{X} - 1_n X_{n+1}^\mathsf{T})/h & n^{-1/2}\sqrt{K_h(\boldsymbol{X}, X_{n+1})} & \check{\boldsymbol{X}} & \check{Y} & 1_n(w_T - g_T/(2C_0))^\mathsf{T} & 1_n & 0_n \\ X_{n+1}^\mathsf{T} & 0 & 0_d^\mathsf{T} & n^{-1/2}h^{-d/2} & a^\mathsf{T} & b & (w_T - g_T/(2C_0))^\mathsf{T} & 1 & 1 \end{pmatrix},$$

where by (18), the inductive hypothesis that $\|w_T\|_2 \leq R$ and Lemma B.10, we have

$$\|g_T - \nabla f(w_T)\|_2 \leq 18D(M+3)(R+1)n\xi = \epsilon.$$

Moreover, by Lemma B.9 (with $C_1 := 2/C_0$ and $C_2 := 2C_0$ therein), and since $T + 1 \leq \lfloor RC_0/\epsilon \rfloor$, we have $\|w_{T+1}\|_2 \leq R$. This proves the claim by induction. Now, let $T := \lceil 4C_0^2 \log n \rceil$ and further assume that $n \geq n_1$, where $n_1$ is large enough that $T \leq \lfloor RC_0/\epsilon \rfloor$ (note that $RC_0/\epsilon$ is quartic in $n$ whereas $T$ is logarithmic). Then by Lemma B.9 again and (16), we deduce that

$$\|w_T - w_*\|_2 \leq \exp\left(-\frac{T}{2C_0^2}\right) \cdot C_0(M+1)n\sqrt{D} + \frac{T\epsilon}{2C_0} \leq \frac{C_3}{n}.$$

Let $\mathrm{TF}^{(T+1)} \in \mathcal{T}(d_e, d_{\mathrm{ffn}}, 1, B)$ be a transformer block such that for any $\boldsymbol{Z} \in \mathbb{R}^{(n+1) \times d_e}$, the $(d+1)$th column of $\mathrm{TF}^{(T+1)}(\boldsymbol{Z})$ is equal to the sum of the $(d+1)$th column and the $(2d+D+4)$th column of $\boldsymbol{Z}$ (this can be done by using only the FFN layer). Further, define $\mathrm{TF} := \mathrm{TF}^{(T+1)} \circ \cdots \circ \mathrm{TF}^{(0)} \in \mathcal{T}(d_e, d_{\mathrm{ffn}}, L^{(0)} + T + 1, B)$. Then, writing $w_{T,1}$ and $w_{*,1}$ for the first entries of $w_T$ and $w_*$ respectively, we have for $n \geq n_0 \vee n_1$ that

$$\Big|\mathrm{Read}_{M,d} \circ \mathrm{TF} \circ \mathrm{Embed}_{d_e}\big((X_i, Y_i)_{i \in [n]}, X_{n+1}\big) - \tilde{m}_n(X_{n+1})\Big|$$
$$= \big|(-M) \vee w_{T,1} \wedge M - (-M) \vee w_{*,1} \wedge M\big| \leq |w_{T,1} - w_{*,1}| \leq \frac{C_3}{n}, \tag{19}$$

with probability at least $1 - n^{C_0/(2\alpha+d)} \exp\big(-n^{2\alpha/(2\alpha+d)}/C_0\big)$. On the other hand, the left-hand side of (19) is bounded by $2M$, so by replacing $C_3$ with $C_4$ if necessary, the conclusion holds for all $n \in \mathbb{N}$. Finally, note that $L^{(0)} + T + 1 \leq \lceil C_5 \log(en) \rceil$, so the lemma holds with $C := C_0 \vee C_4 \vee C_5$. □

## C. Covering Numbers of Transformers

In this section, we provide bounds for the covering numbers of transformers with linear attention. Similar results on the covering numbers of transformers with ReLU attention can be found in (Bai et al., 2023).

**Lemma C.1.** *Let $n, d_e \in \mathbb{N}$ and $B, \delta, R > 0$. Suppose that $\boldsymbol{Q}, \boldsymbol{K}, \boldsymbol{V}, \boldsymbol{Q}', \boldsymbol{K}', \boldsymbol{V}' \in \mathbb{R}^{d_e \times d_e}$ satisfy $\|\boldsymbol{Q}\|_{\max} \vee \|\boldsymbol{K}\|_{\max} \vee \|\boldsymbol{V}\|_{\max} \vee \|\boldsymbol{Q}'\|_{\max} \vee \|\boldsymbol{K}'\|_{\max} \vee \|\boldsymbol{V}'\|_{\max} \leq B$ and $\|\boldsymbol{Q} - \boldsymbol{Q}'\|_{\max} \vee \|\boldsymbol{K} - \boldsymbol{K}'\|_{\max} \vee \|\boldsymbol{V} - \boldsymbol{V}'\|_{\max} \leq \delta$. Let $\boldsymbol{Z} \in \mathbb{R}^{(n+1) \times d_e}$ be such that $\|\boldsymbol{Z}\|_{\max} \leq R$. Then the linear attention function from Definition 2.1 satisfies*

$$\big\|\mathrm{Attn}_{\boldsymbol{Q},\boldsymbol{K},\boldsymbol{V}}(\boldsymbol{Z}) - \mathrm{Attn}_{\boldsymbol{Q}',\boldsymbol{K}',\boldsymbol{V}'}(\boldsymbol{Z})\big\|_{\max} \leq 3B^2 R^3 (n+1)^{3/2} d_e^{9/2} \delta.$$

*Proof of Lemma C.1.* From Definition 2.1,

$$
\begin{aligned}
\big\|\mathrm{Attn}_{\boldsymbol{Q},\boldsymbol{K},\boldsymbol{V}}(\boldsymbol{Z}) - \mathrm{Attn}_{\boldsymbol{Q}',\boldsymbol{K}',\boldsymbol{V}'}(\boldsymbol{Z})\big\|_{\max} &= \big\|\boldsymbol{Z}\boldsymbol{Q}(\boldsymbol{Z}\boldsymbol{K})^{\mathsf{T}}\boldsymbol{Z}\boldsymbol{V} - \boldsymbol{Z}\boldsymbol{Q}'(\boldsymbol{Z}\boldsymbol{K}')^{\mathsf{T}}\boldsymbol{Z}\boldsymbol{V}'\big\|_{\max} \\
&\leq \big\|\boldsymbol{Z}(\boldsymbol{Q}-\boldsymbol{Q}')(\boldsymbol{Z}\boldsymbol{K})^{\mathsf{T}}\boldsymbol{Z}\boldsymbol{V}\big\|_{\mathrm{op}} + \big\|\boldsymbol{Z}\boldsymbol{Q}'\{\boldsymbol{Z}(\boldsymbol{K}-\boldsymbol{K}')\}^{\mathsf{T}}\boldsymbol{Z}\boldsymbol{V}\big\|_{\mathrm{op}} \\
&\qquad\qquad\qquad + \big\|\boldsymbol{Z}\boldsymbol{Q}'(\boldsymbol{Z}\boldsymbol{K}')^{\mathsf{T}}\boldsymbol{Z}(\boldsymbol{V}-\boldsymbol{V}')\big\|_{\mathrm{op}} \\
&\leq \|\boldsymbol{Z}\|_{\mathrm{op}}^3\|\boldsymbol{Q}-\boldsymbol{Q}'\|_{\mathrm{op}}\|\boldsymbol{K}\|_{\mathrm{op}}\|\boldsymbol{V}\|_{\mathrm{op}} + \|\boldsymbol{Z}\|_{\mathrm{op}}^3\|\boldsymbol{Q}'\|_{\mathrm{op}}\|\boldsymbol{K}-\boldsymbol{K}'\|_{\mathrm{op}}\|\boldsymbol{V}\|_{\mathrm{op}} \\
&\qquad\qquad\qquad + \|\boldsymbol{Z}\|_{\mathrm{op}}^3\|\boldsymbol{Q}'\|_{\mathrm{op}}\|\boldsymbol{K}'\|_{\mathrm{op}}\|\boldsymbol{V}-\boldsymbol{V}'\|_{\mathrm{op}} \\
&\leq \{(n+1)d_{\mathrm{e}}\}^{3/2}d_{\mathrm{e}}{}^3\|\boldsymbol{Z}\|_{\max}^3\|\boldsymbol{Q}-\boldsymbol{Q}'\|_{\max}\|\boldsymbol{K}\|_{\max}\|\boldsymbol{V}\|_{\max} \\
&\qquad + \{(n+1)d_{\mathrm{e}}\}^{3/2}d_{\mathrm{e}}{}^3\|\boldsymbol{Z}\|_{\max}^3\|\boldsymbol{Q}'\|_{\max}\|\boldsymbol{K}-\boldsymbol{K}'\|_{\max}\|\boldsymbol{V}\|_{\max} \\
&\qquad + \{(n+1)d_{\mathrm{e}}\}^{3/2}d_{\mathrm{e}}{}^3\|\boldsymbol{Z}\|_{\max}^3\|\boldsymbol{Q}'\|_{\max}\|\boldsymbol{K}'\|_{\max}\|\boldsymbol{V}-\boldsymbol{V}'\|_{\max} \\
&\leq 3B^2 R^3(n+1)^{3/2}d_{\mathrm{e}}{}^{9/2}\delta,
\end{aligned}
$$

as required. $\square$

**Lemma C.2.** *Let $n, d_{\mathrm{e}}, d_{\mathrm{ffn}} \in \mathbb{N}$ and $B, \delta, R > 0$. Suppose that $\boldsymbol{W}_1, \boldsymbol{W}_1' \in \mathbb{R}^{d_{\mathrm{ffn}} \times d_{\mathrm{e}}}$, $\boldsymbol{W}_2, \boldsymbol{W}_2' \in \mathbb{R}^{d_{\mathrm{e}} \times d_{\mathrm{ffn}}}$, $b_1, b_1' \in \mathbb{R}^{d_{\mathrm{ffn}}}$ and $b_2, b_2' \in \mathbb{R}^{d_{\mathrm{e}}}$ satisfy $\|\boldsymbol{W}_1\|_{\max} \vee \|\boldsymbol{W}_2\|_{\max} \vee \|b_1\|_{\infty} \vee \|b_2\|_{\infty} \vee \|\boldsymbol{W}_1'\|_{\max} \vee \|\boldsymbol{W}_2'\|_{\max} \vee \|b_1'\|_{\infty} \vee \|b_2'\|_{\infty} \leq B$ and $\|\boldsymbol{W}_1 - \boldsymbol{W}_1'\|_{\max} \vee \|\boldsymbol{W}_2 - \boldsymbol{W}_2'\|_{\max} \vee \|b_1 - b_1'\|_{\infty} \vee \|b_2 - b_2'\|_{\infty} \leq \delta$. Let $\boldsymbol{Z} \in \mathbb{R}^{(n+1) \times d_{\mathrm{e}}}$ be such that $\|\boldsymbol{Z}\|_{\max} \leq R$. Then the feed-forward network function from Definition 2.2 satisfies*

$$
\big\|\mathrm{FFN}_{\boldsymbol{W}_1,\boldsymbol{W}_2,b_1,b_2}(\boldsymbol{Z}) - \mathrm{FFN}_{\boldsymbol{W}_1',\boldsymbol{W}_2',b_1',b_2'}(\boldsymbol{Z})\big\|_{\max} \leq 2(B+1)(R+1)(n+1)d_{\mathrm{e}}{}^{3/2}d_{\mathrm{ffn}}{}^{3/2}\delta.
$$

*Proof of Lemma C.2.* From Definition 2.2, since ReLU is 1-Lipschitz from $\|\cdot\|_{\max}$ to $\|\cdot\|_{\max}$,

$$
\begin{aligned}
&\big\|\mathrm{FFN}_{\boldsymbol{W}_1,\boldsymbol{W}_2,b_1,b_2}(\boldsymbol{Z}) - \mathrm{FFN}_{\boldsymbol{W}_1',\boldsymbol{W}_2',b_1',b_2'}(\boldsymbol{Z})\big\|_{\max} \\
&= \big\|\mathrm{ReLU}(\boldsymbol{Z}\boldsymbol{W}_1^{\mathsf{T}} + 1_{n+1}b_1^{\mathsf{T}})\boldsymbol{W}_2^{\mathsf{T}} + 1_{n+1}b_2^{\mathsf{T}} - \mathrm{ReLU}(\boldsymbol{Z}\boldsymbol{W}_1'^{\mathsf{T}} + 1_{n+1}b_1'^{\mathsf{T}})\boldsymbol{W}_2'^{\mathsf{T}} - 1_{n+1}b_2'^{\mathsf{T}}\big\|_{\max} \\
&\leq \big\|\mathrm{ReLU}(\boldsymbol{Z}\boldsymbol{W}_1^{\mathsf{T}} + 1_{n+1}b_1^{\mathsf{T}})\boldsymbol{W}_2^{\mathsf{T}} - \mathrm{ReLU}(\boldsymbol{Z}\boldsymbol{W}_1'^{\mathsf{T}} + 1_{n+1}b_1'^{\mathsf{T}})\boldsymbol{W}_2^{\mathsf{T}}\big\|_{\max} \\
&\quad + \big\|\mathrm{ReLU}(\boldsymbol{Z}\boldsymbol{W}_1'^{\mathsf{T}} + 1_{n+1}b_1'^{\mathsf{T}})\boldsymbol{W}_2^{\mathsf{T}} - \mathrm{ReLU}(\boldsymbol{Z}\boldsymbol{W}_1'^{\mathsf{T}} + 1_{n+1}b_1'^{\mathsf{T}})\boldsymbol{W}_2'^{\mathsf{T}}\big\|_{\max} + \|b_2 - b_2'\|_{\infty} \\
&\leq \|\boldsymbol{W}_2^{\mathsf{T}}\|_{\mathrm{op}}\big\|\mathrm{ReLU}(\boldsymbol{Z}\boldsymbol{W}_1^{\mathsf{T}} + 1_{n+1}b_1) - \mathrm{ReLU}(\boldsymbol{Z}\boldsymbol{W}_1'^{\mathsf{T}} + 1_{n+1}b_1')\big\|_{\mathrm{op}} \\
&\quad + \big\|\mathrm{ReLU}(\boldsymbol{Z}\boldsymbol{W}_1'^{\mathsf{T}} + 1_{n+1}b_1'^{\mathsf{T}})\big\|_{\mathrm{op}}\|\boldsymbol{W}_2^{\mathsf{T}} - \boldsymbol{W}_2'^{\mathsf{T}}\|_{\mathrm{op}} + \|b_2 - b_2'\|_{\infty} \\
&\leq \sqrt{d_{\mathrm{ffn}}d_{\mathrm{e}}}B\sqrt{(n+1)d_{\mathrm{ffn}}}\big\|\boldsymbol{Z}\boldsymbol{W}_1^{\mathsf{T}} + 1_{n+1}b_1^{\mathsf{T}} - \boldsymbol{Z}\boldsymbol{W}_1'^{\mathsf{T}} - 1_{n+1}b_1'^{\mathsf{T}}\big\|_{\max} \\
&\quad + \sqrt{(n+1)d_{\mathrm{ffn}}}\big\|\boldsymbol{Z}\boldsymbol{W}_1'^{\mathsf{T}} + 1_{n+1}b_1'^{\mathsf{T}}\big\|_{\max}\sqrt{d_{\mathrm{ffn}}d_{\mathrm{e}}}\,\delta + \delta \\
&\leq Bd_{\mathrm{ffn}}\sqrt{(n+1)d_{\mathrm{e}}}\big(\big\|\boldsymbol{Z}\boldsymbol{W}_1^{\mathsf{T}} - \boldsymbol{Z}\boldsymbol{W}_1'^{\mathsf{T}}\big\|_{\max} + \|b_1 - b_1'\|_{\infty}\big) + \delta d_{\mathrm{ffn}}\sqrt{(n+1)d_{\mathrm{e}}}\big(\big\|\boldsymbol{Z}\boldsymbol{W}_1'^{\mathsf{T}}\big\|_{\max} + \|b_1'\|_{\infty}\big) + \delta \\
&\leq Bd_{\mathrm{ffn}}\sqrt{(n+1)d_{\mathrm{e}}}\big(R\delta d_{\mathrm{e}}\sqrt{(n+1)d_{\mathrm{ffn}}} + \delta\big) + \delta d_{\mathrm{ffn}}\sqrt{(n+1)d_{\mathrm{e}}}\big(RBd_{\mathrm{e}}\sqrt{(n+1)d_{\mathrm{ffn}}} + B\big) + \delta \\
&= 2BR\delta(n+1)d_{\mathrm{e}}{}^{3/2}d_{\mathrm{ffn}}{}^{3/2} + 2B\delta\sqrt{(n+1)d_{\mathrm{e}}}d_{\mathrm{ffn}} + \delta \\
&\leq 2(B+1)(R+1)(n+1)d_{\mathrm{e}}{}^{3/2}d_{\mathrm{ffn}}{}^{3/2}\delta,
\end{aligned}
$$

as required. $\square$

**Lemma C.3.** *Let $n, d, d_{\mathrm{e}}, d_{\mathrm{ffn}}, L \in \mathbb{N}$ with $d_{\mathrm{e}} \geq d + 2$ and take $B, \delta, R > 0$. For $\ell \in [L]$, let $\boldsymbol{\theta}^{(\ell)}, \boldsymbol{\theta}'^{(\ell)} \in \mathbb{R}^{d_{\mathrm{e}} \times d_{\mathrm{e}}} \times \mathbb{R}^{d_{\mathrm{e}} \times d_{\mathrm{e}}} \times \mathbb{R}^{d_{\mathrm{e}} \times d_{\mathrm{e}}} \times \mathbb{R}^{d_{\mathrm{e}} \times d_{\mathrm{ffn}}} \times \mathbb{R}^{d_{\mathrm{ffn}} \times d_{\mathrm{e}}} \times \mathbb{R}^{d_{\mathrm{ffn}}} \times \mathbb{R}^{d_{\mathrm{e}}}$. Let $\boldsymbol{\theta} := (\boldsymbol{\theta}^{(\ell)})_{\ell \in [L]}$ and $\boldsymbol{\theta}' := (\boldsymbol{\theta}'^{(\ell)})_{\ell \in [L]}$. Suppose that every entry of each parameter in $\boldsymbol{\theta}$ and $\boldsymbol{\theta}'$ is bounded by $B$, and that every entry of each parameter in $\boldsymbol{\theta} - \boldsymbol{\theta}'$ is bounded by $\delta$. Let $\boldsymbol{Z} \in \mathbb{R}^{(n+1) \times d_{\mathrm{e}}}$ be such that $\|\boldsymbol{Z}\|_{\max} \leq R$. Then the transformer function from Definition 2.3 satisfies*

$$
\big\|\mathrm{TF}_{\boldsymbol{\theta}}(\boldsymbol{Z}) - \mathrm{TF}_{\boldsymbol{\theta}'}(\boldsymbol{Z})\big\|_{\max} \leq 6^L(B+1)^{3L}(R+1)^{4L}(n+1)^{5L/2}d_{\mathrm{e}}{}^{6L}d_{\mathrm{ffn}}{}^{3L/2}\delta.
$$

*Further, if $(x_i, y_i) \in [0,1]^d \times [-M-1, M+1]$ for each $i \in [n]$ and $x_{n+1} \in [0,1]^d$, then*

$$
\begin{aligned}
\big|\mathrm{Read}_{M,d} \circ \mathrm{TF}_{\boldsymbol{\theta}} &\circ \mathrm{Embed}_{d_{\mathrm{e}}}\big((x_i, y_i)_{i \in [n]}, x_{n+1}\big) - \mathrm{Read}_{M,d} \circ \mathrm{TF}_{\boldsymbol{\theta}'} \circ \mathrm{Embed}_{d_{\mathrm{e}}}\big((x_i, y_i)_{i \in [n]}, x_{n+1}\big)\big| \\
&\leq 6^L(B+1)^{3L}(M+2)^{4L}(n+1)^{5L/2}d_{\mathrm{e}}{}^{6L}d_{\mathrm{ffn}}{}^{3L/2}\delta.
\end{aligned}
$$

*Proof of Lemma C.3.* By Lemmas C.1 and C.2, for $\ell \in [L]$, by Lipschitz composition,

$$\left\| \mathrm{Block}_{\boldsymbol{\theta}^{(\ell)}}(\boldsymbol{Z}) - \mathrm{Block}_{\boldsymbol{\theta}'^{(\ell)}}(\boldsymbol{Z}) \right\|_{\max} \leq 6(B+1)^3(R+1)^4(n+1)^{5/2}d_{\mathrm{e}}{}^6 d_{\mathrm{ffn}}{}^{3/2}\delta.$$

Therefore, by composition over the $L$ layers,

$$\left\| \mathrm{TF}_{\boldsymbol{\theta}}(\boldsymbol{Z}) - \mathrm{TF}_{\boldsymbol{\theta}'}(\boldsymbol{Z}) \right\|_{\max} \leq 6^L(B+1)^{3L}(R+1)^{4L}(n+1)^{5L/2}d_{\mathrm{e}}{}^{6L} d_{\mathrm{ffn}}{}^{3L/2}\delta.$$

The second result follows from the first, on observing that the output of $\mathrm{Embed}_{d_{\mathrm{e}}}$ is bounded by $M+1$, and $\mathrm{Read}_{M,d}$ is 1-Lipschitz from $\|\cdot\|_{\max}$ to $|\cdot|$. $\qquad\square$

**Lemma C.4.** *Let $n, d, d_{\mathrm{e}}, d_{\mathrm{ffn}}, L, \Gamma \in \mathbb{N}$ with $d_{\mathrm{e}} \geq d+2$ and take $B, M > 0$ and $\delta \in [0, M]$. Define the $\delta$-covering number in $\|\cdot\|_\infty$-norm of the class $\mathcal{F}(d_{\mathrm{e}}, d_{\mathrm{ffn}}, L, B, M)$ of functions from $\left([0,1]^d \times \mathbb{R}\right)^n \times [0,1]^d$ to $\mathbb{R}$ by*

$$N\big(\mathcal{F}(d_{\mathrm{e}}, d_{\mathrm{ffn}}, L, B, M), \delta, \|\cdot\|_\infty\big)$$
$$:= \inf\left\{ |\mathcal{F}'| : \mathcal{F}' \subseteq \mathcal{F}(d_{\mathrm{e}}, d_{\mathrm{ffn}}, L, B, M), \sup_{f \in \mathcal{F}(d_{\mathrm{e}}, d_{\mathrm{ffn}}, L, B, M)} \inf_{f' \in \mathcal{F}'} \|f - f'\|_\infty \leq \delta \right\}.$$

*Then the log-covering number (entropy) satisfies*

$$\log N\big(\mathcal{F}(d_{\mathrm{e}}, d_{\mathrm{ffn}}, L, B, M), \delta, \|\cdot\|_\infty\big) \leq 24 L d_{\mathrm{e}}(d_{\mathrm{e}} + d_{\mathrm{ffn}}) \log\big((B+1)^L(M+2)^{2L}(n+1)^L d_{\mathrm{e}}{}^L d_{\mathrm{ffn}}{}^L/\delta\big).$$

*Proof of Lemma C.4.* Define $\Theta := \big([-B,B]^{d_{\mathrm{e}} \times d_{\mathrm{e}}} \times [-B,B]^{d_{\mathrm{e}} \times d_{\mathrm{e}}} \times [-B,B]^{d_{\mathrm{e}} \times d_{\mathrm{e}}} \times [-B,B]^{d_{\mathrm{e}} \times d_{\mathrm{ffn}}} \times [-B,B]^{d_{\mathrm{ffn}} \times d_{\mathrm{e}}} \times [-B,B]^{d_{\mathrm{ffn}}} \times [-B,B]^{d_{\mathrm{e}}}\big)^L$ and take $C := 6^L(B+1)^{3L}(M+2)^{4L}(n+1)^{5L/2}d_{\mathrm{e}}{}^{6L}d_{\mathrm{ffn}}{}^{3L/2}$. The dimension of $\Theta$ is $L(3d_{\mathrm{e}}{}^2 + 2d_{\mathrm{e}}d_{\mathrm{ffn}} + d_{\mathrm{e}} + d_{\mathrm{ffn}}) \leq 4L d_{\mathrm{e}}(d_{\mathrm{e}} + d_{\mathrm{ffn}})$. Let $\Theta'$ be a $\delta/C$-cover of $(\Theta, \|\cdot\|_\infty)$ of cardinality at most $\{2(B+1)C/\delta\}^{4L d_{\mathrm{e}}(d_{\mathrm{e}}+d_{\mathrm{ffn}})}$. Take $\mathcal{F}' := \big\{\mathrm{Read}_{M,d} \circ \mathrm{TF}_{\boldsymbol{\theta}} \circ \mathrm{Embed}_{d_{\mathrm{e}}} : \boldsymbol{\theta} \in \Theta'\big\}$ so by Lemma C.3, $\mathcal{F}'$ is a $\delta$-cover of $\mathcal{F}(d_{\mathrm{e}}, d_{\mathrm{ffn}}, L, B, M)$ in $\|\cdot\|_\infty$-norm with

$$\log|\mathcal{F}'| \leq \log\big(\{2(B+1)C/\delta\}^{4L d_{\mathrm{e}}(d_{\mathrm{e}}+d_{\mathrm{ffn}})}\big) = 4L d_{\mathrm{e}}(d_{\mathrm{e}} + d_{\mathrm{ffn}})\log\{2(B+1)C/\delta\}$$
$$= 4L d_{\mathrm{e}}(d_{\mathrm{e}} + d_{\mathrm{ffn}})\log\big(2 \cdot 6^L(B+1)^{3L+1}(M+2)^{4L}(n+1)^{5L/2}d_{\mathrm{e}}{}^{6L}d_{\mathrm{ffn}}{}^{3L/2}/\delta\big)$$
$$\leq 4L d_{\mathrm{e}}(d_{\mathrm{e}} + d_{\mathrm{ffn}})\log\big((B+1)^{4L}(M+2)^{5L}(n+1)^{11L/2}d_{\mathrm{e}}{}^{6L}d_{\mathrm{ffn}}{}^{3L/2}/\delta\big)$$
$$\leq 24L d_{\mathrm{e}}(d_{\mathrm{e}} + d_{\mathrm{ffn}})\log\big((B+1)^L(M+2)^{2L}(n+1)^L d_{\mathrm{e}}{}^L d_{\mathrm{ffn}}{}^L/\delta\big),$$

as required. $\qquad\square$

# D. Risk Decomposition

**Theorem D.1.** *Let $n, d, d_{\mathrm{e}}, d_{\mathrm{ffn}}, L, \Gamma \in \mathbb{N}$ with $d_{\mathrm{e}} \geq d+2$ and take $B > 0$. Take $\hat{f}_\Gamma$ as in Definition 2.4. There exists a universal constant $C > 0$ such that*

$$\mathbb{E}\big\{R(\hat{f}_\Gamma)\big\} \leq \inf_{f \in \mathcal{F}(d_{\mathrm{e}}, d_{\mathrm{ffn}}, L, B, M)} 2R(f) - \sigma^2 + C(M+1)^5 L d_{\mathrm{e}}(d_{\mathrm{e}} + d_{\mathrm{ffn}})\frac{L\log\big\{(B+1)nd_{\mathrm{e}}d_{\mathrm{ffn}}\big\} + \log\Gamma}{\Gamma}.$$

*Proof of Theorem D.1.* Throughout the proof, we write $C_1, C_2, \ldots$ to denote universal positive constants. Let $f^\star \in \mathcal{F}(d_{\mathrm{e}}, d_{\mathrm{ffn}}, L, B, M)$ satisfy $R(f^\star) \leq \inf_{f \in \mathcal{F}(d_{\mathrm{e}}, d_{\mathrm{ffn}}, L, B, M)} R(f) + 1/\Gamma$. By the Doob–Dynkin lemma (e.g., Kallenberg, 2021, Lemma 1.14), let $f_0 : \left([0,1]^d \times \mathbb{R}\right)^n \times [0,1]^d \to \mathbb{R}$ be a Borel-measurable function satisfying $f_0\big(\mathcal{D}_n, X_{n+1}\big) = \mathbb{E}(Y_{n+1} \mid \mathcal{D}_n, X_{n+1})$ almost surely, so that $R(f_0) \geq \sigma^2$. Since $\hat{R}_\Gamma(\hat{f}_\Gamma) \leq \hat{R}_\Gamma(f^\star)$ and as $\mathbb{E}\big\{\hat{R}_\Gamma(f)\big\} = R(f)$ for all bounded, measurable functions $f$, we have

$$\mathbb{E}\big\{R(\hat{f}_\Gamma)\big\} = R(f_0) + \mathbb{E}\big\{R(\hat{f}_\Gamma) - R(f_0) - 2\hat{R}_\Gamma(\hat{f}_\Gamma) + 2\hat{R}_\Gamma(f_0)\big\} + 2\,\mathbb{E}\big\{\hat{R}_\Gamma(\hat{f}_\Gamma) - \hat{R}_\Gamma(f_0)\big\}$$
$$\leq R(f_0) + \mathbb{E}\big\{R(\hat{f}_\Gamma) - R(f_0) - 2\hat{R}_\Gamma(\hat{f}_\Gamma) + 2\hat{R}_\Gamma(f_0)\big\} + 2\big\{R(f^\star) - R(f_0)\big\}$$
$$\leq \mathbb{E}\big\{R(\hat{f}_\Gamma) - R(f_0) - 2\hat{R}_\Gamma(\hat{f}_\Gamma) + 2\hat{R}_\Gamma(f_0)\big\} + \inf_{f \in \mathcal{F}(d_{\mathrm{e}}, d_{\mathrm{ffn}}, L, B, M)} 2R(f) + \frac{2}{\Gamma} - \sigma^2. \qquad (20)$$

To bound the expectation, we apply the bound given by Györfi et al. (2002, Theorem 11.4) with $\epsilon = 1/2$ and $\alpha = \beta = t/2$ therein, noting that $|Y_{n+1}^{(\gamma)}| \leq M + 1$ for each $\gamma \in [\Gamma]$ and $\sup_{f \in \mathcal{F}(d_e, d_{\mathrm{ffn}}, L, B, M)} \|f\|_\infty \leq M$. As $L_1$-covering numbers are bounded by $L_\infty$-covering numbers, there exists a universal constant $C_1 > 0$ such that for all $t > 0$,

$$\mathbb{P}\Big(R(\hat{f}_\Gamma) - R(f_0) - 2\hat{R}_\Gamma(\hat{f}_\Gamma) + 2\hat{R}_\Gamma(f_0) \geq t\Big)$$
$$\leq C_1 N\Big(\mathcal{F}(d_e, d_{\mathrm{ffn}}, L, B, M), \frac{t}{C_1(M+1)}, \|\cdot\|_\infty\Big) \exp\Big(-\frac{\Gamma t}{C_1(M+1)^4}\Big).$$

Hence, by Lemma C.4, taking $t := C_1 s (M+1)^4 / \Gamma$ for $s \in [1, \Gamma]$,

$$\mathbb{P}\Big(R(\hat{f}_\Gamma) - R(f_0) - 2\hat{R}_\Gamma(\hat{f}_\Gamma) + 2\hat{R}_\Gamma(f_0) \geq \frac{C_1 s(M+1)^4}{\Gamma}\Big)$$
$$\leq C_1 N\big(\mathcal{F}(d_e, d_{\mathrm{ffn}}, L, B, M), s/\Gamma, \|\cdot\|_\infty\big) e^{-s}$$
$$\leq C_1 \exp\Big(24 L d_e(d_e + d_{\mathrm{ffn}}) \log\big\{(B+1)^L (M+2)^{2L} (n+1)^L d_e{}^L d_{\mathrm{ffn}}{}^L \Gamma\big\}\Big) e^{-s}. \tag{21}$$

In fact, taking $C_1 \geq 12$ without loss of generality, (21) holds for all $s \in [1, \infty)$ because $\|\hat{f}_\Gamma\|_\infty \leq M$ and $\|f_0\|_\infty \leq M + 1$, so $R(\hat{f}_\Gamma) \leq 4(M+1)^2$ and $\hat{R}_\Gamma(f_0) \leq 4(M+1)^2$. Therefore, for all $u > 0$, with probability at least $1 - C_2 e^{-u}$, as $d_e \geq 3 > e$,

$$R(\hat{f}_\Gamma) - R(f_0) - 2\hat{R}_\Gamma(\hat{f}_\Gamma) + 2\hat{R}_\Gamma(f_0)$$
$$\leq \frac{C_2(M+1)^4}{\Gamma}\Big\{u + L d_e(d_e + d_{\mathrm{ffn}}) \log\big\{(B+1)^L (M+2)^{2L} (n+1)^L d_e{}^L d_{\mathrm{ffn}}{}^L \Gamma\big\}\Big\}.$$

Integrating the tail probability, we obtain

$$\mathbb{E}\big\{R(\hat{f}_\Gamma) - R(f_0) - 2\hat{R}_\Gamma(\hat{f}_\Gamma) + 2\hat{R}_\Gamma(f_0)\big\}$$
$$\leq \frac{C_3(M+1)^4 L d_e(d_e + d_{\mathrm{ffn}}) \log\big\{(B+1)^L (M+2)^{2L} (n+1)^L d_e{}^L d_{\mathrm{ffn}}{}^L \Gamma\big\}}{\Gamma}$$
$$\leq C_4(M+1)^5 L d_e(d_e + d_{\mathrm{ffn}}) \frac{L \log\big\{(B+1) n d_e d_{\mathrm{ffn}}\big\} + \log \Gamma}{\Gamma}. \tag{22}$$

The result follows from (20) and (22). □

# E. Proofs of Main Results

*Proof of Theorem 3.1.* Throughout the proof, $C_1', C_2' > 0$ are quantities depending only on $d$, $\alpha$, $M$, $c_X$ and $C_X$. By Proposition B.11, since $n^{(p+1)/(2\alpha+d)} \leq n^2$, there exists a transformer $\mathrm{TF} \in \mathcal{T}(d_e, d_{\mathrm{ffn}}, L, B)$ such that if $f_{\mathrm{TF}} := \mathrm{Read}_{M,d} \circ \mathrm{TF} \circ \mathrm{Embed}_{d_e} \in \mathcal{F}(d_e, d_{\mathrm{ffn}}, L, B, M)$, then

$$\big|R(f_{\mathrm{TF}}) - R(f_{\mathrm{LocPol}})\big| = \Big|\mathbb{E}\big(\{Y_{n+1} - f_{\mathrm{TF}}(\mathcal{D}_n, X_{n+1})\}^2\big) - \mathbb{E}\big(\{Y_{n+1} - f_{\mathrm{LocPol}}(\mathcal{D}_n, X_{n+1})\}^2\big)\Big|$$
$$= \Big|\mathbb{E}\Big(\{f_{\mathrm{LocPol}}(\mathcal{D}_n, X_{n+1}) - f_{\mathrm{TF}}(\mathcal{D}_n, X_{n+1})\}\{2Y_{n+1} - f_{\mathrm{TF}}(\mathcal{D}_n, X_{n+1}) - f_{\mathrm{LocPol}}(\mathcal{D}_n, X_{n+1})\}\Big)\Big|$$
$$\leq (4M + 2) \mathbb{E}\Big(\big|f_{\mathrm{TF}}(\mathcal{D}_n, X_{n+1}) - f_{\mathrm{LocPol}}(\mathcal{D}_n, X_{n+1})\big|\Big)$$
$$\leq (4M + 2)\Big\{C_1'/n + 2M n^{C_1'/(2\alpha+d)} \exp\big(-n^{2\alpha/(2\alpha+d)}/C_1'\big)\Big\} \leq \frac{C_2'}{n}.$$

□

*Proof of Theorem 3.2.* Throughout the proof, $C_1', C_2', \dots > 0$ are quantities depending only on $d$, $\alpha$, $M$, $c_X$ and $C_X$. Let $f_{\mathrm{LocPol}}(\mathcal{D}_n, X_{n+1}) := \tilde{m}_n(X_{n+1})$ be the truncated local polynomial estimator as defined in Appendix A with degree $p$, bandwidth $h = n^{-1/(2\alpha+d)}$ and kernel $K(x) := (1 - \|x\|_1)_+^2$. By Theorem A.6, we have

$$R(f_{\mathrm{LocPol}}) = \mathbb{E}\big(\{Y_{n+1} - f_{\mathrm{LocPol}}(\mathcal{D}_n, X_{n+1})\}^2\big)$$
$$= \mathbb{E}(\varepsilon_{n+1}^2) + \mathbb{E}\Big(\int_{[0,1]^d} \{\tilde{m}_n(x) - m(x)\}^2 f_X(x)\, dx\Big) \leq \sigma^2 + C_1' n^{\frac{-2\alpha}{2\alpha+d}}. \tag{23}$$

By Theorem D.1, since $R(f_{\mathrm{TF}}) \geq \inf_{f \in \mathcal{F}(d_{\mathrm{e}}, d_{\mathrm{ffn}}, L, B, M)} R(f)$, we have

$$\mathbb{E}\{R(\hat{f}_\Gamma)\} \leq 2R(f_{\mathrm{TF}}) - \sigma^2 + C_4'(M+1)^5 L d_{\mathrm{e}}(d_{\mathrm{e}} + d_{\mathrm{ffn}}) \frac{L \log\{(B+1)nd_{\mathrm{e}}d_{\mathrm{ffn}}\} + \log\Gamma}{\Gamma}.$$

Thus, by Theorem 3.1, with $d_{\mathrm{e}} := 2d + 2D + 5$, $d_{\mathrm{ffn}} := 3(D+1)(28 + 2p)$, $L = \lceil C \log(en) \rceil$, $B = Cn^2$ and $\Gamma \geq Cn^{2\alpha/(2\alpha+d)} \log^3(en)$, we have

$$\mathbb{E}\{R(\hat{f}_\Gamma)\} - \sigma^2 \leq 2(R(f_{\mathrm{TF}}) - \sigma^2) + C_5' \frac{\log^3(en) + \log(en)\log\Gamma}{\Gamma}$$

$$\leq 2(R(f_{\mathrm{LocPol}}) - \sigma^2) + \frac{C_3'}{n} + C_5' \frac{\log^3(en) + \log(en)\log\Gamma}{\Gamma} \leq C_6' n^{\frac{-2\alpha}{2\alpha+d}},$$

where the final inequality follows from (23). $\qquad\square$

## F. Approximating Linear Attention by Softmax

For $j \in [n]$, $x \in \mathbb{R}^n$ and $t \geq 0$, define $g_j(t) := \mathrm{softmax}(tx)_j = e^{tx_j}/\left(\sum_{i=1}^n e^{tx_i}\right)$ to be the $j$th coordinate of the softmax transformation of $tx$. Then

$$g_j'(t) = \frac{x_j e^{tx_j}\left(\sum_{i=1}^n e^{tx_i}\right) - e^{tx_j}\left(\sum_{i=1}^n x_i e^{tx_i}\right)}{\left(\sum_{i=1}^n e^{tx_i}\right)^2},$$

so $g_j'(0) = x_j/n - \sum_{i=1}^n x_i/n^2$. By Taylor's theorem,

$$\mathrm{softmax}(tx)_j = g_j(t) = g_j(0) + tg_j'(0) + O(t^2) = \frac{1}{n} + t\left\{\frac{x_j}{n} - \frac{\sum_{i=1}^n x_i}{n^2}\right\} + O(t^2),$$

as $t \to 0$. Hence, for $\boldsymbol{Z} \in \mathbb{R}^{n \times d_{\mathrm{e}}}$ and $\boldsymbol{Q}, \boldsymbol{K}, \boldsymbol{V} \in \mathbb{R}^{d_{\mathrm{e}} \times d_{\mathrm{e}}}$, we have

$$\mathrm{softmax}\left(t\boldsymbol{Z}\boldsymbol{Q}(\boldsymbol{Z}\boldsymbol{K})^\mathsf{T}\right)\boldsymbol{Z}\boldsymbol{V} = \frac{\mathbf{1}_{n \times n}\boldsymbol{Z}\boldsymbol{V}}{n} + t\left\{\frac{\boldsymbol{Z}\boldsymbol{Q}(\boldsymbol{Z}\boldsymbol{K})^\mathsf{T}\boldsymbol{Z}\boldsymbol{V}}{n} - \frac{\boldsymbol{Z}\boldsymbol{Q}(\boldsymbol{Z}\boldsymbol{K})^\mathsf{T}\mathbf{1}_{n \times n}\boldsymbol{Z}\boldsymbol{V}}{n^2}\right\} + O(t^2). \tag{24}$$

Below, we outline how to construct a transformer with two softmax attention layers followed by $O(\log n)$ FFN layers such that if the input matrix is $\boldsymbol{Z}' := (\boldsymbol{Z}, \mathbf{0}_{n \times 2d_{\mathrm{e}}}) \in \mathbb{R}^{n \times 3d_{\mathrm{e}}}$, then its output approximates the output $(\boldsymbol{Z} + \boldsymbol{Z}\boldsymbol{Q}(\boldsymbol{Z}\boldsymbol{K})^\mathsf{T}\boldsymbol{Z}\boldsymbol{V}, 0_{n \times 2d_{\mathrm{e}}})$ of a linear attention layer.

**First attention layer:** We take $\boldsymbol{Q}^{(1)} = \boldsymbol{K}^{(1)} := \mathbf{0}_{3d_{\mathrm{e}} \times 3d_{\mathrm{e}}}$ and

$$\boldsymbol{V}^{(1)} := \begin{pmatrix} \mathbf{0}_{d_{\mathrm{e}} \times d_{\mathrm{e}}} & n\boldsymbol{I}_{d_{\mathrm{e}}} & \mathbf{0}_{d_{\mathrm{e}} \times d_{\mathrm{e}}} \\ \mathbf{0}_{2d_{\mathrm{e}} \times d_{\mathrm{e}}} & \mathbf{0}_{2d_{\mathrm{e}} \times d_{\mathrm{e}}} & \mathbf{0}_{2d_{\mathrm{e}} \times d_{\mathrm{e}}} \end{pmatrix}.$$

Then

$$\mathrm{Attn}^{(1)}(\boldsymbol{Z}') := \boldsymbol{Z}' + \mathrm{softmax}\left(\boldsymbol{Z}'\boldsymbol{Q}^{(1)}(\boldsymbol{Z}'\boldsymbol{K}^{(1)})^\mathsf{T}\right)\boldsymbol{Z}'\boldsymbol{V}^{(1)} = \begin{pmatrix} \boldsymbol{Z} & \mathbf{1}_{n \times n}\boldsymbol{Z} & \mathbf{0}_{n \times d_{\mathrm{e}}} \end{pmatrix}.$$

**Second attention layer:** We take

$$\boldsymbol{Q}^{(2)} := \begin{pmatrix} \mathbf{0}_{d_{\mathrm{e}} \times 2d_{\mathrm{e}}} & t\boldsymbol{Q} \\ \mathbf{0}_{2d_{\mathrm{e}} \times 2d_{\mathrm{e}}} & \mathbf{0}_{2d_{\mathrm{e}} \times d_{\mathrm{e}}} \end{pmatrix}, \qquad \boldsymbol{K}^{(2)} := \begin{pmatrix} \mathbf{0}_{d_{\mathrm{e}} \times 2d_{\mathrm{e}}} & \boldsymbol{K} \\ \mathbf{0}_{2d_{\mathrm{e}} \times 2d_{\mathrm{e}}} & \mathbf{0}_{2d_{\mathrm{e}} \times d_{\mathrm{e}}} \end{pmatrix} \quad \text{and} \quad \boldsymbol{V}^{(2)} \begin{pmatrix} \mathbf{0}_{d_{\mathrm{e}} \times 2d_{\mathrm{e}}} & \frac{n}{t}\boldsymbol{V} \\ \mathbf{0}_{2d_{\mathrm{e}} \times 2d_{\mathrm{e}}} & \mathbf{0}_{2d_{\mathrm{e}} \times d_{\mathrm{e}}} \end{pmatrix}.$$

Then, writing $\boldsymbol{Z}^{(1)} := \mathrm{Attn}^{(1)}(\boldsymbol{Z}') = (\boldsymbol{Z}, \mathbf{1}_{n \times n}\boldsymbol{Z}, \mathbf{0}_{n \times d_{\mathrm{e}}})$ for the output of the first attention layer, we have by (24) that

$$\mathrm{Attn}^{(2)}(\boldsymbol{Z}^{(1)}) := \boldsymbol{Z}^{(1)} + \mathrm{softmax}\left(\boldsymbol{Z}^{(1)}\boldsymbol{Q}^{(2)}(\boldsymbol{Z}^{(1)}\boldsymbol{K}^{(2)})^\mathsf{T}\right)\boldsymbol{Z}^{(1)}\boldsymbol{V}^{(2)}$$

$$= \begin{pmatrix} \boldsymbol{Z} & \mathbf{1}_{n \times n}\boldsymbol{Z} & \frac{\mathbf{1}_{n \times n}\boldsymbol{Z}\boldsymbol{V}}{t} + \boldsymbol{Z}\boldsymbol{Q}(\boldsymbol{Z}\boldsymbol{K})^\mathsf{T}\boldsymbol{Z}\boldsymbol{V} - \frac{\boldsymbol{Z}\boldsymbol{Q}(\boldsymbol{Z}\boldsymbol{K})^\mathsf{T}\mathbf{1}_{n \times n}\boldsymbol{Z}\boldsymbol{V}}{n} + O(t) \end{pmatrix}.$$

Define $S := \boldsymbol{Z}^\mathsf{T} 1_n \in \mathbb{R}^{d_\mathrm{e}}$. Then

$$\mathrm{Attn}^{(2)}(\boldsymbol{Z}^{(1)})_{i,:} = \left( Z_i^\mathsf{T} \quad S^\mathsf{T} \quad \frac{S^\mathsf{T}\boldsymbol{V}}{t} + \left(\boldsymbol{ZQ}(\boldsymbol{ZK})^\mathsf{T}\boldsymbol{ZV}\right)_i - \frac{Z_i^\mathsf{T}\boldsymbol{QK}^\mathsf{T}SS^\mathsf{T}\boldsymbol{V}}{n} + O(t) \right).$$

**FFN layers:** We take the FFN layers such that for $z, S, u \in \mathbb{R}^{d_\mathrm{e}}$,

$$\mathrm{FFN}\left((z^\mathsf{T},\, S^\mathsf{T},\, u^\mathsf{T})\right) \approx \left( z^\mathsf{T} + u^\mathsf{T} - \frac{S^\mathsf{T}\boldsymbol{V}}{t} + \frac{z^\mathsf{T}\boldsymbol{QK}^\mathsf{T}SS^\mathsf{T}\boldsymbol{V}}{n} \quad 0_{d_\mathrm{e}}^\mathsf{T} \quad 0_{d_\mathrm{e}}^\mathsf{T} \right).$$

Note that the right-hand side of the above equation only involves multiplication and summation. Thus, if we want the approximation error to be $O(n^{-C})$ for some $C > 0$, then it suffices to take the width of the FFN layers to be constant in $n$ and the depth to be $\Theta(\log n)$; see Lemma B.5. Finally, taking $t = n^{-C}$ for some $C > 0$ large enough, we deduce that

$$\mathrm{FFN} \circ \mathrm{Attn}^{(2)} \circ \mathrm{Attn}^{(1)}(\boldsymbol{Z}') = \left( \boldsymbol{Z} + \boldsymbol{ZQ}(\boldsymbol{ZK})^\mathsf{T}\boldsymbol{ZV} \quad 0_{n \times 2d_\mathrm{e}} \right) + O(n^{-1}).$$

