# OpenReview forum: "Efficient and Minimax Optimal In-context Nonparametric Regression with Transformers"
_ICML.cc/2026/Conference — ICML 2026 regular_

### Official Review · Reviewer_7X5C · 2026-03-09

**Soundness:** 4
**Presentation:** 4
**Significance:** 3
**Originality:** 3
**Overall Recommendation:** 5
**Confidence:** 4

**Summary:**

This paper investigates the theoretical foundations of in-context learning (ICL) for nonparametric regression. Specifically, the authors aim to determine whether a pretrained transformer can achieve the minimax-optimal rate of convergence for $\alpha$-Hölder smooth regression functions. The core contribution is a constructive proof demonstrating that a transformer can efficiently approximate a truncated Local Polynomial Estimator (LPR). The authors show that the transformer architecture achieves this through a division of labor:
(1). FFNs are used to compute a kernel-weighted monomial basis. Crucially, by applying these FFNs identically across all tokens (leveraging the standard position-wise parameter sharing of transformers) , the model parametrisation remains completely independent of the context length $n$, effectively linearizing the nonparametric problem without bloating the parameter count.
(2). Attention layers are employed to implement gradient descent. The authors prove (via Lemma B.9) that by taking $T$ gradient descent steps, the error bounds decay exponentially, specifically tracking an $\exp(-\frac{C_1 T}{2C_2})$ convergence rate toward the optimal least-squares solution $w_*$.

Because of this exponential convergence, the paper establishes that the transformer only requires $L = \Theta(\log n)$ layers (and thus $\Theta(\log n)$ parameters) to approximate the LPR to an accuracy of $O(1/n)$. Finally, by bounding the covering numbers of this logarithmically sized transformer class, the authors prove that the empirical risk minimizer achieves the minimax-optimal rate of $O(n^{-2\alpha/(2\alpha+d)})$ in mean squared error. Notably, this is accomplished while requiring substantially fewer parameters and pretraining sequences ($\Omega(n^{2\alpha/(2\alpha+d)}\log^3 n)$) than previous works in the literature.

**Compliance With Llm Reviewing Policy:**

Affirmed.

**Key Questions For Authors:**

1. Since the empirical value of ICL is most prominent in the few-shot regime, does your proposed asymptotic framework offer any insights, algebraic properties, or non-trivial guarantees when the sample size $n$ is small?
2. The extension to Softmax attention at the end of Section 3 relies on a Taylor expansion around $t \to 0$, which incurs an $O(t^2)$ approximation error. Given that the proposed Transformer requires $L = \Theta(\log n)$ layers, does this $O(t^2)$ error accumulate or compound across layers? If so, does it break the $O(1/n)$ approximation error bound established in Theorem 3.1?
3. Could the authors comment on whether the proposed Attention-as-LPR framework can be extended to achieve minimax optimal rates for Sobolev or Besov function classes?
4. How does the technical difficulty of your LPR construction compare to recent works approximating NW estimators (Shen et al., 2025) or B-splines (Kim et al., 2024)? Specifically, does the $\Theta(\log n)$ parameter efficiency stem primarily from choosing LPR, or from the Attention-as-GD mapping?

**Limitations:**

Yes

**Strengths And Weaknesses:**

Strengths:
1. Approximation Framework: Instead of treating the Transformer as a standard passive approximator, the paper models it as an active executor of an LPR algorithm. This efficient approximation keeps the error tightly bounded without increasing model complexity. This is the key reason the framework ultimately achieves the minimax-optimal convergence rate.
2. Parameter Efficiency: The authors effectively use the position-wise parameter sharing of FFNs to keep the parameter count independent of the context length $n$. This provides a clear explanation for why standard Transformer architectures are suited for high-dimensional nonparametric ICL.
3. Clarity and Generalizability: The paper is well-structured. Figures 5 and 6 are particularly helpful for visualizing the algorithmic mapping before diving into the appendix proofs. Additionally, using Taylor expansion to show how the framework can extend from linear attention to standard softmax attention is a practical addition.

Weaknesses:
1. Asymptotic Rates vs. Few-Shot Reality: The analysis focuses heavily on asymptotic convergence rates ($n \to \infty$). However, the primary empirical value of ICL is its performance in the few-shot regime where $n$ is very small. The current large-sample limit framework does not address the algebraic properties or emergent behaviors of few-shot ICL.
2. i.i.d. Assumption: The data generation process relies on an i.i.d. assumption. This does not reflect the dependent data settings found in real-world autoregressive next-token prediction tasks, though the authors acknowledge this limitation in the summary.
3. ERM Optimization Gap: The optimality guarantees assume a global ERM can be successfully found. Since Transformer training is a highly non-convex optimization problem, there is a distinct gap between the theoretical optimum and practical trainability. While this is a common issue in deep learning theory, it remains a limitation when using these guarantees to explain actual empirical success.

---

> ### Author Rebuttal · Authors · 2026-03-30
>
> Thank you for this valuable feedback;
> we are pleased that you find the paper to be
> impactful and well structured.
>
> Asymptotic Rates vs. Few-Shot Reality: Our results are given in terms of two key quantities:
> the context length $n$ and the number of pretraining sequences $\Gamma$.
> While our main theorems are non-asymptotic
> (they hold for every $n \in \mathbb{N}$), they are stated only
> up to unspecified constant factors.
> On the other hand, minimax lower bounds already establish the fundamental difficulty of nonparametric regression when $n$ is small.
> Therefore, one may consider alternative supervised learning
> settings which may be more appropriate for
> few-shot learning.
> Nonetheless, we believe that the nonparametric regression framework
> is important for two reasons.
> First, it provides a natural model for
> in-context tabular data problems
> (Hollman et al., 2025).
> Second, in sufficiently smooth problems, one may obtain
> dimension-independent rates.
> For example, if $\alpha = d$ then the minimax rate is
> $n^{-2\alpha / (2\alpha + d)} = n^{-2/3}$.
>
> i.i.d. Assumption: We agree that the i.i.d. assumption is strong, although it
> does simplify the exposition.
> It would not be hard to extend the results to
> strongly mixing dependent data, for example,
> a popular model for time series.
> Technically, this would simply require the use of
> concentration inequalities for strongly mixing rather than
> independent data, and should yield essentially the same results.
> It should also be possible to relax the `identically distributed'
> assumption; all we require is that the regression functions
> are fixed across data points within each pretraining sequence.
>
> ERM Optimization Gap: There is a well documented gap in the theory
> between the performance of empirical risk minimising
> neural networks and neural networks trained by gradient
> descent methods.
> While some progress towards characterising
> the limit points of gradient-based algorithms
> has been made recently,
> it is generally limited to simple settings,
> requiring, for example,
> infinitely deep and wide residual networks
> (Barboni et al., 2025; Lu et al., 2020).
> In the specific setting of transformer architectures,
> the training dynamics of shallow networks
> are studied by, e.g.,
> Huang et al., 2023,
> Wu et al., 2023, and
> Song et al, 2024;
> the multi-layer case appears to be  more challenging to analyse.
> As such, we (and many other authors working on deep learning theory)
> focus on the statistical rather than computational
> aspects of the problem.
>
> Key Question 1: Please see our response to your first point.
>
> Key Question 2:
> While errors do indeed accumulate through the layers,
> they do so in an additive manner.
> As such, so long as we approximate
> each softmax attention layer
> with accuracy $t=O(1/n)$,
> we retain the minimax estimation rate.
>
> Key Question 3:
> Our results do generalise to Sobolev and Besov classes
> as we require only integrated
> (and not pointwise or uniform) mean squared error bounds.
> More precisely, our results hold for
> the (fractional) Sobolev spaces $W^{\alpha,p}$ where $p\geq 2$,
> which coincide with the Besov spaces $B^{\alpha}_{p,p}$.
> For $p<2$, it is known that linear methods fail
> to achieve minimax optimality in certain settings
> (Donoho et al., 1996).
>
> Key Question 4:
> Our proof establishing optimal rates with $\Theta(\log n)$
> parameters relies crucially on
> both the approximation by local polynomials and on the use
> of gradient descent to solve the kernel least squares problem.
> For instance,
> Kim et al., 2024
> use B-spline expansions instead of local polynomials,
> where for Hölder (or Besov)
> functions with smoothness $\alpha$,
> a number of basis functions of order $n^{d/(2\alpha+d)}$ are needed.
> As such, it is impossible
> to achieve a good approximation error with only
> $\Theta(\log n)$ parameters using this approach.
> Moreover, Shen et al., 2025
> did not use gradient descent to compute the
> Nadaraya--Watson (local constant) estimator,
> giving worse approximation bounds for a fixed
> number of parameters and handling only $\alpha \in (0, 1]$.
> With local polynomials, gradient descent is essential to
> avoid inverting the monomial basis matrix;
> matrix inversion is non-linear and not easily approximated
> directly with transformers.
>
> **References**
>
> Hollman et al., 2025.
> Accurate predictions on small data with a tabular foundation model.
> Nature, 637(8045), 319--326.
>
> Song et al, 2024.
> Unraveling the gradient descent dynamics of transformers.
> NeurIPS, 37, 92317--92351.
>
> Wu et al., 2023.
> On the convergence of encoder-only shallow transformers. NeurIPS, 36, 52197--52237.
>
> Huang et al., 2023.
> In-context convergence of transformers,
> arXiv:2310.05249.
>
> Donoho et al., 1996.
> Density estimation by wavelet thresholding.
> Annals of Statistics, 24(2), 508--539.
>
> Kim et al., 2024.
> Transformers are Minimax Optimal Nonparametric In-Context Learners.
> NeurIPS, 37, 106667--106713.
>
> Shen et al., 2025.
> Understanding in-context learning on structured manifolds:
> Bridging attention to kernel methods.
> arXiv: 2506.10959.

---

> > ### Author Rebuttal · Reviewer_7X5C · 2026-04-05
> >
> > The authors have addressed all of my concerns in their response.

---

> > > ### Author Response · Authors · 2026-04-07
> > >
> > > Many thanks for your positive assessment.

---

### Official Review · Reviewer_7P46 · 2026-03-13

**Soundness:** 4
**Presentation:** 3
**Significance:** 2
**Originality:** 2
**Overall Recommendation:** 4
**Confidence:** 4

**Summary:**

"In-context learning" can be formulated as a non-parametric regression problem, where target functions are sampled from a distribution $m \sim P_\mathcal{H}$. The goal is to be able to predict $m(X_{n+1})$ given both $(X_i, m(X_i))$ for $i \in [n]$ as well as $X_{n+1}$, where the $X_{i}$'s are i.i.d. sampled from some distribution $P_X$. (For simplicity of this summary, we ignore noise terms in the training data). This task can be accomplished with so-called "truncated local polynomial estimators", and when the sample space of $P_\mathcal{H}$ consists of Holder-continuous functions, the population risk of the local polynomial estimator can be bounded. Indeed, with high probability the local polynomial estimator admits both a closed form expression, and a variational formulation as the minimizer of a strongly convex objective. It is this variational formulation that is able to be encoded into a pretrained transformer with linear attention---feed forward layers are used to approximate polynomials, and gradient descent on the objective is executed with linear attention blocks. The authors prove via construction that transformers with linear attention and only $\Theta(\log n)$ parameters can obtain the minimax-optimal rate of convergence in this setting.

**Compliance With Llm Reviewing Policy:**

Affirmed.

**Final Justification:**

See my response to the author's rebuttal.

**Key Questions For Authors:**

- (Q1) What important, difficult question relevant to mathematical deep learning is this paper addressing?
- (Q2) Can you argue that your results reflect, or otherwise inform, how trained transformers actually behave?
- (Q3) What aspects of your main results are unique to transformers? Can similar arguments be made for other recurrent architectures?
- (Q4) (minor) How essential are the cutoffs beginning on line 250? Can the argument be modified to remove them?

**Limitations:**

Yes.

**Strengths And Weaknesses:**

This paper is focused with a clear, understandable objective. The mathematics is appropriately rigorous. The proofs appear to be correct, and I am confident that any errors could probably be patched without compromising their main results.

However, it has been observed many times that certain neural network architectures can recapitulate well-known algorithms; that transformers with linear attention can do so with yet another algorithm is not surprising. In fact, it is likely that any recurrent architecture that both
1. allows for arbitrary interleaving of matrix operations (as multiple layers of linear attention does), and
2. uses MLPs to approximate non-linear operations

should be able to implement (or approximate) essentially any algorithm simply by choosing weights in a straight-forward way. There have been many papers written along these same lines. Although this line of thought is somewhat interesting in its own right, it is not clear that this type of research has ever led to any insights as to how deep networks actually behave, or led to any insights on the important and difficult questions in the mathematics of deep learning. Specifically,
1. It is not clear that these constructions reflect the behavior of networks trained from data (in fact, I find it rather unlikely).
2. When two network architectures (for example, LSTM and Transformers) can be shown to implement the same algorithm by clever choice of weights, yet one architecture clearly outperforms the other on relevant tasks, then it is not clear what insights the artificial constructions actually provide.

Due to these fundamental limitations, I am recommending **weak rejection**, but I am open to increasing my score. This paper is certainly above average among the ML papers I have reviewed recently, and it deserves to be published somewhere. Barring any major flaws discovered by other reviewers, I feel it is technically sound. But is not likely to be relevant for anyone who is not playing the very specific, academic game of recapitulating well-known algorithms as neural networks.

---

> ### Author Rebuttal · Authors · 2026-03-30
>
> Thank you for this valuable feedback;
> we are pleased that you find the paper to be clear,
> understandable and sound.
> We agree that there is now a growing literature on
> the ability of neural networks to (approximately) recover
> classical statistical algorithms,
> and indeed it is generally the case that an
> appropriately flexible architecture can replicate any
> sufficiently regular algorithm.
> Our main messages here are twofold.
> First, our approximation of local polynomial estimators
> is to be treated largely as a proof technique and not as
> the central result of the paper.
> Indeed, our main Theorem 3.2 makes no reference to local
> polynomials, and instead shows that any empirical
> risk-minimising transformer achieves an excess
> population risk that is minimax optimal up to a constant
> factor.
> This result is new in the setting of in-context
> $\alpha$-smooth regression problems where $\alpha > 0$.
> Second, we show not only that transformers are able
> to replicate local polynomial methods, but also that
> they are able to do so very efficiently, requiring
> only $\Theta(\log n)$ parameters to attain optimal rates.
> In contrast, other recurrent architectures
> (or proof strategies)
> may require more parameters and therefore need more
> pretraining sequences.
> Thus, we offer some
> theoretical justification for the practical statistical
> successes of transformers over other architectures.
>
> Reponse to Q1:
> We are the first to show that a transformer with $\Theta(\log n)$ parameters
> can achieve the minimax optimal rate for in-context nonparametric regression.
> Previous literature requires at least polynomially many parameters in $n$.
> Our result highlights the expressive power of transformers and offers
> new insights into their empirical statistical successes.
>
> Response to Q2:
> There is a well documented gap in the theory
> between the performance of empirical risk minimising
> neural networks and neural networks trained by gradient
> descent methods
> (Barboni et al., 2025; Lu et al., 2020).
> As such, we make no claims on the properties
> of algorithmically trained transformers,
> focusing instead on the statistical rather
> than computational aspects of the problem.
>
> Response to Q3:
> Our proofs exploit two key properties to achieve
> optimal rates with just $\Theta(\log n)$ parameters.
> First, the use of gradient descent to compute
> the local polynomial estimator as a least squares
> solution is highly parameter-efficient due to
> fast convergence in strongly convex settings
> and the ability of transformers to
> implement matrix multiplication with $O(1)$ parameters.
> Second, the construction of local monomial basis functions
> and piecewise polynomial kernels is efficient and straightforward
> when using ReLU feed-forward layers.
> This second property holds for other recurrent architectures
> but the first is less clear;
> we do not know whether LSTM networks
> (for example) are equally efficient.
> Another reason that we focus on transformers is because they are currently
> the most popular and widely used deep learning architecture in practice.
>
> Response to Q4:
> The local polynomial estimator may be
> truncated at any constant greater than or equal to $M$.
> Some form of truncation is generally essential
> for minimax optimality here;
> without this, the population risk of
> the procedure may be infinite
> (e.g. Györfi et al., 2002, Problem 10.3).
> Nonetheless, the purpose of Section 2.5 is solely to
> elucidate the proof of the main result Theorem 3.2.
>
> **References**
>
> Barboni et al., 2025.
> Understanding the training of infinitely deep and wide ResNets with
> conditional optimal transport.
> Communications on Pure and Applied Mathematics 78(11), 2149--2205.
>
> Lu et al., 2020.
> A mean field analysis of deep ResNet and beyond: Towards provably
> optimization via overparameterization from depth.
> Proceedings of the 37th International Conference on Machine Learning,
> 119, 6426--6436.
>
> Györfi et al., 2002.
> A Distribution-Free Theory of Nonparametric Regression.
> Springer, New York.

---

> > ### Author Rebuttal · Reviewer_7P46 · 2026-04-03
> >
> > Thank you for a thorough response. I find the paper interesting, and the proof technique may be applied in other contexts. Although I still have doubts about the novelty and utility of this endeavor, there is likely and audience for this paper. I will raise my score to "Weak Accept"

---

> > > ### Author Response · Authors · 2026-04-07
> > >
> > > Thank you for recognising the merits of our work and raising your score.

---

### Official Review · Reviewer_2E5P · 2026-03-14

**Soundness:** 4
**Presentation:** 3
**Significance:** 2
**Originality:** 3
**Overall Recommendation:** 4
**Confidence:** 4

**Summary:**

The work improves upon existing risk bounds on in-context learning (ICL). The technical innovation is  showing that transformers with log n  parameters are able to attain the optimal rate of convergence. This improves on the setting of Kim et al. (2024) and Shen
et al. (2025).

**Compliance With Llm Reviewing Policy:**

Affirmed.

**Key Questions For Authors:**

N/A

**Strengths And Weaknesses:**

The work is highly technical and interesting. Its presentation perhaps can be improved as often one encounters a "wall" of definitions without much intuition to drive the technical result. Nevertheless, the technical quality of the work is high, and the results are improving upon existing results.

The presentation could also benefit some empirical validation. While the work is theoretical, it may be more complete with some experiment, e.g., Shen et al.

The significance, given the previous works is not high. While this work improves upon the number of transformers parameters, the current trend does not hint that reducing the number of parameters is significant.

---

> ### Author Rebuttal · Authors · 2026-03-30
>
> Thank you for this valuable feedback; we are
> pleased that you find the work to be interesting and
> of high quality.
>
> To lighten the beginning of Section 2, we propose to include the following
> motivating introduction:
>
> "In the ICL framework, pretraining data is modelled as
> a collection of $\Gamma$ pretraining
> sequences, each generated according to a similar
> (but not identical) mechanism. We consider an in-context
> regression model, supposing that each such sequence,
> indexed by $\gamma \in [\Gamma]$,
> consists of covariates $X_i^{(\gamma)} \in \mathbb{R}^d$
> and responses $Y_i^{(\gamma)} \in \mathbb{R}$ for $i \in [n]$.
> We assume that the covariates $X_i^{(\gamma)}$
> and errors $\varepsilon_i^{(\gamma)} \in \mathbb{R}$
> are all drawn from a common distribution;
> the responses are then generated as
> $Y_i^{(\gamma)} = m^{(\gamma)}(X_i^{(\gamma)}) + \varepsilon_i^{(\gamma)}$, where
> the regression functions $m^{(\gamma)}$ may vary across
> pretraining indices."
>
> As you have pointed out, the primary focus of this paper
> is on the statistical theory of deep learning with
> transformers.
> Empirical evidence for the success of transformers
> for in-context regression is widely available;
> see, for example, Hollman et al., 2025.
> To the best of our knowledge,
> Shen et al., 2025 also do not present
> any experimental results,
> and we are concerned that simulation studies
> may detract from the main theoretical focus of our work.
>
> We understand that current trends in deep learning
> suggest that overparametrisation is not a serious issue
> in practice.
> However, our aim is to demonstrate the relative
> efficiency with which transformers can
> learn smooth regression functions.
> By achieving optimal in-context rates with just
> $\Theta(\log n)$ parameters, our work presents
> a theoretical justification for the widely observed
> phenomenon that transformers typically outperform
> other deep learning architectures in practice.
>
> **References**
>
> Hollman et al., 2025.
> Accurate predictions on small data with a tabular foundation model.
> Nature, 637(8045), 319--326.
>
> Shen et al., 2025.
> Understanding in-context learning on structured manifolds:
> Bridging attention to kernel methods.
> arXiv: 2506.10959.

---

> > ### Author Rebuttal · Reviewer_2E5P · 2026-04-06
> >
> > I thank for authors for their response. As said and graded, the work is important and I hope to see it in the upcoming ICML. As the review process gets more formal these days, I am forced to acknowledge reading the response with a rating: I hope my rating is not diminishing in any way. Sometimes, an author response cannot fix everything and my comments are a bit harder to resolve in a short rebuttal. I am kindly repeating my main comments and acknowledge that despite my concerns I believe this paper can be a valuable addition to ICML:
> >
> > (i) presentation is highly technical. I kindly ask the authors to take a look at page 3 of the submission (specifically section 1.4 and section 2.1) and compare it to Shen et al. (Understanding In-Context Learning on Structured
> > Manifolds: Bridging Attention to Kernel Methods), specifically Section 2. While this seems a meaningless exercise, one can see the density of mathematical symbols in the current submission. Broadly, oftentimes, technical details obscure intuition and I think there is a deep intuition in this work that yet to be fleshed out.
> >
> > (ii) experiments: perhaps we relate different meaning to experimental validation. Shen et al. (Understanding In-Context Learning on Structured Manifolds: Bridging Attention to Kernel Methods, https://arxiv.org/pdf/2506.10959v2), Figure 1, Figure 2, Figure 3, while synthetic, are interesting experimental intuition about the work. Perhaps the authors referred to v1 in their response.

---

> > > ### Author Response · Authors · 2026-04-07
> > >
> > > Thank you for your feedback and for your positive assessment of the quality of our work.
> > >
> > > To address the point you raised about presentation, in our revision we will expand the introduction and early sections to provide more intuition behind the construction, in particular clarifying how the transformer implements local polynomial-type behaviour in context, and highlighting the key ideas before introducing the formal definitions (some of which will be deferred to the supplement). We will also streamline some of the notation to improve readability.
> > >
> > > We apologise that we were indeed studying v1 rather than v2 of Shen et al.~(2025).  In our revision, we will be happy to include empirical experiments to study the behaviour of a trained transformer. In particular, we will compare its performance with standard local polynomial regression.

---

### Official Review · Reviewer_oV6M · 2026-03-18

**Soundness:** 3
**Presentation:** 2
**Significance:** 2
**Originality:** 3
**Overall Recommendation:** 3
**Confidence:** 3

**Summary:**

This paper propose a construction of transformers solving $\alpha$-Hölder smooth nonparametric regression in context. The authors approach this by fitting a local polynomial in-context. The construction works as follows. Each transformer block is a linear attention layer followed by a Relu-activated MLP layer. The construction consists of $\Theta(\log n)$ blocks where $n$ is the number of in-context examples. The first three blocks preprocess all $X_i$ by centralizing, rescaling and calculating kernel matrices. The kernel is chosen to be effectively calculated by Relu MLP layers. The next $\Theta(\log n)$ blocks approximate the kernel-weighted monomial basis, and the last $\Theta(\log n)$ blocks solve a kernel least square problem using $\Theta(\log n)$ steps of gradient descent. The local polynomial estimators can be read out from the least square solution. The achieved mean squared error is $O(n^{-2\alpha/(2\alpha+d)})$ with $O(n^{2\alpha/(2\alpha+d)}\log^3 n)$ pretrain sequences.

**Compliance With Llm Reviewing Policy:**

Affirmed.

**Key Questions For Authors:**

Is there any empirical result matching the parameter complexity/minimax error/sample complexity of this contruction in a trained transformer?

**Limitations:**

No. The author could mention whether the proposed construction/parameter complexity/minimax error/sample complexity can be achieved by a trained transformer. Currently it is unclear.

**Strengths And Weaknesses:**

Strength: The construction is more parameter-efficient than previous works.

Weakness: There is no optimization analysis or empirical results justifying the significance of this construction.

---

> ### Author Rebuttal · Authors · 2026-03-30
>
> Thank you for this valuable feedback.
> Our main result is stated for a transformer that
> achieves the minimal empirical risk
> within an appropriate function class.
> In particular, we show that such a pretrained transformer
> also achieves an excess population risk that matches
> (up to constant factors) the minimax lower bound
> for smooth nonparametric regression.
> Empirical evidence for the success of transformers
> for in-context regression is widely available;
> see, for example, Hollman et al., 2025.
> There is a well documented gap in the theory
> between the performance of empirical risk minimising
> neural networks and neural networks trained by gradient
> descent methods.
> While some progress towards characterising
> the limit points of gradient-based algorithms
> has been made recently,
> it is generally limited to simple settings,
> requiring, for example,
> infinitely deep and wide residual networks
> (Barboni et al., 2025; Lu et al., 2020).
> In the specific setting of transformer architectures,
> the training dynamics of shallow networks
> are studied by, e.g.,
> Huang et al., 2023,
> Wu et al., 2023, and
> Song et al, 2024;
> the multi-layer case appears to be substantially
> more challenging to analyse.
> As such, we (and many other authors working on deep learning theory)
> focus on the statistical rather than optimisation
> aspects of the problem.
> In our analysis, we characterise the size of the transformer and the rate
> of convergence in terms of the context length $n$,
> and there are constant factors
> depending on other problem parameters,
> including polynomial dependence on the covariate dimension.
> In practice, people often use large transformers, and we view our
> contribution as a theoretical justification of the efficiency and
> expressive power of transformers, in the sense that $\Theta(\log n)$
> parameters are sufficient to perform optimal nonparametric regression.
>
> **References**
>
> Hollman et al., 2025.
> Accurate predictions on small data with a tabular foundation model.
> Nature, 637(8045), 319--326.
>
> Barboni et al., 2025.
> Understanding the training of infinitely deep and wide ResNets with
> conditional optimal transport.
> Communications on Pure and Applied Mathematics 78(11), 2149--2205.
>
> Lu et al., 2020.
> A mean field analysis of deep ResNet and beyond: Towards provably
> optimization via overparameterization from depth.
> Proceedings of the 37th International Conference on Machine Learning,
> 119, 6426--6436.
>
> Song et al., 2024.
> Unraveling the gradient descent dynamics of transformers.
> Advances in Neural Information Processing Systems, 37, 92317--92351.
>
> Wu et al., 2023.
> On the convergence of encoder-only shallow transformers. Advances in Neural Information Processing Systems, 36, 52197--52237.
>
> Huang et al., 2023.
> In-context convergence of transformers,
> arXiv:2310.05249.

---

> > ### Author Rebuttal · Reviewer_oV6M · 2026-04-03
> >
> > My concern remains un addressed as the lack of discussions on how the real trained transformers works.

---

> > > ### Author Response · Authors · 2026-04-07
> > >
> > > Thank you for your follow-up. In our revision, we will be happy to include empirical experiments to study the behaviour of a trained transformer. In particular, we will compare its performance with standard local polynomial regression.
> > >
> > > We would like to clarify further that our main Theorem 3.2 is stated for transformers trained via empirical risk minimisation. In particular, we show that a transformer with $\Theta(\log n)$ parameters, pretrained with $\Omega(n^{2\alpha/(2\alpha+d)} \log^3 n)$ sequences, achieves the minimax optimal rate for nonparametric regression.  Crucially, we do not claim, nor require, that a trained transformer explicitly implements local polynomial regression.  Rather, Theorem 3.1 should be interpreted as an expressivity result: transformers with $\Theta(\log n)$ parameters are sufficiently expressive for in-context nonparametric regression. Approximating local polynomial regression is one constructive way to establish this, but it is by no means the only possible mechanism.  To prove Theorem 3.2, we decompose the excess risk into approximation and stochastic errors. Our construction serves to control the approximation error, while our Theorem 3.2 shows that empirical risk minimisation achieves the minimax rate.
> > >
> > > When the exact empirical risk minimiser is not attained (e.g., when using gradient-based training), an additional optimisation error term appears in the excess risk; see, e.g., Schmidt-Hieber (2020). We will include a discussion about this in the final version of our paper. Providing guarantees for optimisation dynamics of deep transformers is a challenging and largely separate problem. As is common in statistical learning theory, we focus on the statistical and approximation aspects, and leave a detailed analysis of optimisation for future work. We will clarify this point in the final version.
> > >
> > >
> > >
> > > **References**
> > >
> > > Schmidt-Hieber, J., 2020. Nonparametric regression using deep neural networks with ReLU activation function. Annals of Statistics.

---

### Decision · Program_Chairs · 2026-04-30

**Decision:**

Accept (regular)

**Comment:**

The paper studies minimax optimality of in-context learning. The reviewers highlight that the results are interesting while the setting itself is somewhat artificial and does not link directly to how transformers behave in practice. Despite this criticism I believe that the paper will be of interest in the community and I suggest to accept the paper.